# Structural energetics of cold sensitivity

Kevin Y. Choi[1,2,3,5], Xiaoxuan Lin[4,5], Yifan Cheng[1,4 ✉] & David Julius[2 ✉]

Thermosensitive transient receptor potential (TRP) ion channels enable somatosensory nerve fibres to detect changes in our thermal environment over a wide physiologic range[1–3]. In mammals, the menthol receptor, TRPM8, is activated by temperatures below approximately 26 °C and is essential for the perception of cold or chemical cooling agents[4–6]. A fascinating, yet still unachieved goal is to elucidate mechanisms, both structural and thermodynamic, whereby TRPM8 or other thermosensitive channels are gated by changes in ambient temperature. Recent studies using cryogenic electron microscopy have attempted to address this challenging question but are limited by difficulties in visualizing temperature-evoked conformational sub-states or assessing the energetic landscape governing gating transitions[7,8]. Here we close this gap by combining cryogenic electron microscopy with hydrogen–deuterium exchange mass spectrometry to elucidate a mechanism for cold-evoked activation of TRPM8. First, we visualize TRPM8 channels in cellular membranes, where bona fide menthol- and cold-evoked open states are captured. We also identify a new 'semi-swapped' architecture in which interdigitation of channel sub-units is rearranged substantially following repositioning of the S6 transmembrane helix and elements of the pore region. We then use hydrogen–deuterium exchange mass spectrometry to pinpoint the pore and TRP helices as the regions exhibiting the greatest stimulus-evoked energetic changes that drive channel gating. Specifically, cold-evoked stabilization of the outer pore region repositions the pore lining S6 transmembrane helix while enabling binding of a regulatory lipid to stabilize the open channel. Structural mechanisms associated with activation are validated by comparison of human TRPM8 with the menthol-sensitive but relatively cold-insensitive avian orthologue. We propose a free energy landscape and conformational pathway whereby cold or cooling agents activate this thermosensory receptor.

Theoretical or experimental approaches have ascribed the unusually robust temperature sensitivity of transient receptor potential (TRP) channels to changes in heat capacity (for example, through solvation of buried hydrophobic residues) or partial protein unfolding transitions[9–11]. Mutagenesis and structural studies have identified numerous residues or regions that modulate TRP channel thermosensitivity or exhibit temperature-dependent conformational changes, indicating that temperature-evoked channel gating represents a distributive process[9,12–20]. At the same time, stimulus-evoked changes in heat capacity or protein folding may still be manifest within specific regions of the channel that are critical parts of a gating pathway. It now seems clear that identifying such regions and understanding their contributions to temperature-evoked channel gating will require the application of structural approaches together with methods that can assess the energetic landscape over specific temperature ranges.

On the structural front, biophysical properties of thermosensitive TRP channels indicate the existence of numerous sub-states associated with multi-modal regulation, adaptation or inherent thermal instability[21–23]. In the case of TRPM8, structural heterogeneity may

be exacerbated by the channel's cold sensitivity, resulting in a range of conformational sub-states that are too transient or numerous to register as principal structural subclasses during cryogenic electron microscopy (cryo-EM) analysis[24,25] (Extended Data Fig. 1 and Supplementary Table 1). Indeed, of the previously described TRPM8 structures[22,24–28], none convincingly represent a purely cold-evoked open state or provide structural insights into thermal gating mechanisms, which may reflect the fact that they have all been determined with detergent-extracted and solubilized material. Here we address this limitation by capturing new TRPM8 structures in cell-derived vesicles prepared without detergents[29,30].

On the thermodynamic front, previous studies of thermosensitive TRP channels (most notably the capsaicin- and heat-activated TRPV1 channel), have been carried out with methods that assess global energetics associated with temperature changes[10,11]. We now address this limitation by using hydrogen–deuterium exchange mass spectrometry (HDX–MS) to assess contributions from local regions and develop an energetic landscape across the TRPM8 protein sequence. Together, our results provide a structural and energetic framework to explain

[1]Department of Biochemistry and Biophysics, University of California San Franscisco, San Francisco, CA, USA. [2]Department of Physiology, University of California San Franscisco, San Francisco, CA, USA. [3]Chemistry and Chemical Biology Graduate Program, University of California San Francisco, San Francisco, CA, USA. [4]Howard Hughes Medical Institute, University of California San Francisco, San Francisco, CA, USA. [5]These authors contributed equally: Kevin Y. Choi, Xiaoxuan Lin. ✉e-mail: yifan.cheng@ucsf.edu; david.julius@ucsf.edu

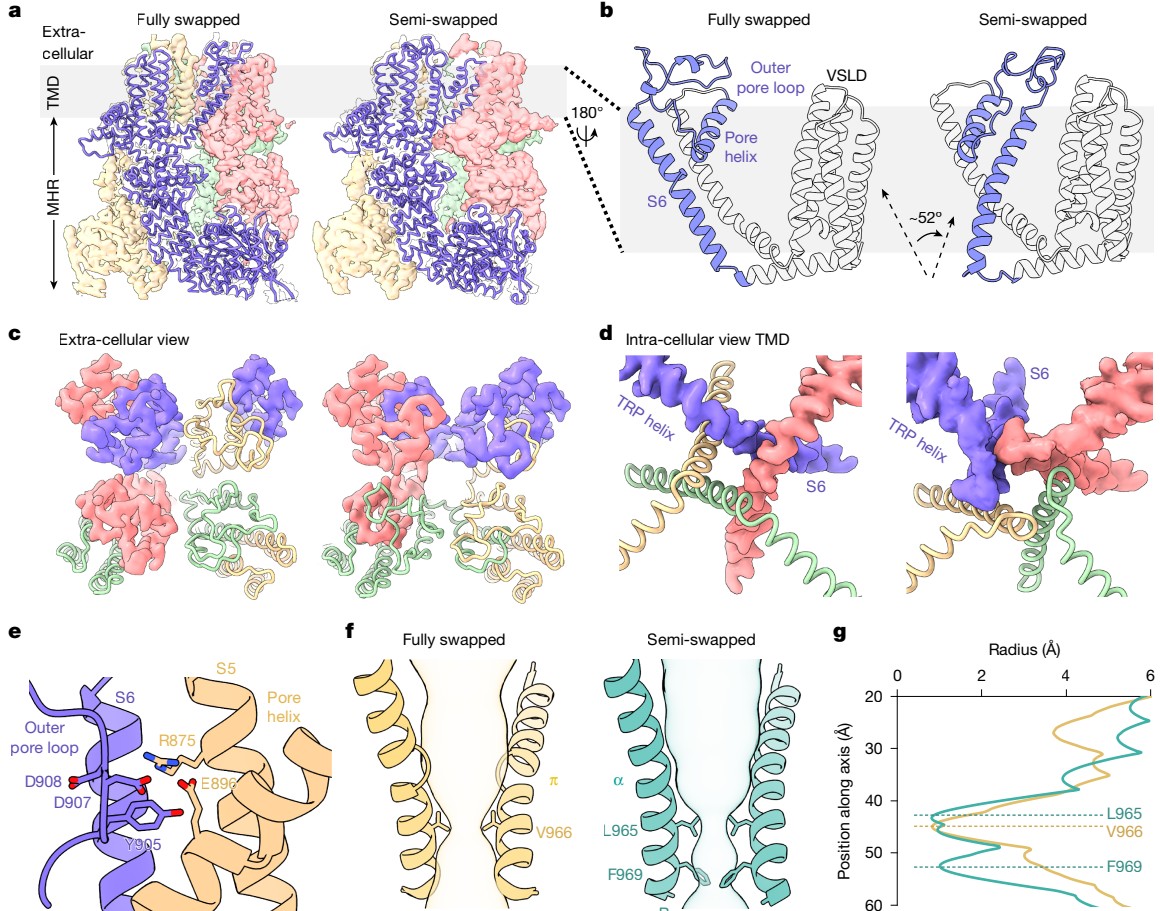

**Fig. 1 | Fully and semi-swapped configurations of avian TRPM8 in vesicles.**
**a**, Side view of fully (left) and semi-swapped (right) channel structures determined in cell membrane vesicles maintained at 4 °C before vitrification and in the absence of menthol or calcium. A ribbon diagram of one subunit is fitted into the density. TMD, transmembrane domain (shaded area); MHR, cytoplasmic melastatin homology region. **b**, Reconfiguration from fully (left) to semi-swapped (right) channel, illustrated for the transmembrane domain of one subunit, is characterized by approximately 52° bend of S6 at the junction with the TRP domain, repositioning the outer pore loops, but not the pore helix. **c**, Top view showing density and ribbon diagram model of outer pore loop for

fully (left) and semi-swapped (right) configuration. **d**, Close-up bottom views of S6–TRP helices in fully (left) and semi-swapped (right) configurations. Sub-unit colour scheme is the same for **a**, **c** and **d**. **e**, In the semi-swapped configuration, the pore domain is stabilized by extensive interactions between Y905 in the pore loop and charged residues in the pore helix and S5 of the adjacent subunit. **f**, A closed lower gate constriction is formed by V966 of S6 in the fully swapped configuration (π-helical form, left panel), or by L965 and F969 in the semi-swapped structure (fully α-helical form, right panel). **g**, Pore profiles of closed gate in fully and semi-swapped structure.

TRPM8 cold sensitivity while demonstrating the power of combining structural analysis with thermodynamic measurements to reveal a conformational landscape underlying physiologic function.

## New TRPM8 architecture in cell vesicles

To visualize TRPM8 in cellular bilayers, we began with an avian ortho-logue for which detailed structural information already exists[24,31]. HEK293 cells heterologously expressing the *Parus major* (*Pm*) channel bearing an N-terminal green fluorescent protein (GFP) tag were sonicated and inside-out vesicles enriched by affinity chromatography. Our initial analysis was stymied by poor signal-to-noise, which we then mitigated by introducing a second purification step using size exclusion chromatography, yielding micrographs with greatly improved contrast and sample distribution (Supplementary Fig. 1). These raw micrographs showed clear features of TRPM8 in a cellular bilayer, which, with further processing, yielded featured two-dimensional class averages that confirmed the presence of cytoplasmic elements facing outward from the vesicle (Supplementary Fig. 1). From these data we obtained high resolution (3–3.5 Å) structures, including the previously described desensitized closed states[24,27,32] (Extended Data

Fig. 2 and Supplementary Tables 1 and 2), characterized by a homo-tetrameric arrangement in which each subunit consists of an N-terminal cytoplasmic domain followed by six transmembrane helices (S1–S6) arranged in a classic domain swap architecture with the central ion permeation pathway formed by S5 and S6 helices together with the intervening pore helix and outer pore loop.

In addition to these familiar structures, we identified a new sub-state whose most notable feature is a distinct domain swap architecture (Fig. 1a). In this configuration, the S6 helix and the associated pore loop are not interdigitated with the neighbouring subunit but instead remain associated with their cognate S1–S4 voltage sensor-like domain (VSLD). This new 'semi-swapped' configuration reflects a large reposi-tioning of the S6 helix such that it bends by around 52° to replace the S6 helix from a neighbouring subunit that otherwise engages in the canonical domain swap architecture (henceforth referred to as 'fully swapped') (Fig. 1b). These changes within the pore region are accom-panied by exuberant movements in the associated outer pore loops, whereas the relative positions of the pore helices remain unchanged (Fig. 1b). Critical evidence for the semi-swapped configuration is pro-vided by clear densities demonstrating reconfiguration of the connect-ing regions on both ends of the S6 helix: the pore helix and pore loop

of one subunit separate to form the pore domain in conjunction with the neighbouring subunit, whereas the linker between the S6 and TRP helices adopts an acutely bent conformation (Fig. 1c,d). Transitions between the fully and semi-swapped configurations do not require tetramer disassembly (Supplementary Video 1).

Another notable feature of the semi-swapped configuration can be seen at the inter-protomer interface, where the S6 helix and outer pore loop of one subunit engages with S5 and pore helices of the neighbouring subunit such that Y905 is buried in an amphipathic pocket (Fig. 1e). This new semi-swapped configuration differs not only from the canonical domain-swapped architecture, but also from previously described non-swapped configurations seen in tetrameric voltage-gated potassium channels[33,34]. Using the cell-derived vesicle approach, we observed both fully- and semi-swapped structures in the absence or presence of calcium (5 mM) and found that human TRPM8 adopts these same configurations (Supplementary Table 1), demonstrating a conserved architecture across avian and mammalian species.

Notably, in the semi-swapped configuration, a π- to α-helical transition occurs in the vicinity of N958, introducing a register shift by a single residue in the lower half of the S6 helix (Fig. 1f), reminiscent of an agonist-induced gating reconfiguration seen in other TRP channels that is achieved through a π to α transition[23,35]. Consequently, the lower hydrophobic restriction, formed by V966 in the fully swapped configuration, is now constituted by L965 and F969 in the semi-swapped state (Fig. 1f,g). This reorientation of the S6 helix is accompanied by repositioning Y905 at the selectivity filter, thereby further widening this region (Fig. 1g).

In our previous structures of *Pm* TRPM8 (ref. 24), we observed both α and π configurations in S6, which we attributed to closed and desensitized states, respectively, as the π configuration was seen only in the presence of calcium, which is required for functional desensitization. In our vesicle preparation we find that the S6 π configuration is seen only in the fully swapped architecture with or without calcium (Supplementary Table 1), indicating that this state, previously designated as desensitized, may represent a distinct closed state or a desensitized state that persists after removal of calcium.

## Menthol modulates equilibrium between states

Among the population of well-resolved avian TRPM8 particles, we observed a decrease in those corresponding to the fully swapped configuration (with a concomitant increase in the semi-swapped configuration) in the presence of menthol (Fig. 2a), indicating that interconversion between these states occurs and is physiologically relevant. To further probe the native state ensemble of TRPM8, we performed HDX–MS experiments to assess both intrinsic and ligand-dependent dynamics of purified, detergent-solubilized avian (*Pm*) or human (*Homo sapiens* (*Hs*)) channels (Supplementary Table 3). The HDX rate measures the kinetics of exchange between backbone amide hydrogens and the solvent deuterium. Exchange is governed by transient breakage of hydrogen bonds during thermal fluctuation; hence, HDX–MS provides a direct readout of protein dynamics and the underlying folding free energy ($\Delta G$) that can be mapped to specific residues[36,37].

In these experiments, most peptides derived from proteolysis following H–D exchange exhibited unimodal mass distributions, indicative of a single dominant conformation within the corresponding regions (Fig. 2b and Supplementary Figs. 2 and 3). In contrast, peptides covering the pore and TRP helices each showed clear bimodal mass envelopes that increased in mass over labelling time, indicating the coexistence of two conformational populations[38] (Fig. 2c,d and Supplementary Figs. 3–6). One mass envelope was observed to exchange more than 100-fold faster than the other, corresponding to a difference in folding free energy of at least 3 kcal mol⁻¹ (Methods). Notably, the ratio of heavier to lighter mass envelopes increased progressively over labelling time, indicating that these two populations undergo interconversion

on the HDX–MS time scale (minutes to hours) (Fig. 2b–d and Supplementary Figs. 4–6).

The regions exhibiting bimodality (pore and TRP helices) align closely with those predicted to undergo large conformational transitions between our fully and semi-swapped models, leading us to propose that the two mass envelopes reflect deuterium uptake associated with these two structural states. Given the additional stabilizing interactions formed at the outer pore region in the semi-swapped configuration of the avian channel (Fig. 1e), we infer that the slower-exchanging population represents the semi-swapped state, whereas the faster-exchanging population corresponds to the fully swapped state. This interpretation is corroborated further by the observation that addition of menthol increased the proportion of the slower-exchanging population (semi-swapped) (Fig. 2e), consistent with the conformational redistribution observed in single-particle cryo-EM.

## Menthol stabilizes the TRP helix

We resolved structures of human and avian TRPM8 channels in vesicles in the presence of menthol (Supplementary Table 1). These maps enabled us to model the VSLD (proposed binding site for menthol and other channel modulators)[24,28,39,40], thereby revealing a menthol-like density (absent in unliganded datasets) (Fig. 3a,b and Extended Data Fig. 3). When docked and refined into the cryo-EM density, our models reveal several key interactions between the lower cavity of the VSLD and menthol (Fig. 3b). For example, the hydroxyl group of menthol forms close contact with arginine (R832 in avian or R842 in human TRPM8) (2.9 Å), which is within hydrogen bonding distance. A network of hydrophobic side chains along the inner cavity of the VSLD forms extensive interactions with menthol, probably contributing to its stabilization within the pocket (Fig. 3b). Compared with the closed, unliganded structures, binding of menthol is accompanied by repositioning of the TRP helix such that it is in closer proximity to the VSLD (Extended Data Fig. 3), yielding an open channel (Extended Data Fig. 8b; see also section 'A cold-activated TRPM8 structure' below).

To investigate the thermodynamic basis driving the menthol-modulated dynamic equilibrium, we performed differential HDX to identify residues showing changes in deuterium labelling in the presence of menthol. Exchange rates for peptides covering the TRP helix slowed by tenfold, corresponding to a stabilization energy of 1.4 kcal mol⁻¹, whereas HDX for the remaining transmembrane domain was unchanged (Fig. 3c,d). This stabilization of the TRP helix includes Y995 (or Y1005 for the human channel), a residue that interacts directly with menthol in our structural models, highlighting the TRP helix as a main energetic contributor to ligand-induced conformational changes. Agonist-induced stabilization was seen whether menthol was added to purified TRPM8 protein or to whole cells before protein extraction, indicating that the observed changes reflect intrinsic properties of the channel rather than alterations in cellular or membrane properties (Extended Data Fig. 4).

## Structural basis of differential cold sensitivity

TRPM8 channels show pronounced species-specific differences in thermal sensitivity, with avian orthologues being relatively insensitive to cold compared with their mammalian counterparts[41,42] (Supplementary Fig. 7). Despite this functional divergence, the two orthologues share high similarity in both sequence (92%) and structure (r.m.s.d. = 1.0 Å for PDB IDs 6O6A and 8BDC). However, our structural analyses in vesicles, as well as HDX–MS measurements, show that the human channel exhibits much greater heterogeneity than appreciated previously. We therefore propose that differential cold sensitivity between mammalian and avian species arises from distinct thermodynamic properties rather than principal structural differences in the low-energy states.

Building on this rationale, we sought to identify regions within the mammalian channel that exhibit temperature-induced dynamics

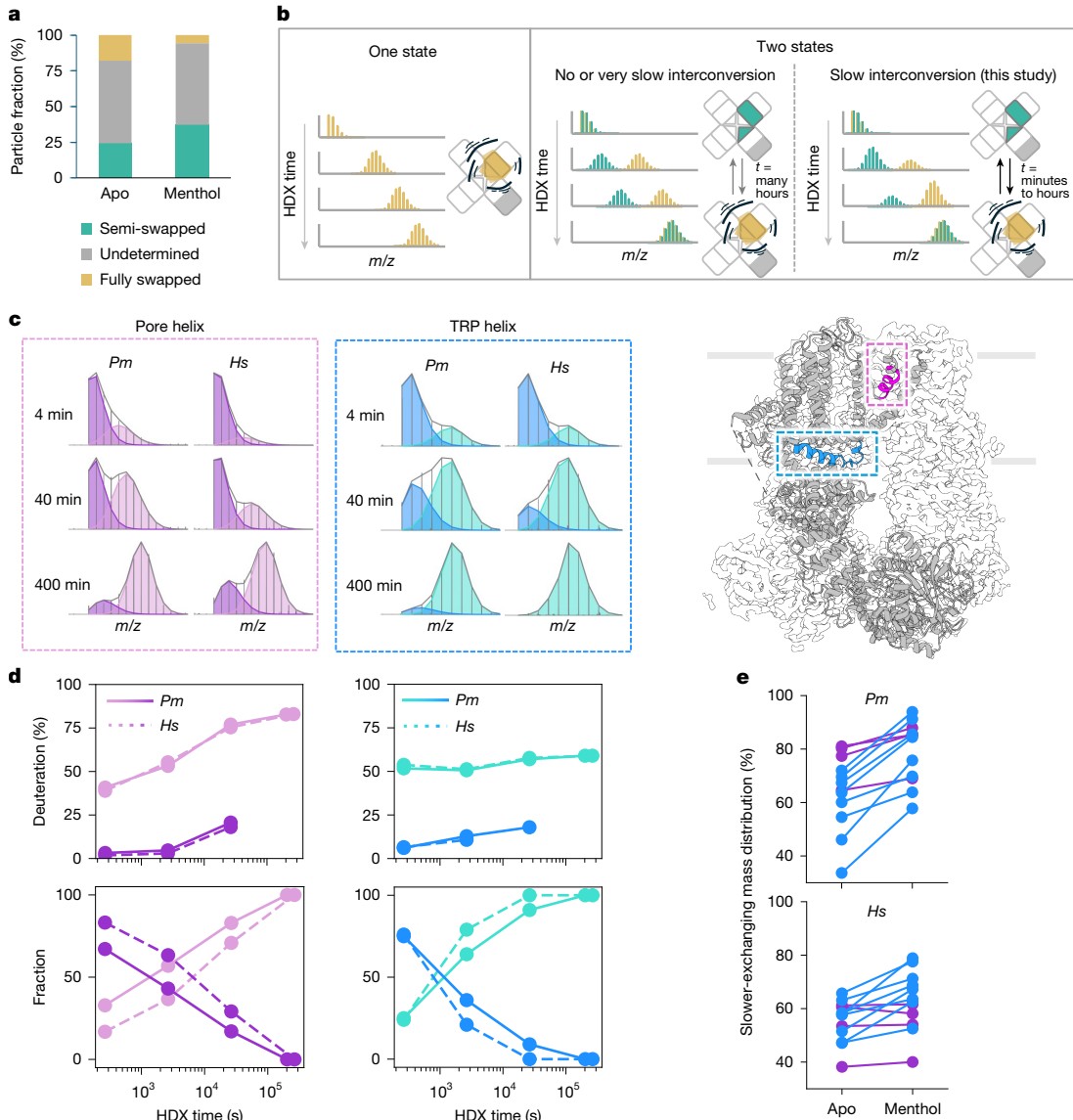

**Fig. 2 | Menthol modulates the dynamic equilibrium of TRPM8. a**, Particle distribution in the apo or menthol-treated pmTRPM8 datasets corresponding to semi-swapped (green; 24% apo, 37% menthol), undetermined (grey; 58% apo, 57% menthol) and fully swapped (yellow; 18% apo, 6% menthol) configurations. Samples were incubated at 4 °C for 2–15 min with or without menthol (1 mM) before vitrification. Particles categorized as 'undetermined' refer to those for which resolution of key regions that distinguish fully from semi-swapped configurations was insufficient for unambiguous categorization. **b**, Schematic depicting mass distributions for a deuterated peptide. Left, a 'one state' scenario in which peptides with a single conformation or rapid interconversion between conformations show unimodal mass distributions. Right, a 'two state' scenario in which peptides with two conformations show bimodal mass distributions, with intensity shifts reflecting interconversion rate between the conformations. **c**, Representative mass spectra of peptides from the pore (purple) and TRP (blue) helices (coloured in the pmTRPM8 semi-swapped structure, right panel) over HDX times at 4 °C for apo avian (*Pm*) and human (*Hs*) channels show bimodal mass distributions (dark/light) characteristic of the coexistence of two slowly interconverting populations. Peptides correspond to residues 891–901 (*Pm*) and 901–911 (*Hs*) for pore helix, and 982–989 (*Pm*) and 992–999 (*Hs*) for TRP helix. **d**, Kinetics of deuterium uptake (top) and fraction change (bottom) for the interconverting populations of peptides described in **c**. **e**, Fraction of the slower-exchanging mass distribution (dark purple or dark-blue peaks in **c**; presumptive semi-swapped configuration) increases with menthol at 22 °C ($t_{HDX}$ = 30 s), demonstrating agonist-dependent population redistribution.

distinct from its avian counterpart using temperature-dependent HDX–MS coupled with van't Hoff analysis. To do so, we performed HDX on avian and human TRPM8 at four temperatures spanning the activation threshold of around 26 °C for the human channel and converted HDX rates to $\Delta G$ values (Extended Data Fig. 5; Methods). We noted that many regions showed increased stability (more negative $\Delta G$) as temperature decreased, indicating enthalpically driven stabilization (Extended Data Fig. 5a). Van't Hoff analysis showed nonlinearity in many regions of human TRPM8, indicating changes in heat capacity ($\Delta C_p$) over the experimental temperature ranges (Extended Data Fig. 5b). In contrast, van't Hoff analysis for avian TRPM8 showed weak non-linear trends, indicating an almost constant $C_p$. We then calculated the standard folding enthalpy ($\Delta H°$) within each temperature window (37–30 °C, 30–22 °C and 22–4 °C), which enabled us to generate sequence-wide, residue-level enthalpic profiles for both orthologues (Fig. 4a). We noted that, although the overall energetics ($\Delta G$) for many regions were similar between orthologues—often differing by less than 1 $k_BT$ (spontaneous thermal fluctuation energy)—key differences emerged when examining the enthalpic contribution ($\Delta H°$) to folding energy. We found that many regions in the human channel showed greater changes in $\Delta H°$ than the avian channel, highlighting the prevalence of local $\Delta C_p$ and temperature-induced dynamics (Fig. 4a).

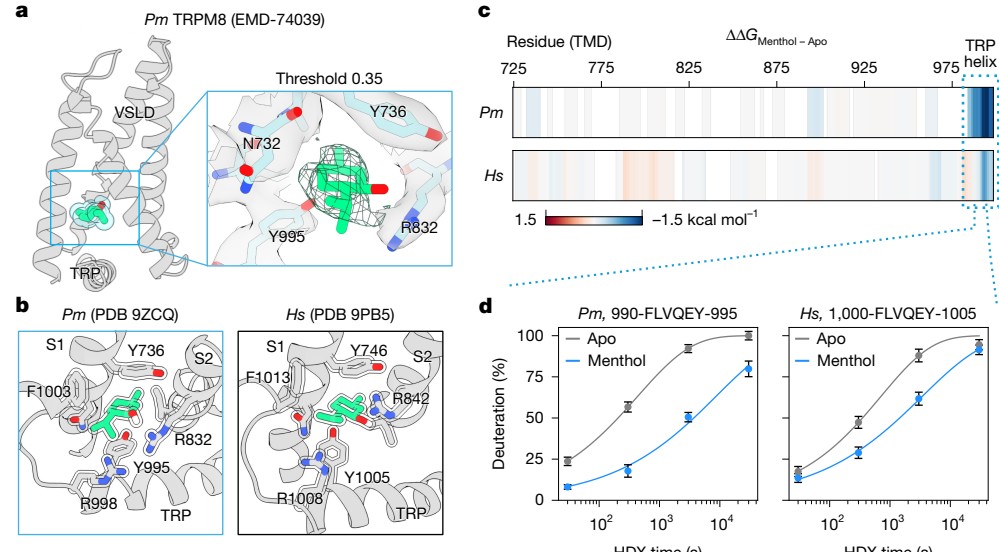

**Fig. 3 | Menthol binding stabilizes TRP helix. a**, Ribbon diagram of VSLD indicating location of bound menthol molecule with enlarged view highlighting density of ligand (mesh) and binding pocket (transparent surface). **b**, Atomic details of menthol-binding pocket in avian (left) or human (right) channel showing sidechains forming direct interactions with menthol. Residues are annotated. **c**, Changes in folding free energy ($\Delta\Delta G$) for TRPM8 upon menthol binding at 22 °C. Residue numbers correspond to the avian sequence. **d**, Kinetics of deuterium uptake for the TRP helix indicate that menthol binding stabilizes this region, more so for the avian than the human channel. Solid lines, stretched-exponential fit of the uptake data. Error bars, s.d. of triplicates.

These observations are consistent with the heat capacity model for TRP channel gating described previously[9]. We therefore expected that regions providing the main energetic driving force to channel opening should exhibit the most pronounced decrease in standard folding enthalpy upon cooling (a positive $\Delta C_p$) across the channel's activation threshold. Our enthalpic profiles pinpoint the pore helix of human TPRM8 as such a region (Fig. 4a). The HDX of the pore helix slowed upon cooling, more dramatically than the rest of the protein or the avian orthologue, reflecting increased local stability with decreased temperature (Fig. 4b). Van't Hoff analysis showed that the pore helix of the human channel exhibited a pronounced decrease in $\Delta H°$ within the temperature window covering the activation threshold (30–22 °C), whereas such an effect was much weaker in the avian channel (Fig. 4c,d). The cold-induced enthalpic stabilization of the pore helix demonstrates that this region plays a principal role in conferring cold sensitivity to the human channel.

It has been shown that a single amino acid within the outer pore region is linked to species-specific cold sensitivity[41] (Supplementary Fig. 7). This residue is invariantly a tyrosine in birds (Y905 for *Pm*) and valine in mammals (V915 for *Hs*) (Fig. 4e). We see that, in the semi-swapped avian channel, Y905 is nestled within a pocket lined by a preponderance of charged amino acids (R875, E896, D907, D908) (Fig. 1e), predicting that this interaction, and thus cold sensitivity of the avian channel, can be modulated by increasing extracellular pH. Indeed, calculated pKa values for D907 and D908 (8.16 and 5.43, respectively) are shifted from their standard values (3.8) or calculated values in the fully swapped configuration (5.26 and 3.92) (Extended Data Fig. 6). Consistent with this, we found that cold-evoked responses of HEK293 cells expressing the *Pm* channel were enhanced markedly as the pH of the perfusate was increased from 7.4 to 9.0 (Fig. 4f and Supplementary Fig. 7). In contrast, cells expressing the human channel showed robust cold-evoked responses throughout this same pH range, with a small but significant increase observed at higher pH, consistent with the observation that high extracellular pH weakly modulates mammalian TRPM8 cold sensitivity[43] (Fig. 4f). Furthermore, exchanging valine with tyrosine (V915Y) converted cold sensitivity of mammalian TRPM8 to more closely resemble that of the avian channel (Fig. 4f and Supplementary Fig. 7). This also resulted in a more stable semi-swapped mammalian channel that can be better resolved with cryo-EM (Extended Data Fig. 2 and Supplementary Table 1). Conversely, the Y905V avian channel mutant showed enhanced cold sensitivity—a shift that was even more robust when the entire pore loop consisting of 34 amino acids from human TRPM8 (V915 to L948) was substituted for the cognate region of the *Pm* channel (Extended Data Fig. 7 and Supplementary Fig. 7). Consistent with these functional results, this chimeric channel shows enthalpic stabilization in between wild-type avian and human channels (Extended Data Fig. 7). Together, these findings pinpoint the outer pore region as a principal nexus for structural and energetic determinants of temperature sensitivity.

## A cold-activated TRPM8 structure

To understand how temperature-dependent energetic changes drive conformational changes, it is necessary to visualize TRPM8 in a cold-evoked open state. Our calcium imaging experiments indicate that destabilizing the species-specific Y905 interface potentiates cold-evoked responses of the avian channel (Fig. 4f). Indeed, under conditions that favour this state (high pH, 4 °C), we were able to determine an open state structure in the semi-swapped configuration using detergent-purified channel protein (Fig. 5). Compared with the semi-swapped structure obtained at physiologic pH, where the constriction is formed by two residues (L965 and F969) (Fig. 1f), the S6 helix undergoes an upward translation by one helical turn such that F969 alone now forms the lower constriction (Fig. 5a,b). Concomitantly, the F969 sidechain is reoriented perpendicular to the plane of the membrane, resulting in a widening of the pore from less than 0.5 to around 2 Å in radius. Furthermore, we observed a strong cation-like density coordinated in the centre of the F969 side chains, which we interpret as calcium passing through the gate as it is absent with EGTA (Fig. 5a,c and Extended Data Fig. 8). These observations indicate that the structure determined at high pH and low temperature corresponds to a bona fide cold-evoked open state, where F969 can coordinate permeating ions through a π-cation cage-like interaction. Such a lower gate configuration is not unprecedented; for example, the calcium-selective channel TRPV5 forms a permeation pathway involving an analogous tetrameric arrangement of tryptophan residues[44]. This configuration differs from

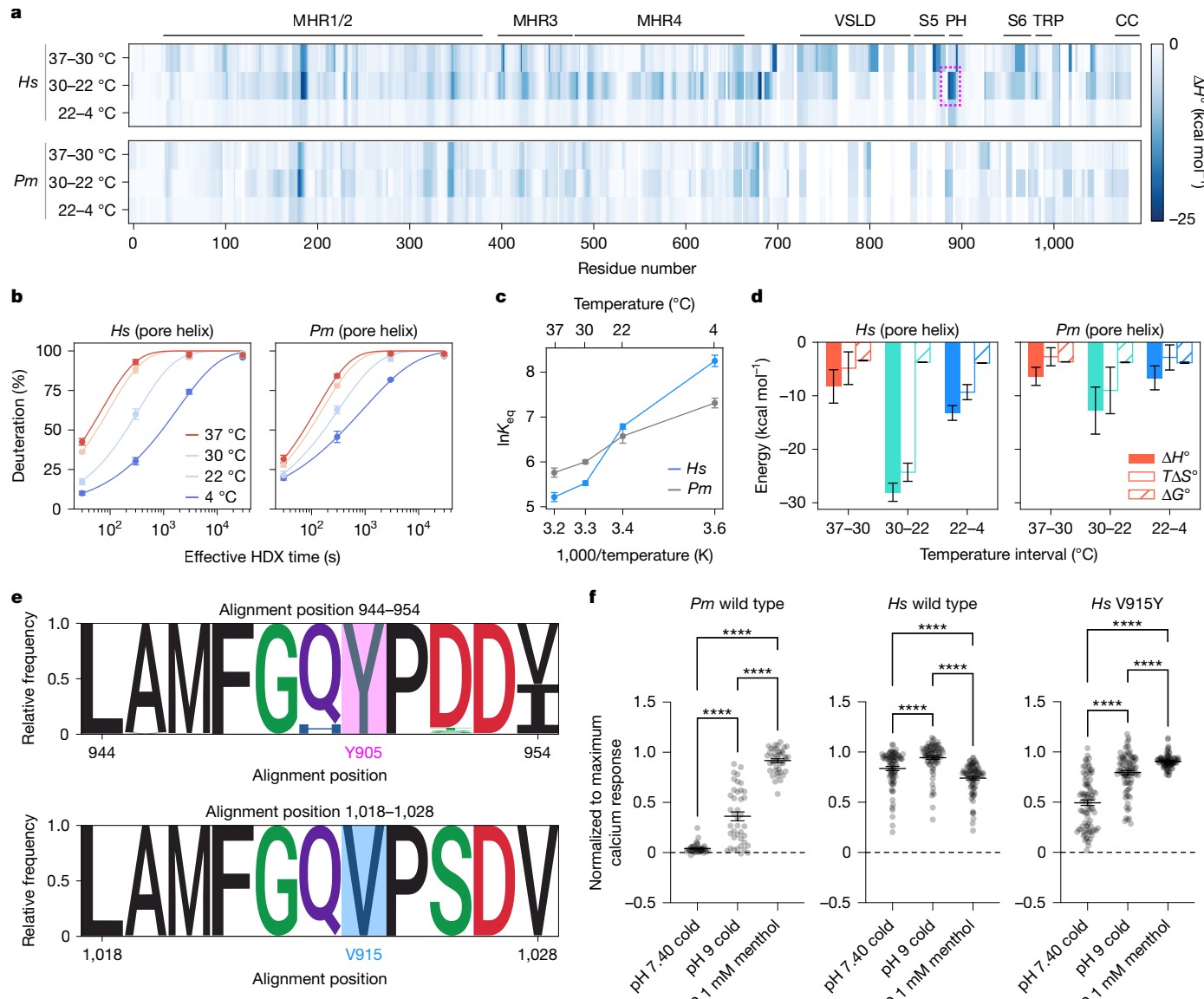

**Fig. 4 | The outer pore region confers species-dependent cold sensitivity.**
**a**, Standard folding enthalpy ($\Delta H°$) for *Hs* or *Pm* TRPM8 across three temperature ranges: 37–30 °C, 30–22 °C and 22–4 °C. The coloured box highlights that the pore helix (PH) in the human channel exhibits a greater $\Delta H°$ at the temperature range spanning the channel's activation threshold (30–22 °C). This feature is absent in the avian channel and at other temperature ranges. Residue numbers correspond to the avian sequence. **b**, Kinetics of deuterium uptake for representative peptides in pore helix at different temperatures. The peptides correspond to the region highlighted in **a** (*Hs*, residues 901–911; *Pm*, residues 891–901). Solid lines, stretched-exponential fit of the uptake data. Effective HDX times represent labelling times after correcting for the temperature effect on $k_{chem}$. **c**, Van't Hoff analysis of peptides in **b**. **d**, Enthalpy–entropy compensation of the pore helix upon temperature changes. Thermodynamic parameters in

standard state (25 °C and 1 atm pressure) are derived from the same peptides in **b** and **c**. Error bars in **b**–**d**, s.d. of triplicates. **e**, Sequence logo of Sauropsida (avian, top) and mammalian TRPM8 (bottom) at the outer pore region (corresponding to Y905 for *Pm* or V915 for *Hs* TRPM8). Relative frequency of amino acids at alignment positions are represented on the *y* axis. **f**, HEK293T cells expressing avian or human TRPM8, or the 'avian-ized' human V915Y TRPM8 mutant, were exposed to various stimuli, as indicated, and responses measured with Fura-2 ratiometric calcium dye (normalized to maximal calcium response following addition of 10 µM ionomycin). Each dot represents a single cell; $n = 40$ for *Pm*, 79 for *Hs* and 86 for *Hs* (V915Y). Multi-measure one-way analysis of variance with Tukey's post hoc analysis; ***$P < 0.001$, ****$P < 0.0001$. ln$K_{eq}$, natural logarithm of $K_{eq}$.

a previously described open state of mouse TRPM8 obtained in the presence of a cooling compound (cryosim-3) plus an electrophilic agent (allyl isothiocyanate) and soluble di-C8-phosphoinoside(PI)P$_2$, in which the gate is formed by a valine residue in S6 with a π-helical configuration[22].

How is destabilization of the Y905 interface coupled with channel gating? Our structure shows that loss of this interface results in exposure of buried hydrophobic residues at the S5–S6 region that are instead stabilized by a presumptive lipid acyl chain invading this cleft (Fig. 5a,d and Extended Data Fig. 8). The corresponding density

connects to a lipid headgroup occupying a binding site for di-C8-PIP$_2$ described previously[22,28], located at the cleft between the pre-S1 region and the VSLD (Fig. 5d). We therefore designate this density as representing a PI(4,5)-bisphosphate lipid, consistent with the finding that endogenous phosphoinositides co-purify with TRPM8 in the absence of charged amphiphiles[22] and that PIP$_2$ is required for channel activation[45,46]. When modelled, the length of the protruding acyl chain corresponds to 18:0-20:4 PIP$_2$, which is the predominant, biologically active species in mammalian plasma membranes[47,48]. Although the position of the S6 helix fluctuates in the absence of stimuli, it is stabilized at a

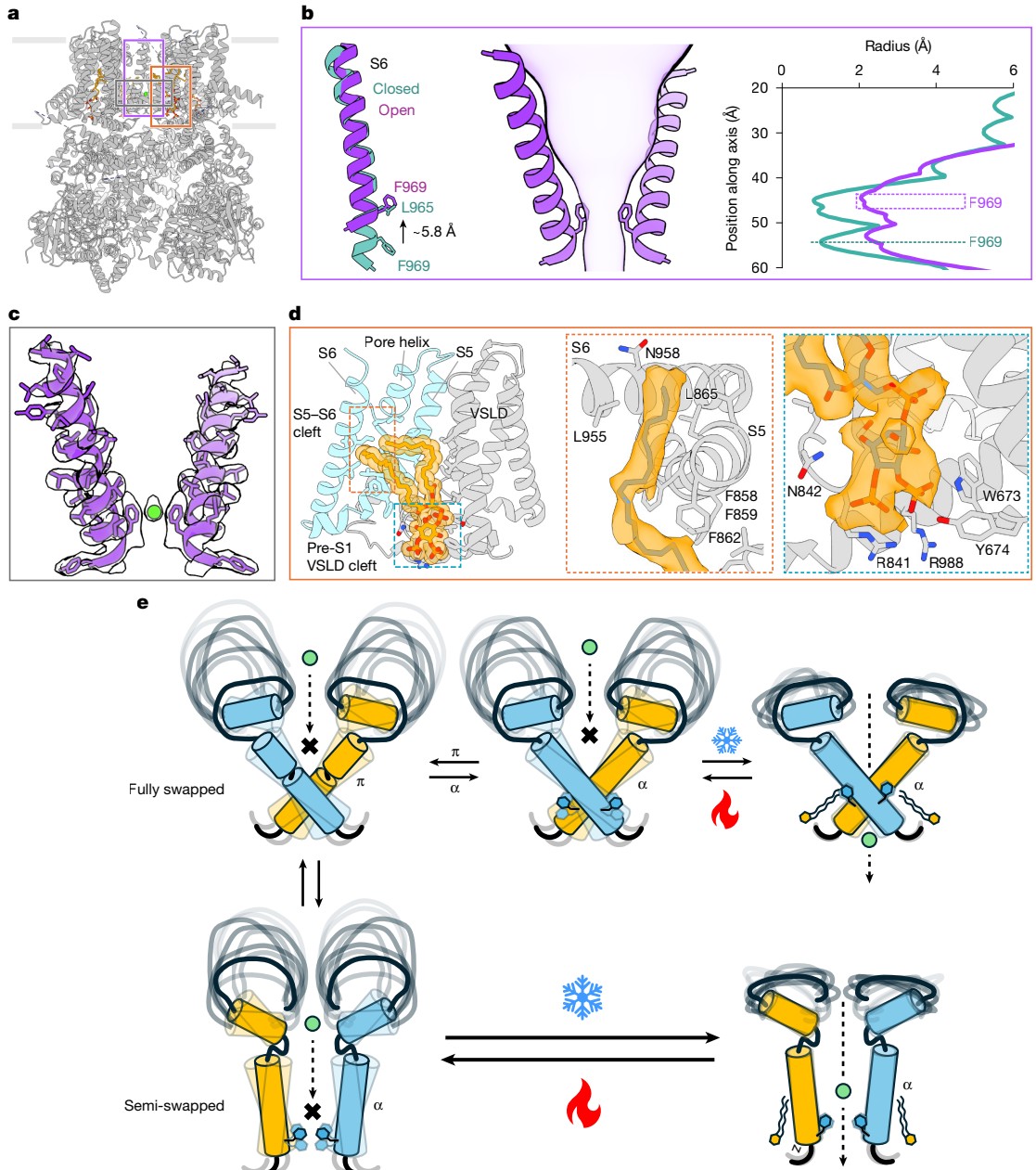

**Fig. 5 | Activation of TRPM8 by cold. a**, Structural overview of wild-type *Pm* TRPM8 in semi-swapped configuration with an open pore determined at 4 °C, pH 9. **b**, The S6 helix in the open semi-swapped structure translates upward by one helical turn when compared with the closed semi-swapped structure; this is accompanied by sidechain rotation of F969 (left). An open lower gate constriction is now formed by F969 (middle and right). **c**, Density and model of the S6 helix with a coordinated ion in the open lower gate. **d**, PIP₂ lipid docked into cryo-EM density with close-ups of the hydrophobic S5–S6 cleft and interactions between the PIP₂ headgroup and side chains within the pre-S1 and

VSLD region. **e**, Schematic model depicting two parallel conformational pathways of cold-evoked TRPM8 opening in fully (upper) and semi-swapped (lower) configurations. Closed channels exhibit intrinsic interconversions between the two configurations. Although S6 of the semi-swapped closed channel is in α form, S6 of the fully swapped closed channel undergoes π and α transitions. In both fully and semi-swapped configurations, cold stabilizes the outer pore region, repositions the α form of S6, allowing the tail of regulatory PIP₂ lipids to stabilize a conformation where the side chain of a phenylalanine at lower S6 reorients upwards to form a π cage for cation permeation.

position upshifted by one helical turn by the full-length arachidonyl lipid tail of the endogenous PIP₂. Consequently, L955 occupies the lipid-contacting cleft in place of N958 and the acyl chain is nestled in a pocket formed exclusively by hydrophobic residues (Fig. 5d). Furthermore, this helical register shift repositions F969 such that the side chain flips upward, disrupting the tight hydrophobic seal seen in the closed state (Fig. 5b,c). As in the case of TRPV1 (ref. 49), this analysis further highlights the importance of examining structures in the presence of native phosphoinositide lipids, rather than soluble mimetics such as di-C8-PIP₂.

To mitigate concerns about non-specific effects of high pH, we also determined the structure of the chimeric avian channel containing the human pore loop under activating conditions (4 °C at neutral pH). Here we identified a structural subclass representing the same open gate configuration as seen with the wild-type avian channel at 4 °C and pH 9 (Extended Data Fig. 7c,d). Furthermore, the outer pore region of this chimera showed thermodynamic properties closer to those of the wild-type human channel (Extended Data Fig. 7b).

We also determined the structure of human TRPM8 at 4 °C and neutral pH in vesicles (following pre-incubation at 4 °C or 37 °C), where we

observed similar features, including an unresolved outer pore loop interface and open gate configuration. We see both fully and semi-swapped configurations, but the open state is modelled clearly only in the former, presumably reflecting the energetic and structural heterogeneity of mammalian orthologues in the semi-swapped configuration (Extended Data Figs. 2 and 8 and Supplementary Table 1). In all cases, S6 adopts the same α-helical conformation as seen in the avian open channel as well as a presumptive acyl chain density in the S5–S6 cleft (Extended Data Fig. 8). We hypothesize that because the Y905 interface seen in avian TRPM8 is absent in the human channel, binding of PIP$_2$ occurs more readily at lower temperatures, directly facilitating similar gating rearrangements. At higher temperatures, the intrinsically greater dynamics of the mammalian channel may destabilize lipid binding to disfavour the open state. The same open configuration was observed for either the avian or human channel in the presence of menthol, indicating that although cold and menthol act at opposing ends of the S6 helix, the resulting structural rearrangements converge on a common gating mechanism (Extended Data Fig. 8 and Supplementary Table 1).

## Discussion

Cryo-EM analysis of membrane proteins has evolved to enable the study of receptors and ion channels without having to first extract them from their cell membrane environment. In this study, we demonstrate the utility of this approach for capturing conformational states of TRPM8 previously unresolved by us or others using detergent-extracted protein. We demonstrate that the semi-swapped architecture inter-converts with the canonical fully swapped state in a dynamic equilibrium that is biased by menthol or cold. This was achieved by combining cryo-EM analysis of cell-derived vesicles with energetic measurements of purified proteins using HDX–MS.

Our findings are consistent with an emerging consensus that the dynamic nature of the outer pore and specific protein–lipid interactions are critical factors in specifying the physiological properties of thermosensitive TRP channels[12,14,20,23,50]. Differential cold sensitivity appears to originate from measurable differences in $\Delta C_p$ involving the outer pore loop interface, which is consistent with corresponding structural differences observed for the human and avian channels. This indicates that the outer pore region of mammalian channels evolved with a broader energetic landscape suitable for sensing temperature over a physiologically relevant thermal range. In any case, our structures of avian and human TRPM8 indicate that the functional consequence of outer pore loop dynamics is to initiate reconfiguration of the S6 helix and regulate accessibility of a hydrophobic tunnel by the long acyl chain of PIP$_2$ as a key regulatory factor (Fig. 5e). In this regard, PI lipids play an essential role in modulating the activity of numerous thermosensitive TRP channels[51], which is now borne out by this and other structural studies demonstrating how the binding of PIP$_2$ or other PI species facilitate (for example, TRPM8) or inhibit (for example, TRPV1) temperature activation[49].

Our results further indicate that opening of the lower gate requires that S6 adopt a full α-helical configuration, which, in the avian channel, is stabilized in the semi-swapped state. In contrast, a π-helical configuration is observed only in the fully swapped channel structure, probably posing a higher energetic barrier to gating. In any case, a unifying observation is that gate opening requires transition from the π to the α configuration, which can occur in the fully swapped configuration by a single-residue register shift in the lower half of the S6 helix, akin to what is seen in TRPV1, or during transition from fully to semi-swapped configurations. Although our data indicate that the fully swapped channel has a higher energetic barrier to making the gating transition, the transition between fully and semi-swapped states is not the gating step per se. Rather, each state can converge on the same open S6 configuration. Together with energetic measurements by HDX, we propose a free energy landscape describing the conformational states

observed structurally by cryo-EM (Extended Data Fig. 9). In the case of avian TRPM8, the stabilizing Y905 interface (Fig. 1e) traps the channel kinetically in a closed semi-swapped state, making the channel insensitive to cold. However, avian TRPM8 channels do show some weak cold sensitivity, which our results indicate arises from the opening of channels in the fully swapped state, where the Y905 interface is not formed. Thus, the existence of the semi-swapped state provides a molecular explanation for species-specific differences in cold sensitivity, at least in regard to birds versus mammals.

Our structures indicate that the interconversion between fully and semi-swapped states requires the movement of pore domains from all four sub-units, which can occur within the confines of the intact tetrameric architecture (Supplementary Video 1). As such, it is not surprising that reconstructions from a significant proportion of particles in our datasets are characterized by an unresolved pore domain within an otherwise well-resolved full-length channel structure. This phenomenon may explain why previously published[25] and numerous unpublished TRPM8 structures from our own studies lack resolution in this region and thus failed to resolve significant conformational changes associated with agonist binding. Such reconstructions are often discarded from further analysis, but our results indicate that they may reflect functionally relevant, dynamic movements. This realization motivates an emerging goal in single-particle cryo-EM, namely, to understand the dynamic processes underlying physiologic function, rather than simply pursuing the highest resolution structures. Similarly, energetic measurements using techniques such as differential scanning calorimetry have provided valuable insights into enthalpic changes associated with TRP channel gating but do so for the channel in toto. In contrast, HDX–MS provides energetic information about specific protein segments, which, in combination with recent advances in cryo-EM, brings us closer to achieving a holistic analysis of temperature-sensitive ion channel function, as we demonstrate here for the cold sensor, TRPM8.

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

## Methods

### Cloning, cell culture and protein expression

N-terminally fused mEGFP constructs of TRPM8 were obtained using Gibson assembly to sub-clone TRPM8 genes into a pFastBac1 vector. Mutagenesis was carried out by either PCR with mutagenic primers, or by Gibson assembly, with Q5 high-fidelity polymerase. Construct sequences were verified using Sanger sequencing and whole-plasmid sequencing. Expi293F cells were maintained at 37 °C with 8% ambient $CO_2$ and grown to a density of $3–3.5 \times 10^6$ cells ml$^{-1}$. To transduce cells for protein overexpression, a baculovirus (generated from two rounds of viral amplification in SF9 cells, according to known protocols) for the N-terminally fused mEGFP *Pm* or *Hs* TRPM8 with a human rhinovirus 3C proteolysis site was added dropwise, for a final concentration of 5–10% (vol%/vol%) to the flask while gently mixing the culture. Baculoviruses used in mammalian cell cultures were sequenced routinely by amplifying the coding region using PCR (M13 primer sites), gel extracting the amplicon and subjecting the purified DNA fragments to Sanger sequencing. Virally transduced Expi293F cultures were left at 37 °C with 8% $CO_2$ for 16 h. To boost expression, the cultures were supplemented with 10 mM sodium butyrate and transferred to an incubator at 30 °C with 5% $CO_2$ and allowed to incubate for a total of 72 h (post-transduction). Cells were then collected by centrifugation (3,000*g*, 10 min, 4 °C), washed once with 1× Dulbecco's phosphate buffered saline (DPBS) pH 7.40 by gently resuspending the cell pellet, and collected by centrifugation (3,000*g*, 10 min, 4 °C). Cell pellets were flash frozen in liquid nitrogen and kept in a −80 °C freezer until use.

### Detergent purification for HDX–MS and cryo-EM samples

All purification steps were carried out at 4 °C or over ice, unless otherwise noted. The avian or human TRPM8 channel was purified with the same general protocol. Briefly, a cell pellet grown from 0.5 l Expi293F cell culture (typically 7–10 g in total) was resuspended into lysis buffer (50 mM HEPES-NaOH pH 7.40, 150 mM NaCl, 50 µg ml$^{-1}$ DNase I, 10 µg ml$^{-1}$ RNase, 0.2 mM AEBSF, 50 µg ml$^{-1}$ soy trypsin inhibitor, 10 µg ml$^{-1}$ leupeptin, 10 µg ml$^{-1}$ pepstatin, 1 mM benzamidine-HCl and aprotinin) for a total dilution of 4:1 lysis buffer:cell pellet. The resuspension was adjusted to 0.5% lauryl maltose neopentyl glycol (LMNG)/0.5% glycodiosgenin (GDN), and rotated gently for 1 h at 4 °C. After centrifugation (35,000*g*, 30 min, 4 °C), the lysate was applied to 1 ml anti-GFP-nanobody-conjugated Sepharose 4B resin (prepared in-house) for 2 h at 4 °C with gentle rotation. The TRPM8-bound resin was washed extensively with wash buffer (20 mM HEPES-NaOH pH 7.40, 150 mM NaCl, 0.05% GDN), then digested in the presence of 10 µg ml$^{-1}$ PreScission protease (prepared in-house) along with 1 mM dithiothreitol for 2 h. The eluate was concentrated to 0.2 ml and injected onto a Superose 6 Increase 10/300 GL column pre-equilibrated in SEC buffer (20 mM HEPES-NaOH pH 7.40, 150 mM NaCl, 0.005% GDN). For the *Pm* structures obtained at pH 9, cells were resuspended into lysis buffer containing 50 mM Tris-HCl pH 9.0, 150 mM NaCl, 5 mM $CaCl_2$, 50 µg ml$^{-1}$ DNase I, 10 µg ml$^{-1}$ RNase, 0.2 mM AEBSF, 50 µg ml$^{-1}$ soy trypsin inhibitor, 10 µg ml$^{-1}$ leupeptin, 10 µg ml$^{-1}$ pepstatin, 1 mM benzamidine-HCl and aprotinin. The resuspended cells were extracted using 0.5% LMNG/0.5% GDN and further purified with anti-GFP nanobody resin. For cryo-EM, the final sample buffer contained 20 mM TRIS pH 9.00, 150 mM NaCl, 5 mM $CaCl_2$ and 0.0025% GDN.

### Cell vesicle purification for cryo-EM

A cell pellet grown from 1.0 l Expi293F cell culture (typically 20–25 g in total) was resuspended into lysis buffer (50 mM HEPES-NaOH pH 7.40, 300 mM KCl) supplemented with 2 mM $CaCl_2$, 50 µg ml$^{-1}$ DNase I, 10 µg ml$^{-1}$ RNase, 0.2 mM AEBSF, 50 µg ml$^{-1}$ soy trypsin inhibitor, 10 µg ml$^{-1}$ leupeptin, 10 µg ml$^{-1}$ pepstatin, 1 mM benzamidine-HCl, and aprotinin. The mixture was homogenized roughly using a Dounce homogenizer over ice, and the resuspended cells were

adjusted to a final volume of 200 ml. The cells were transferred to a metal beaker over ice and lysed using a probe-tip sonicator (2 min total, 5 s on/15 s off, 60% amplitude). The lysate was clarified by centrifugation (12,000*g*, 15 min, 4 °C) and the supernatant was filtered immediately through a 0.8-µm mixed cellulose acetate filter. The filtrate was then passed through a 10-ml bed of ion exchange resin (Q-Sepharose FastFlow, Cytiva) packed in a gravity column and pre-equilibrated in lysis buffer. The flowthrough was collected and adjusted to 0.5 mM fluorinated fos-choline 8 (Anatrace), which is below its critical micelle concentration of around 3 mM, for a final volume of 240 ml. Vesicles were batch bound to 5 ml anti-GFP nanobody Sepharose 4B resin (prepared in-house) for 2 h at 4 °C with constant rotation. After binding, the resin was collected by centrifugation (500*g*, 1 min) and transferred to a gravity column. The affinity resin was washed extensively, and the resuspended beads were adjusted to 10 µg ml$^{-1}$ PreScission protease (prepared in-house) along with 1 mM dithiothreitol. Bound vesicles were eluted by proteolysis for 2 h, and the eluate was collected and concentrated using an Amicon ultra-centrifugal filter (100 kDa, regenerated cellulose) to a final volume of 0.5 ml. After brief centrifugation (10,000*g*, 2 min, 4 °C), the sample was injected onto a Superose 6 Increase 10/300 column (Cytiva) pre-equilibrated with the same elution buffer. The void volume peak (typically 8.5–9 ml) was pooled and concentrated to 30 µl using a 0.5-ml Amicon ultra-centrifugal filter (100 kDa, regenerated cellulose) and kept on ice. Concentrated vesicle samples were used immediately for or warmed to 37 °C for 5–10 min before cryo-EM sample preparation.

### Cryo-EM grid preparation, screening and data collection

Cryo-EM grids for vesicle samples were prepared using either Quantifoil R1.2/1.3 400 mesh holey carbon grids, Quantifoil R1.2/1.3 300 mesh holy carbon grids coated with ultrathin continuous carbon. Detergent samples were prepared using Quantifoil Au R1.2/1.3 300 mesh holey carbon grids. Grids were glow discharged (15 mA, 30 s) and placed in a Vitrobot Mark IV (FEI Company) set at 4 °C with 100% humidity. For vesicle samples, 2.5 µl concentrated sample was applied directly to grids and, after 5 s of incubation, grids were blotted for 2–5 s and plunge-frozen immediately in liquid ethane cooled in a Dewar of liquid nitrogen. To obtain menthol-bound datasets, vesicles were kept at 4 °C for *Pm* TRPM8 or ambient temperature (21–24 °C) for the human channel, and a final concentration of 1 mM menthol was added from a stock of 200 mM menthol dissolved in 100% ethanol. After mixing the menthol-treated samples gently at respective temperatures, samples were applied to grids, for a total agonist treatment time of 2–15 min. For detergent samples, 2.5 µl of purified TRPM8 concentrated to 14–16 mg ml$^{-1}$ was applied directly to grids and, after 10 s of incubation, grids were blotted for 5 s before vitrifying. Grids were transferred and stored under liquid nitrogen until screening or data collection. Grids were screened with a Talos Arctica or Glacios 200 kV cryo-TEM (ThermoFisher Scientific) equipped with a K3 direct detector camera (GATAN), and screening datasets were obtained using SerialEM. Data were collected at the University of California San Francisco (UCSF) Cryo-EM Facility with a Titan Krios cryo-TEM equipped with a K3 camera and Bio Quantum post-column energy filter (counting mode pixel size of 0.8189 Å per pixel after 2× Fourier binning) or at the Janelia Cryo-EM Facility on a Krios equipped with a cold field-emission gun, Selectris X energy filter and Falcon 4i camera (physical pixel size of 0.94). For data collection at UCSF, the zero-loss energy selection slit was set to 10 eV. For vesicle datasets, the target defocus was set at −1.0 to −2.5 µm and, for detergent datasets, the target defocus was set at −0.5 to −2.0 µm.

### Cryo-EM data processing and refinement

Dose-weighted, motion-corrected micrographs were obtained using MotionCor2 (ref. 52) and Fourier cropped by a factor of two, or by pre-processing using cryoSPARC[53]. An initial model of *Pm* TRPM8 in a membrane was generated in cryoSPARC with a screening dataset obtained on

a Glacios cryo-TEM (2,420 micrographs) and refined to approximately 5 Å (440 pixel box size, 0.73 Å per pixel). This initial reconstruction was used to generate a reference volume using relion_image_handler (512 pixel box size, 0.8189 Å per pixel) that was subsequently used in template picking for further processing of datasets obtained on the Krios microscopes. For the vesicle datasets, particles were identified using a combination of Topaz particle picking[54] and template picking in cryoSPARC, followed by several rounds of heterogeneous refinement and two-dimensional classification to deplete non-TRPM8 or obvious junk particles. After generating a consensus refinement with cryoSPARC non-uniform refinement, further classification was carried out in either RELION5 (ref. 55) or cryoSPARC. Three-dimensional classification used either a mask capturing only the channel region, or a spherical mask to limit bilayer signal. Further two-dimensional classification of three-dimensional refinements without a circular mask revealed that a range of membrane curvatures are included in reconstructions, indicating that, in our datasets, three-dimensional classes result from a distribution, rather than specific, membrane curvatures (Supplementary Fig. 1c). For the GDN-purified TRPM8 datasets, an initial reference map was generated from each dataset ab initio from particles identified with blob picker in cryoSPARC from a small sub-set of the micrographs. The ab initio model was then subjected to non-uniform refinement without symmetry and used subsequently to generate templates to process the remaining dataset. After several rounds of heterogeneous refinement, a consensus refinement was generated in cryoSPARC for further three-dimensional classification with cryoSPARC. Three-dimensional class averages were subjected to non-uniform refinement using cryoSPARC. Local resolution estimation of refined maps was carried out in cryoSPARC and visualized in UCSF ChimeraX.

## Model building, pore radius and p$K_a$ calculations

For the unliganded *Pm* semi-swapped structure, an initial atomic model was generated de novo using ModelAngelo (RELION v.5)[56], which provided initial coordinates that defined the connectivity of the outer pore loop, S6 and TRP domain. This model was refined iteratively using Phenix real space refinement and manual refinement in COOT[57] to assess density fit. Initial atomic models of *Pm* or *Hs* TRPM8 were either generated de novo or sourced from previously published structures (Extended Data Table 1). Representative densities from various regions of the corresponding reconstruction fitted with atomic models are shown in Supplementary Figs. 9–12. All models were validated using MolProbity in Phenix[58]. Refinement statistics of cryo-EM maps and model statistics are provided in Extended Data Table 1. A video morphing between the semi- and fully swapped models shown as a space-filling models was generated by interpolating the two PDBs (PDBs 9P90 and 9P91) using the morph command in UCSF ChimeraX (Supplementary Video 1). The HOLE program[59] was used to determine all pore radii for modelled structures. For p$K_a$ calculations, models of the *Pm* semi-swapped closed (PDB 9P91) or fully swap closed/desensitized (PDB 9P90) structures obtained in vesicles were used as inputs for PROPKA3 (ref. 60). A summary of p$K_a$ values for ionizable amino acids found on the extracellular region is provided in Extended Data Fig. 6.

## Hydrogen–deuterium exchange

Supplementary Table 3 summarizes biochemical and statistical details for HDX in this study. HDX was initiated by diluting 2 μl of 10–25 μM GDN-solubilized TRPM8 stock in an $H_2O$ buffer into 28 μl of a matching deuterated buffer to reach 93% D-content. Labelling was conducted at pH measured in deuterated solvent (pD)$_{read}$ 7.0 and temperatures of 4 °C, 22 °C, 30 °C or 37 °C. Labelling times were adjusted to account for the temperature dependence of intrinsic rates ($k_{chem}$) based on the average $k_{chem}$ of the full sequence[61], with details presented in Supplementary Table 3. For HDX in the presence of menthol, two conditions were employed: (1) 1 mM menthol was added to HEK293 cells expressing TRPM8 and maintained throughout purification and exchange;

(2) TRPM8 was purified without menthol, and 1 mM menthol was added only to the $D_2O$ buffer. Under both conditions, HDX was performed at 22 °C. HDX time-points were collected in triplicate.

HDX was quenched at time-points ranging from 3 s to 3 days by adding 30 μl of ice-cold quench buffer (600 mM glycine, 8 M urea, 0.005% GDN, pH 2.5). Quenched samples were incubated on ice for 20 s, further diluted with 50 μl quench buffer without urea and injected immediately into a valve system maintained at 5 °C (Trajan LEAP). Non-deuterated controls and MS/MS runs for peptide assignment followed the same protocol, except that $D_2O$ buffers were replaced by $H_2O$ buffers, with immediate quenching and injection. HDX reactions were performed in random order. No peptide carryover was observed as assessed by injecting quench buffer containing 2 M urea and 0.005% GDN. In-exchange controls accounting for forward deuteration towards 41.5% D in the quenched reaction were performed by mixing $D_2O$ buffer and ice-cold quench buffer before protein addition. Maximally labelled controls ('All D') accounting for back-exchange were performed by incubating samples in $D_2O$ buffer for 24 h at 22 °C, followed by a 1-h heat shock at 48 °C.

## Liquid chromatography–mass spectrometry

Liquid chromatography–mass spectrometry analysis for the first replicate and selected time-points for the second replicate was performed on instruments at University of California Berkeley (Marqusee laboratory), whereas the remaining time-points of replicate two and all of replicate three were analysed at UCSF (Cheng laboratory) using the same columns but different liquid chromatography–mass spectrometry instrumentation. We note that, due to differences in the liquid chromatography dead volumes, the retention times for peptides analysed at UCSF were systematically around 12 s longer than runs analysed at University of California Berkeley and, therefore, different integration bounds were used when analysing the mass spectra for those two data sets. Nonetheless, the two data sets showed high agreement in deuteration levels, as demonstrated in the repeatability assessments in Supplementary Fig. 8 and Supplementary Table 3.

Upon injection, the protein was digested online using a NepII/pepsin protease column (Affipro, catalogue no. AP-PC-006) at 10 °C. The resulting peptides were desalted by flowing across a hand-packed trap column (Thermo Scientific POROS R2 resin, catalogue no. 1112906, with IDEX C-128 1-mm internal diameter × 2-cm cartridge) at 5 °C. Digestion and desalting were completed in 2.5 min at 100 μl min$^{-1}$ of 0.1% formic acid. Peptides were then separated on a C18 analytical column (Waters ACQUITY UPLC BEH C18, 130 Å, 1.7 μm, 1 mm × 50 mm, catalogue no. 186002344) with a 17-min linear gradient of 5–45% (vol/vol) acetonitrile (0.1% formic acid) delivered by a Thermo UltiMate-3000 pump (University of California Berkeley) or Vanquish Neo pump (UCSF). Eluted peptides were analysed on a Thermo Q Exactive (UC-Berkeley) or Q Exactive Plus (UCSF) mass spectrometer in positive mode (full mass spectrometry: resolution 140,000, automatic gain control target 3 × 106, maximum injection time 200 ms, scan range 300–2,000 *m/z*; dd-MS$^2$: resolution 17,500, automatic gain control target 2 × 105, maximum injection time 100 ms, loop count 10, isolation window 2.0 *m/z*, normalized collision energy 28, dynamic exclusion 15 s).

## HDX–MS data analysis and EX2 characterization

Peptides were identified using SearchGUI (v.4.0.25) and Byonic (Protein Metrics) against a search library containing TRPM8, ten additional proteins previously exposed to liquid chromatography–mass spectrometry and their reserved sequences as decoys. Search parameters included unspecific digestion, precursor and fragment tolerance of 10 ppm, precursor charge 1–8 and peptide length 5–50. HDX data were processed in HDExaminer v.3.4 (Trajan), with bimodal fitting accepted when the score exceeded the unimodal score by 0.1 or above.

The observed exchange occurred through EX2 kinetics, reporting on the equilibrium instead of opening rates of amide protons. The identification of EX2 behaviour is supported by (1) the continuous

shifts in the isotopic envelopes towards higher $m/z$ with exchange time (Supplementary Fig. 2); (2) the difference in $k_{obs}$ for TRPM8 labelled at various pD conditions can be attributed solely to the effect of pD on $k_{chem}$ (Supplementary Fig. 3). For peptides exhibiting bimodal mass envelopes, each envelope increases in mass over labelling time with exchange rates slower than $k_{chem}$, and the $k_{obs}$ for both envelopes show pD-dependence consistent with EX2 kinetics, indicating that the two mass envelopes represent two kinetically distinct populations that each exchange through EX2 kinetics (rather than EX1 or carryover)[62] (Supplementary Figs. 3 and 4). To cross-validate those bimodal fits, mass spectra were exported to HX-Express3 (ref. 63) for double-binomial fitting, with an $F$-test to assess the increase in fit quality over a single-binomial model. Bimodality was accepted when $P < 0.01$. Fitting was repeated with 20 iterations using a resampling approach in which random noise (±30% of each isotope peak intensity) was introduced in each iteration (Supplementary Figs. 5 and 6).

Each isotopic distribution was inspected manually for fit quality. Deuteration levels were adjusted for 93% D-content and back-exchange. Except for Fig. 2, for peptides displaying clear bimodality, deuteration levels represent centroids of unimodal fit to represent the overall HDX for two populations.

**Quantifying thermodynamic parameters from HDX–MS**

H–D exchange follows the Linderstrøm–Lang model described by the following scheme

$$N-H{\cdots}O=C \underset{k_{close}}{\overset{k_{open}}{\rightleftarrows}} N-H \overset{k_{chem}}{\longrightarrow} N-D$$
$$\text{closed} \qquad \text{open} \qquad \text{exchanged}$$

where $k_{open}$ and $k_{close}$ are the opening and closing rates of hydrogen bonds for individual amides, and $k_{chem}$ is the chemical exchange rate of the amide in an unstructured polypeptide. At steady-state, the observed H–D exchange rate can then be described as

$$k_{obs} = \frac{k_{open}k_{chem}}{k_{open}+k_{close}+k_{chem}}$$

As peptides in our dataset all exhibited EX2 kinetics (see 'HDX–MS data analysis and EX2 characterization'), where $k_{chem} \ll k_{close}$,

$$k_{obs} = \frac{k_{open}k_{chem}}{k_{open}+k_{close}} = \frac{k_{chem}}{K_{eq}+1}$$

where $K_{eq} = k_{close}/k_{open}$ and describes the equilibrium constant of H-bond closing and opening (folding). Folding free energy ($\Delta G$) is then described as

$$\Delta G = -RT\ln K_{eq} = -RT\ln\left(\frac{k_{chem}}{k_{obs}}-1\right) \approx -RT\ln\frac{k_{chem}}{k_{obs}}$$

As $k_{chem}$ can be calculated reliably from the sequence of the peptide[61], measurement of the H–D exchange rate ($k_{obs}$) enables direct quantification of $\Delta G$ at peptide level. In this study, a stretched-exponential method was used to estimate $\Delta G$ from HDX as described by Hamuro[37]. Briefly, deuteration levels for each peptide were fit with the following equation:

$$\%D(t) = 1 - \exp[(-kt)^b]$$

where $k$ is the exchange rate, and $b$ is the stretch factor that describes variations of $k$ for residues within the peptide. In those calculations, peptides with exchange rates beyond the measurement range of this study were assigned fixed values: peptides fully exchanged at the first time-point (30 s) were assigned $k_{chem} = 10\ s^{-1}$, whereas peptides with no exchange at the final time-point (30 h) were assigned $k_{obs} = 10^{-5}\ s^{-1}$.

Changes in folding free energy ($\Delta\Delta G$) under two conditions (with and without menthol) were then calculated based on the following equation:

$$\Delta\Delta G = -\ln(10)RT\left[-\frac{\gamma}{\ln(10)}\left(\frac{1}{B_1}-\frac{1}{B_2}\right)-\log_{10}\frac{k_1}{k_2}\right]$$

where $\gamma$ is the Euler–Mascheroni constant (approximately 0.577215665). For Fig. 4a and Extended Data Fig. 5a, an average difference of deuteration levels was used to estimate the $\Delta\Delta G$ at temperatures relative to 22 °C, as described by Hamuro[37], as some regions exhibited incomplete exchange within our labelling times, which resulted in low confidence in fitting results with the stretched-exponential method. In this strategy, $\Delta\Delta G$ was calculated based on the following equation:

$$\Delta\Delta G = -\ln(10)RT\frac{W}{p}\sum_{t=1}^{p}(\%D_{1,t}-\%D_{2,t})$$

where $p$ is the number of time-points and $t$ is a specific time-point. For each temperature window, van't Hoff analysis was employed to determine $\Delta H°$, entropy contribution ($T\Delta S°$) and Gibbs free energy ($\Delta G°$) extrapolated to 25 °C, assuming no change in $C_p$, based on the following equation:

$$\ln K_{eq} = -\frac{\Delta H°}{R}\frac{1}{T}+\frac{\Delta S°}{R}$$

In both quantification strategies, residue-level $\Delta\Delta G$ and $\Delta H°$ values were determined from a weighted average of peptide-level parameters, excluding the first two N-terminal residues due to rapid back-exchange. We note that not all peptides identified exhibited complete exchange within our labelling times and, therefore, the thermodynamic parameters quantified here represent the best approximation within the scope of the study. Nonetheless, our approach effectively captures relative energetic changes associated with menthol-binding and temperature changes that are consistent with our structural and functional studies.

**Fura-2-AM calcium imaging and data analysis**

Calcium imaging experiments were carried using Fura-2-AM as a ratiometric fluorescent indicator (as described previously[64]) with temperature monitoring throughout imaging. Briefly, adherent HEK293T cells grown at 37 °C with 5% ambient $CO_2$ were seeded in a 12-well plate (1 ml volume) to a final density of 30–40% confluence and transfected with 0.5 mg ml$^{-1}$ plasmid using Lipofectamine 3000 (ThermoFisher Scientific) according to the manufacturer. Cells were allowed to transfect for 12–14 h. Borosilicate cover slips (12 mm, Bellco Glass) were incubated with a 1:100 solution of Matrigel matrix (Corning) prepared in serum-free DMEM (Gibco) for 1 h, then washed once with DMEM and left in a 24-well plate with DMEM supplemented with 10% bovine calf serum. Transfected cells were dissociated gently and replated directly onto coverslips and allowed to attach for 2–3 h before imaging experiments were carried out. Attached cells were washed once with Ringer's solution (10 mM HEPES-Tris pH 7.40, 140 mM NaCl, 5 mM KCl, 2 mM $CaCl_2$, 2 mM $MgCl_2$, 10 mM glucose; 290–310 mmol kg$^{-1}$), and then incubated for 0.5 h at room temperature in the dark in Ringer's solution supplemented with 10 µg ml$^{-1}$ Fura-2-AM (ThermoFisher Scientific) and 0.02% Pluronic F127. After incubation, cells were washed once with Ringer's solution and allowed to incubate further in Ringer's solution without Fura-2-AM or Pluronic F127 for 0.5 h. Dye-loaded cells were then imaged using an inverted microscope with 340 and 380 nm excitation (Sutter, Lambda LS Illuminator). The temperature for each experiment was held at 32–37 °C by perfusing Ringer's solution through a Peltier device controlled with a heating module (Warner TC-324B) calibrated to hold the desired temperature continuously. Cells were held at this temperature range for 5–10 min before each recording. Bath exchange

into low temperatures was achieved by perfusing Ringer's solution cooled with ice water into the recording chamber. Temperature was monitored and synchronized using a thermistor reading input to the heating module that was digitized through an Axon Digidata 1550B module and monitored with pCLAMP10. Recordings were obtained as stacked videos of individual 340/380 frames with a 1-s interval in Micro-Manager[65]. Videos of ratio images were calculated with an ImageJ macro script according to known procedures[66]. Peak amplitudes of calcium signals corresponding to either cold or menthol, were obtained and normalized to maximum calcium signal obtained using 10 μM ionomycin before further statistical analyses.

## Sequence alignment of TRPM8 orthologues

Sequences of avian or mammalian TRPM8 were obtained from OrthoDB v.12.2 (orthodb.org) by curating the phylogenetic clades of Sauropsida (which includes avian species), and Mammalia, respectively. Multiple sequence alignments were generated for Sauropsida (450 sequences) and Mammalia (390 sequences) using Clustal Omega (https://www.ebi. ac.uk/jdispatcher/msa/clustalo). The resulting outputs were used to plot individual sequence logo representations of the avian or mammalian TRPM8 orthologues near Y905 (for *Pm* TRPM8) or V915 (for *Hs* TRPM8) using R (ggseqlogo)[67].

## Reporting summary

Further information on research design is available in the Nature Portfolio Reporting Summary linked to this article.

## Data availability

All plasmids used in this study have been deposited with AddGene for distribution. Cryo-EM maps and atomic models have been deposited in the EMDB and PDB as follows: *Pm* TRPM8 semi-swapped, closed, unliganded, CaCl₂-free, 4 °C adapted in cell vesicles: EMD-71395, PDB 9P91; *Pm* TRPM8 fully swapped, closed/desensitized, unliganded, CaCl₂-free, 4 °C adapted in cell vesicles: EMD-71394, PDB 9P90; *Pm* TRPM8 fully swapped, closed, unliganded, CaCl₂-free, 4 °C adapted in cell vesicles: EMD-71352, PDB 9P7S; *Pm* TRPM8 undetermined example class 1, unliganded, CaCl₂-free, 4 °C adapted in cell vesicles: EMD-74126; *Pm* TRPM8 undetermined example class 2, unliganded, CaCl₂-free, 4 °C adapted in cell vesicles: EMD-74127; *Pm TRPM8* fully swapped, closed, unliganded, 5 mM CaCl₂, 4 °C adapted in cell vesicles: EMD-74036, PDB 9ZCN; *Pm* TRPM8 semi-swapped, closed, unliganded, 5 mM CaCl₂, 4 °C adapted in cell vesicles: EMD-74037, PDB 9ZCO; *Pm TRPM8* fully swapped, closed, menthol, CaCl₂-free in cell vesicles: EMD-74038, PDB 9ZCP; *Pm* TRPM8 semi-swapped, closed, menthol, CaCl₂-free in cell vesicles: EMD-74039, PDB 9ZCQ; *Pm TRPM8* semi-swapped, open, menthol, CaCl₂-free in cell vesicles: EMD-74123, PDB 9ZEZ; *Pm* TRPM8 fully swapped, closed, putative desensitized menthol, CaCl₂-free in cell vesicles: EMD-74125; *Pm* TRPM8 semi-swapped, high pH, open, CaCl₂, 4 °C in GDN: EMD-71444, PDB 9PAR; *Pm* TRPM8 Chimeric Human(V905-L938) semi-swapped, pH 7.4, open, CaCl₂, 4 °C in GDN: EMD-74040, PDB 9ZCR; Human TRPM8 fully swapped, closed, unliganded, 4 °C in GDN: EMD-71391, PDB 9P8Y; Human TRPM8 fully swapped, closed, desensitized, unliganded, calcium-free 4 °C in cell vesicles: EMD-74042, PDB 9ZCV; Human TRPM8 fully swapped, open, unliganded, calcium-free 4 °C in cell vesicles: EMD-74124, PDB 9ZF0; Human TRPM8 semi-swapped, unliganded, calcium-free 4 °C in cell vesicles EMD-74129; Human TRPM8, fully swapped, open, menthol, CaCl₂-free, 4 °C in cell vesicles: EMD-71454, PDB 9PB5; Human TRPM8 V915Y, fully swapped, closed, CaCl₂, 4 °C in GDN: EMD-74041, PDB 9ZCU; Human TRPM8 V915Y, semi-swapped, CaCl₂, 4 °C in GDN: EMD-74128; Raw cryo-EM data (micrographs and particle stacks) for cold-adapted *Pm* TRPM8 without Ca²⁺ have been deposited to EMPIAR database (13269). Mass spectrometry proteomics data have been deposited to the ProteomeXchange Consortium through the PRIDE partner repository[68] with the dataset identifier PXD064468.

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

**Acknowledgements** We thank S. Marqusee (University of California Berkeley) for providing access to HDX–MS instruments and Byonic software during the early stage of this study, D. Bulkley, G. Gilbert, L. Wang (UCSF EM Core), R. Yan and N. Spellmon (Janelia Cryo-EM Facility) for their assistance with cryo-EM data acquisition and W. Choi for advice with data interpretation and model building. This work was supported by grants from the National Institutes of Health (NIH) (R35NS105038 to D.J. and R35GM140847 to Y.C.). Instruments at the UCSF Cryo-EM facility are supported partially by grants from the NIH (S10OD020054, S10OD021741 and S10OD026881) and Howard Hughes Medical Institute. Y.C. is an Investigator of the Howard Hughes Medical Institute.

**Author contributions** K.Y.C. carried out biochemical, structural and live-cell imaging components of the study. X.L. carried out HDX–MS components of the project. All authors participated in data analysis and evaluation and manuscript preparation. D.J. and Y.C. provided advice and guidance throughout.

**Competing interests** Y.C. serves on the scientific advisory boards for ShuiMu BioSciences and Pamplona Therapeutic Co. The other authors declare no competing interests.

**Additional information**
**Correspondence and requests for materials** should be addressed to Yifan Cheng or David Julius.

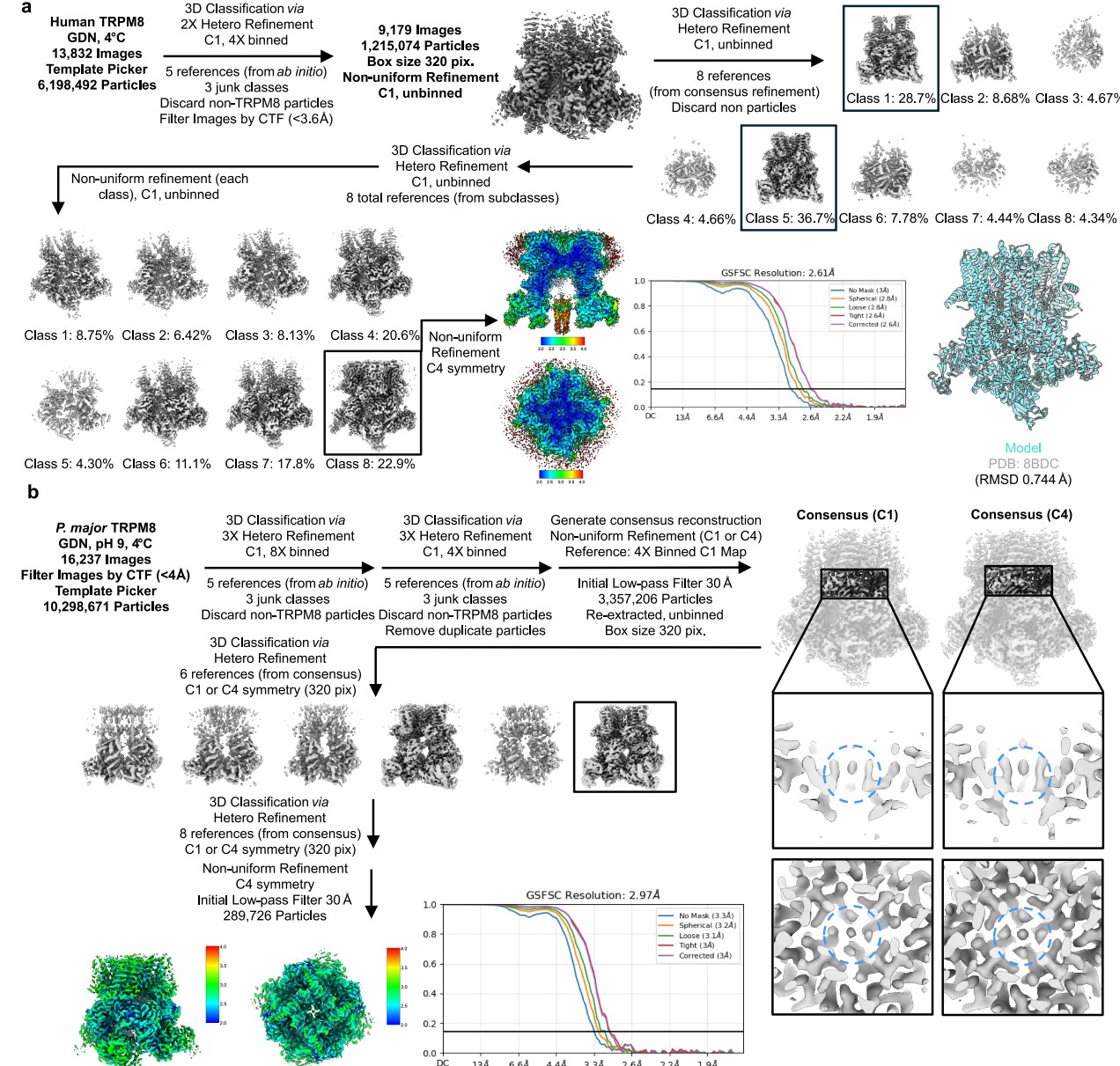

**Extended Data Fig. 1 | Data processing of TRPM8 structures determined in detergent. a**, Overview of data processing for a GDN-purified human TRPM8 structure determined at 4 °C. The consensus refinement is characterized by a well resolved cytosolic domain and poorly resolved transmembrane region. Subsequent classification led to 3D classes with similarly poorly resolved transmembrane regions (bottom right: classes 1–7), and one class with a fully resolved transmembrane domain (bottom right: class 8). Refinement of class 8 with C4 symmetry yielded a map at ~2.6 Å resolution. An overlay of the fitted

model and published hsTRPM8 model (PDB: 8BDC) is shown. This structure resembles the published closed, fully-swapped *pm*TRPM8 structure (PDB: 6O6A; rmsd 1.781 Å), or the closed, fully-swapped *hs*TRPM8 structure (PDB: 8BDC; rmsd 0.744 Å). **b**, Overview of data processing for a GDN-purified *P. major* TRPM8 structure determined in high pH at 4 °C. Consensus refinement with either C1 or C4 symmetry revealed a lower gate configuration that involves F969 and contains a cation-like density. The displayed local resolution maps, GSFSC plots, and orientation plots were generated using ChimeraX and cryoSPARC.

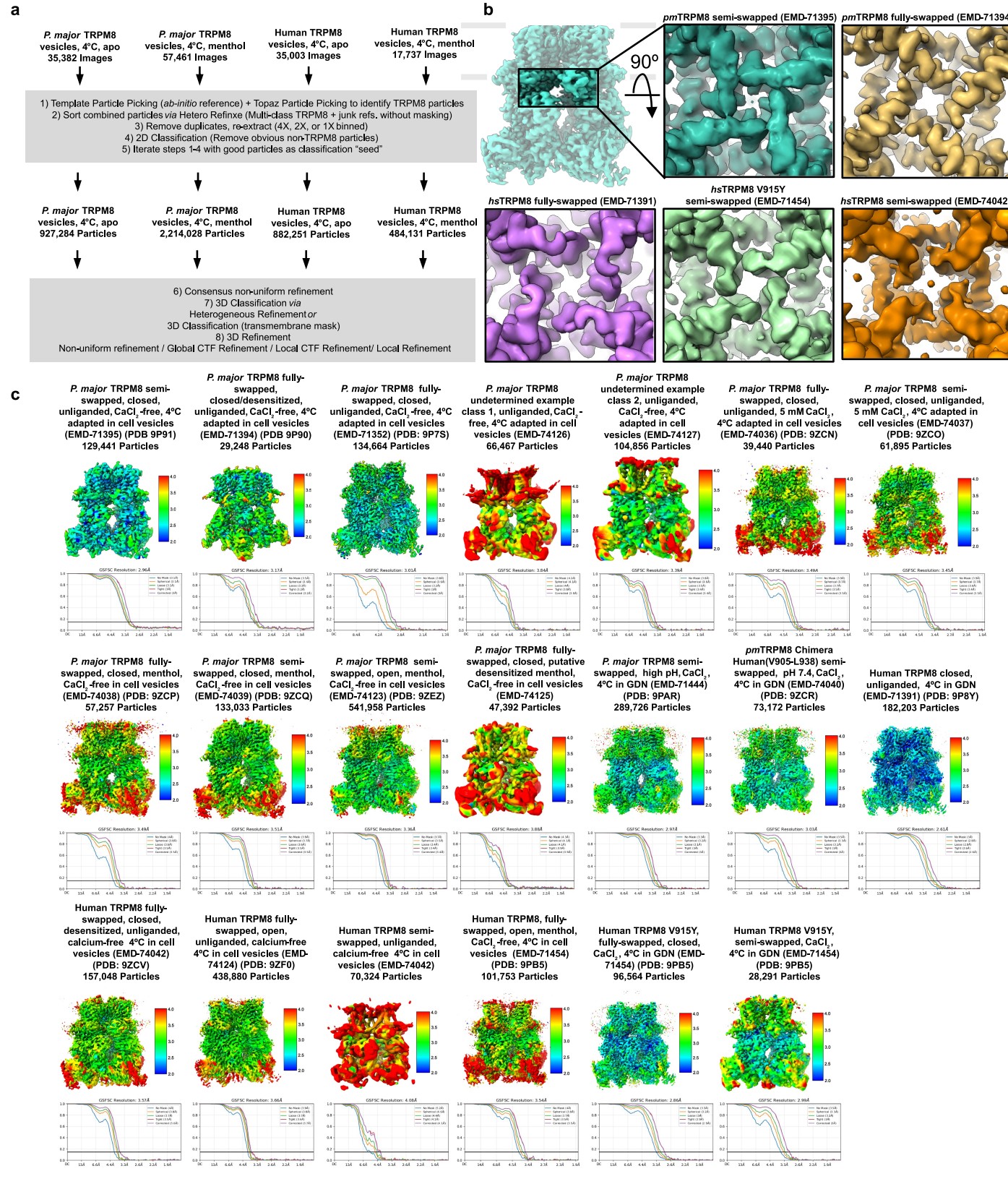

**Extended Data Fig. 2 | Semi- and fully swapped architectures observed in both avian and human TRPM8. a**, Overview of data processing pipeline for TRPM8 structures determined in cell vesicles. **b**, Bottom view of the fully- and semi-swapped channel seen in avian or human TRPM8 reconstructions. "Avian-ized" human channel (V915Y) in GDN shows increased resolution of the semi-swapped configuration, suggesting that the V915Y mutation stabilizes

the semi-swapped configuration of the human channel. The semi-swapped configuration is observed unambiguously in the hsTRPM8 wild type vesicle dataset at 4 °C. **c**, Local resolution maps and GSFSC curves of TRPM8 structures determined in this study. Local resolution maps and GSFSC plots were calculated in cryoSPARC.

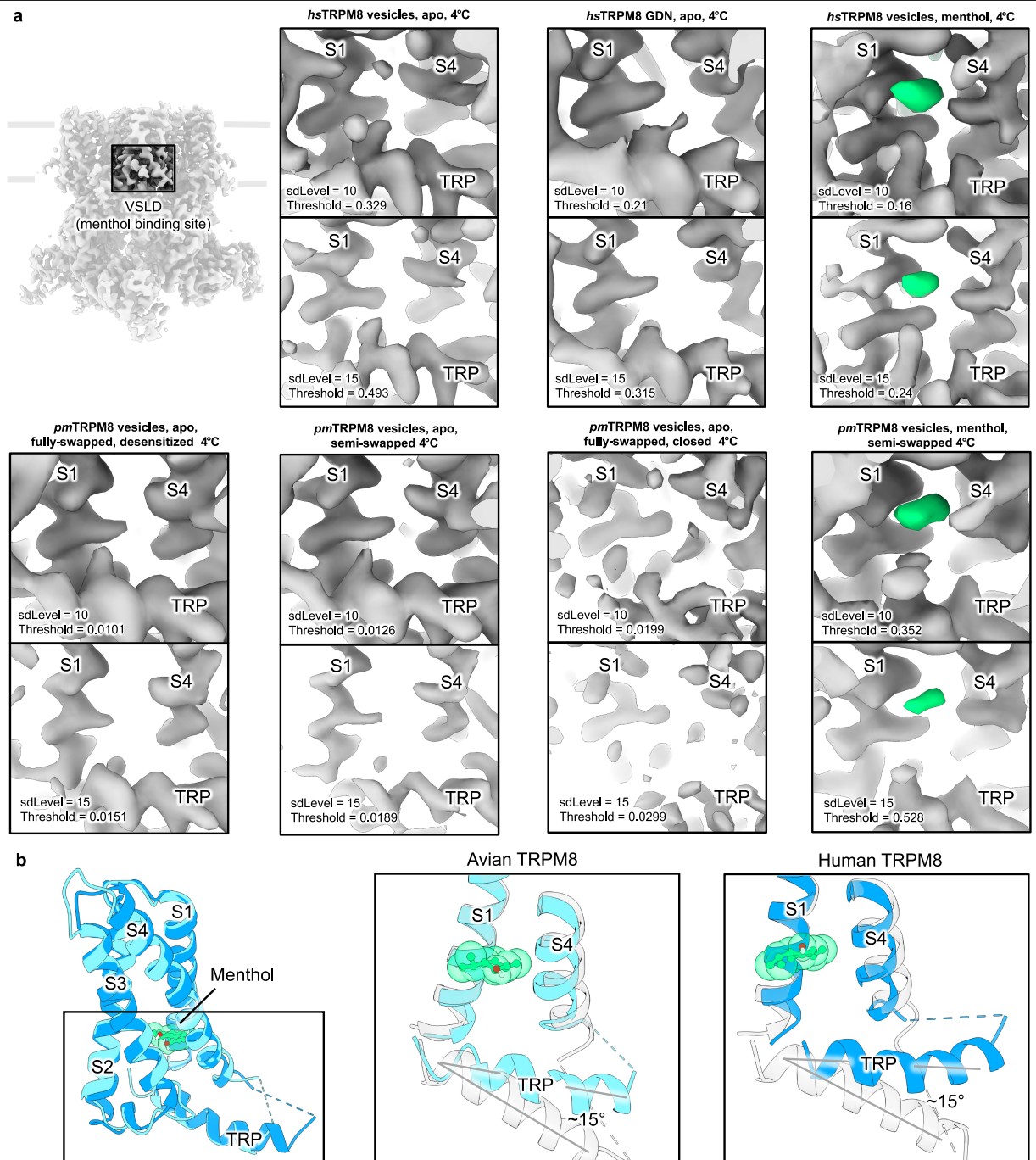

**Extended Data Fig. 3 | Menthol binding site. a**, Representative structures determined in the presence or absence of menthol for the avian or human TRPM8, and close-up view of the VSLD at two map thresholds. Standard deviation levels were set using ChimeraX, and exact map thresholds are displayed. **b**, Overview and overlay of the menthol-bound VSLD of avian (dark-blue) or human (light-blue) TRPM8. RMSD was 0.675 Å for 110 pruned atom pairs, or 2.110 Å for all 132 pairs. **c**, Relative repositioning of the TRP helix observed between the the menthol bound (blue) and unliganded, closed structures (ghosted). In the presence of menthol, the TRP helix repositions by a ~15° upwards tilt towards the VSLD.

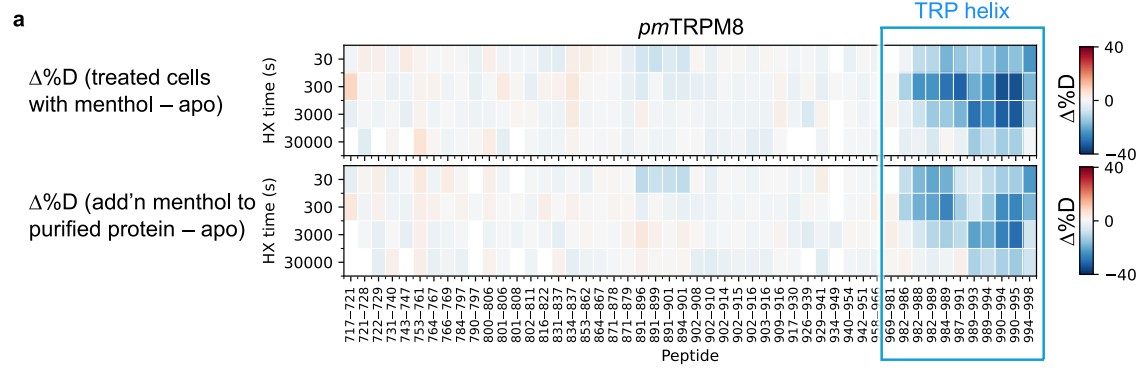

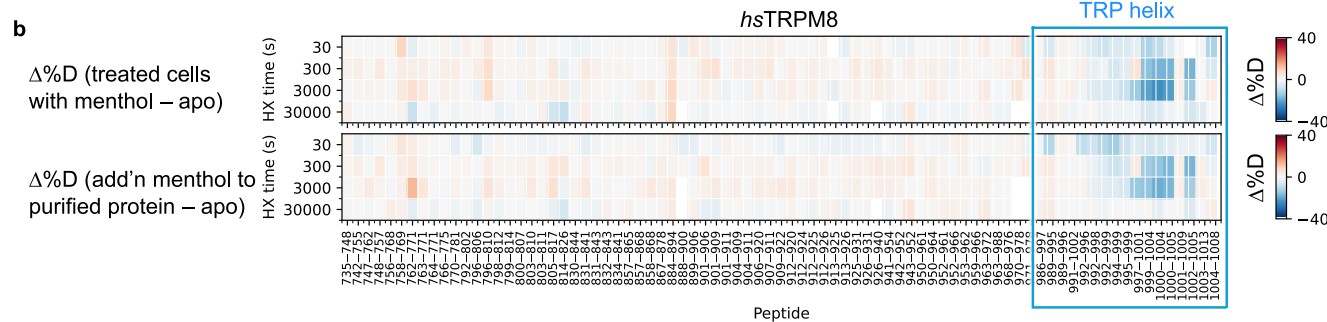

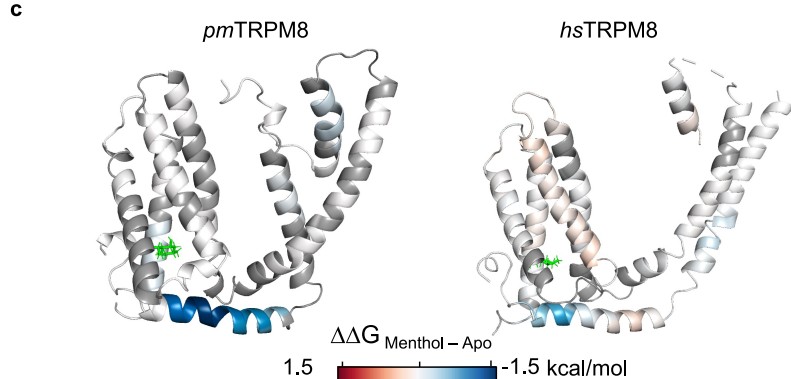

**Extended Data Fig. 4 | Menthol binding stabilizes the TRP helix. a,b,** Heatmaps showing changes in deuteration levels at each labelling time for all TMD peptides of avian (**a**) or human (**b**) TRPM8 with or without menthol measured in one of the following two conditions: (top) 1 mM menthol was added to whole cells prior to protein extraction and maintained throughout purification and exchange; (bottom) 1 mM menthol was added to purified TRPM8 protein only during exchange. **c,** Changes in folding free energy (ΔΔG) for TRPM8 upon menthol binding at 22 °C mapped onto the transmembrane domain.

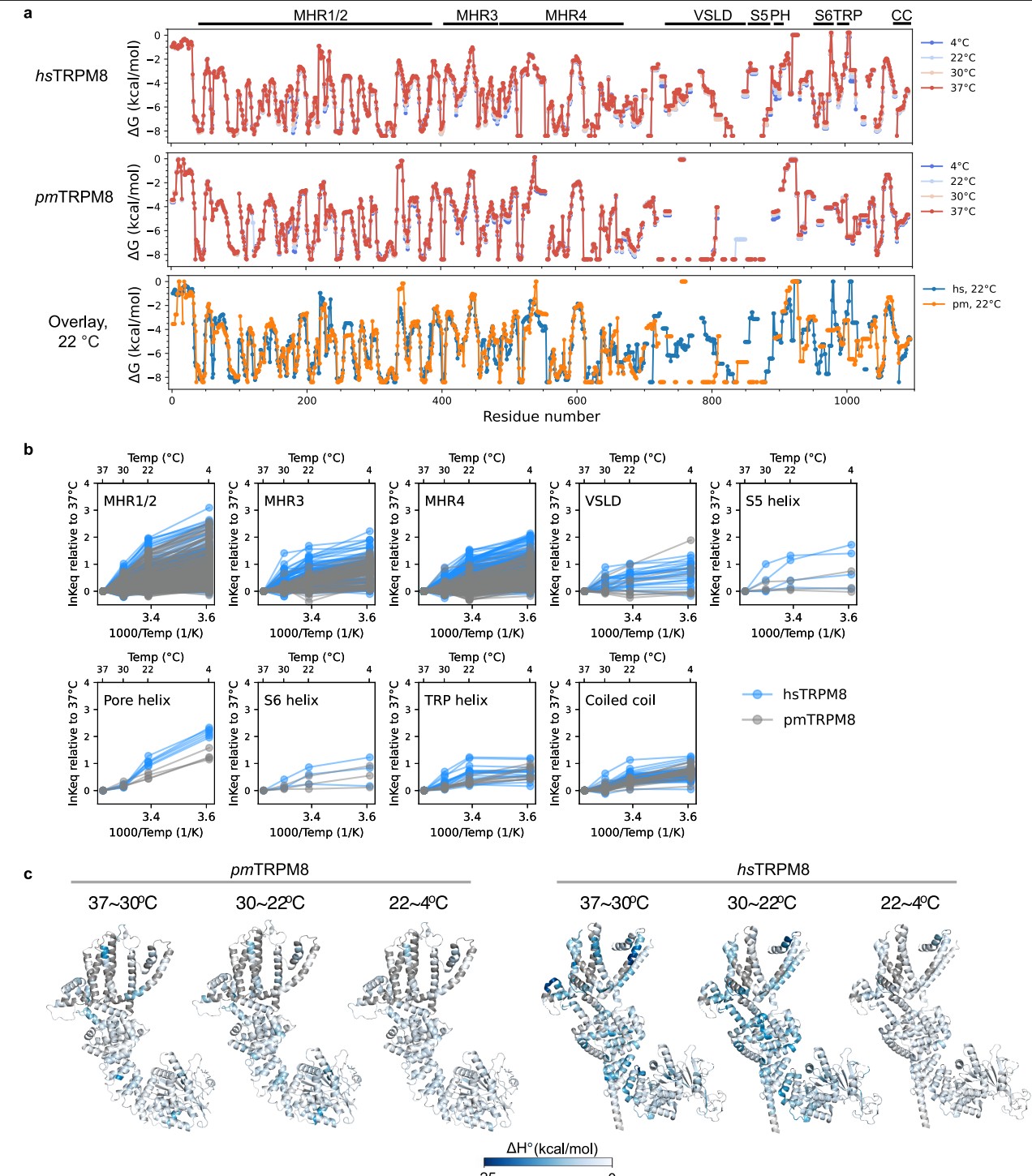

**Extended Data Fig. 5 | Comparison of the energetic profiles of human and avian TRPM8 at different temperatures. a**, Residue-level folding free energy (ΔG) calculated from HDX rates. Residue numbers correspond to the avian sequence. **b**, Van't Hoff analysis at peptide level. Individual curves correspond to peptides from their respective structural domains. The lnKeq values are shown relative to 37 °C to aid visual comparison between the two orthologues. **c**, Standard folding enthalpy (ΔH°) for human and avian TRPM8 at the indicated temperature ranges mapped onto monomer structure.

**a**

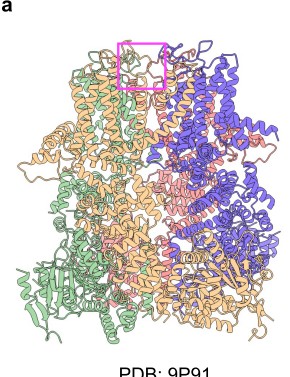

PDB: 9P91

**b**

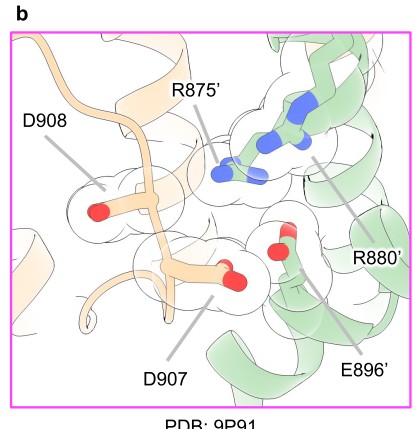

PDB: 9P91

**c**

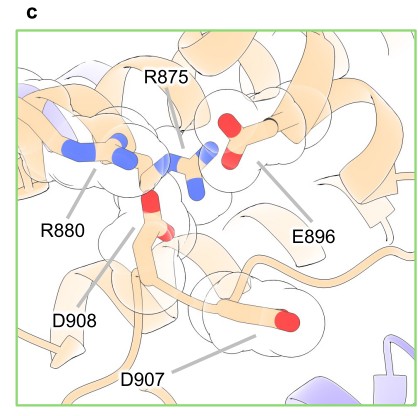

PDB: 9P90

**d**

| Semi-swap closed (PDB: 9P91) | | | | Full-swap closed, desensitized (PDB: 9P90) | | | |
|---|---|---|---|---|---|---|---|
| Residue | Position | pKa | model-pKa | Residue | Position | pKa | model-pKa |
| ASP | 856 | 4.68 | 3.8 | ASP | 856 | 4.84 | 3.8 |
| **ASP** | **907** | **8.16** | **3.8** | **ASP** | **907** | **5.26** | **3.8** |
| ASP | 908 | 5.43 | 3.8 | ASP | 908 | 3.92 | 3.8 |
| ASP | 910 | 4.51 | 3.8 | ASP | 910 | 3.59 | 3.8 |
| ASP | 917 | 3.88 | 3.8 | ASP | 917 | 3.86 | 3.8 |
| ASP | 934 | 2.73 | 3.8 | ASP | 934 | 3.17 | 3.8 |
| GLU | 883 | 5.3 | 4.5 | GLU | 883 | 4.69 | 4.5 |
| GLU | 887 | 4.35 | 4.5 | GLU | 887 | 4.11 | 4.5 |
| GLU | 896 | 4.29 | 4.5 | GLU | 896 | 4.15 | 4.5 |
| GLU | 925 | 4.58 | 4.5 | GLU | 925 | 3.42 | 4.5 |
| GLU | 932 | 5.11 | 4.5 | GLU | 932 | 4.96 | 4.5 |
| HIS | 884 | 6.76 | 6.5 | HIS | 884 | 7.1 | 6.5 |
| LYS | 881 | 9.65 | 10.5 | LYS | 881 | 9.86 | 10.5 |
| LYS | 927 | 10.36 | 10.5 | LYS | 927 | 10.4 | 10.5 |
| ARG | 875 | 13.97 | 12.5 | ARG | 875 | 14.33 | 12.5 |
| ARG | 880 | 10.98 | 12.5 | ARG | 880 | 12.07 | 12.5 |
| ARG | 885 | 12.35 | 12.5 | ARG | 885 | 12.55 | 12.5 |
| ARG | 891 | 12.48 | 12.5 | ARG | 891 | 12.82 | 12.5 |
| ARG | 940 | 13.32 | 12.5 | ARG | 940 | 12.56 | 12.5 |

**Extended Data Fig. 6 | pK$_a$ calculations of ionizable residues for the *P. major* semi-swapped closed or fully-swapped desensitized channel observed in cell vesicles. a**, Overview of the *P. major* semi-swapped TRPM8 structure (PDB: 9P91), and structural models of ionizable residues located near D907 for the (**b**) semi-swapped closed (PDB: 9P91) or (**c**) fully-swapped desensitized (PDB: 9P90) structures. These models show that D907 is in proximity to ionizable residues that comprise the extracellular interprotomer interface formed in the semi-swapped structure (see Fig. 1e), whereas D907 in the fully-swapped, desensitized model is not. **d**, pK$_a$ calculations for the respective PDB models for ionizable residues found on the extracellular side of the channel determined using PROPKA3 at pH 7.4.

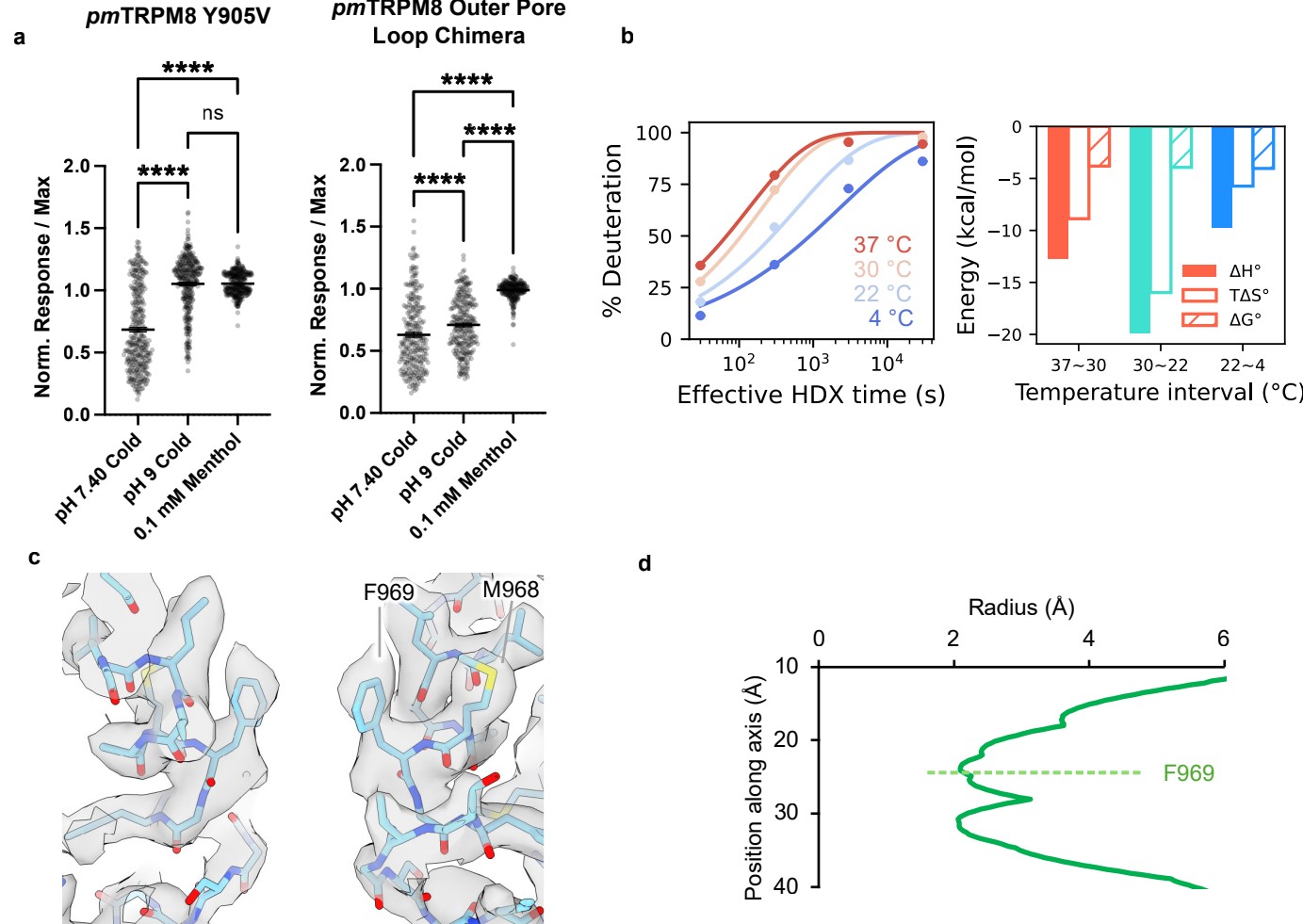

**Extended Data Fig. 7 | Temperature response of "humanized" avian TRPM8 by calcium imaging, HDX-MS, and cryo-EM. a**, Representative calcium imaging responses obtained from HEK293T cells expressing *pm*TRPM8 Y905V and a *pm*TRPM8 chimera replacing the entire outer pore loop sequence with the respective *hs*TRPM8 sequence. Responses to cold or menthol (100 μM) were normalized to maximum calcium signal following application of 10 μM ionomycin. Each dot represents a single cell; n = 336 for pm Y905V; 259 for pm outer pore loop chimera. Multi-measure one-way ANOVA with Tukey's post-hoc analysis; ***P < 0.001, ****P < 0.0001. **b**, Kinetics of deuterium uptake and thermodynamic parameters for peptide 891–901 in pore helix for "humanized" *pm*TRPM8 chimera at different temperatures. The solid lines represent a stretched-exponential fit of the uptake data. **c**, Lower gate density and model of pmTRPM8 human outer loop chimera structure resolved in cold at physiologic pH reveals a similar gating arrangement as the wild-type *pm*TRPM8 channel structure in cold at high pH, and **d**, a similar pore profile calculated using HOLE.

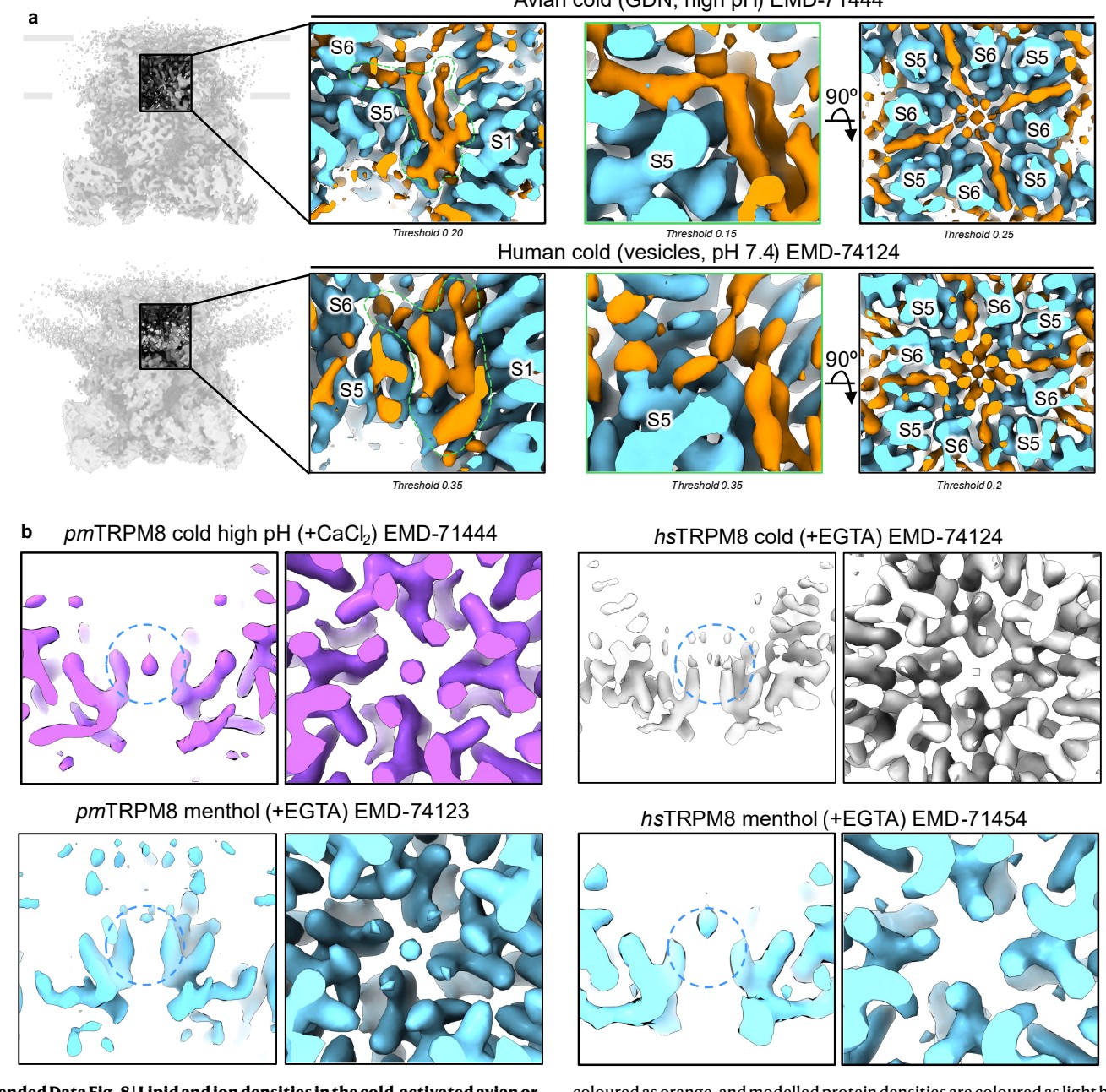

**Extended Data Fig. 8 | Lipid and ion densities in the cold-activated avian or human TRPM8 structures. a**, Details of lipid density observed at the TRPM8 PIP₂ binding site for the avian channel determined in cold at high pH or the human channel determined in cold in membranes at pH 7.4. Lipid densities are coloured as orange, and modelled protein densities are coloured as light blue. **b**, Cross-sectional top or side views of the central transmembrane cavity near the lower gate in the presence or absence of calcium.

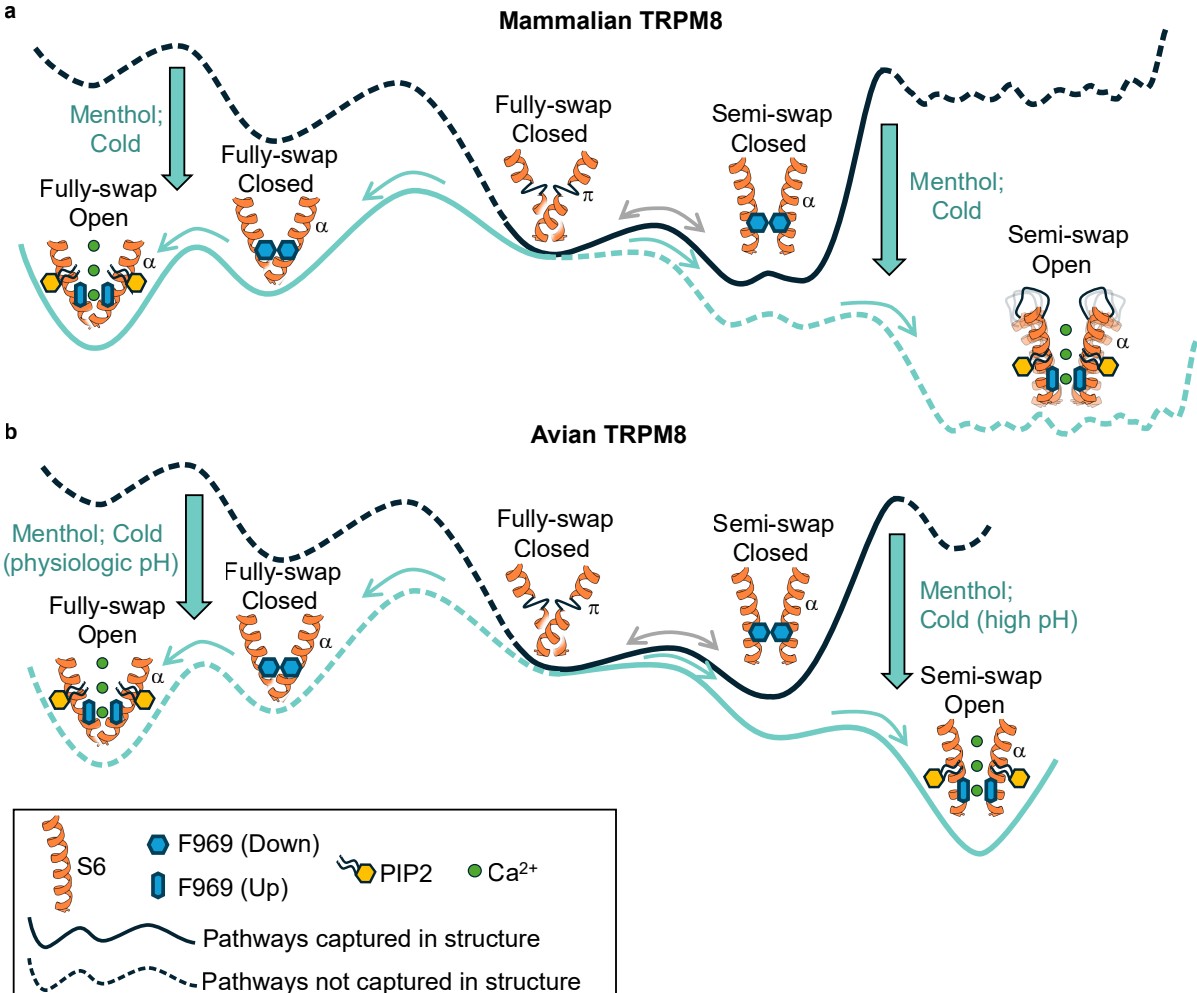

**Extended Data Fig. 9 | Model of TRPM8 activation by cold and menthol.**
Free energy of channel activation. **a**, In mammalian TRPM8, the closed channel exists in an equilibrium between fully- (S6 adopts a p helix) and semi-swapped (S6 adopts an a helix) configurations, with the latter occupying a broader energy well compared to the former. In each case, an a-helical S6 presents a phenylalanine residue at the lower gate, which is stabilized in an upward position to open the pore upon PIP₂ binding. The open channel in the semi-swapped configuration exhibits substantial heterogeneity and resides within a particularly broad energy well. As a result, the conformational heterogeneity at the pore region limits its visualization by cryo-EM and only the open structure in the full-swapped configuration is captured. **b**, Avian TRPM8 similarly samples both fully- and semi-swapped configurations in the absence of stimuli, but the latter exhibits reduced heterogeneity compared to the mammalian channel, facilitating its structural determination. Menthol or cold at high pH biases the equilibration to semi-swapped configuration, promoting channel opening via a mechanism analogous to the mammalian channel. However, in avian TRPM8 the energy well corresponding to the open semi-swapped conformation is narrower than that of mammalian TPRM8, resulting in a more structurally well-defined state that is resolved by cryo-EM.

## Extended Data Table 1 | Cryo-EM data collection, refinement, and validation statistics

| | P. major TRPM8 semi-swapped, closed, unliganded, CaCl₂-free, 4°C adapted in cell vesicles (EMD-71395) (PDB 9P91) | P. major TRPM8 fully-swapped, closed/desensitized, unliganded, CaCl₂-free, 4°C adapted in cell vesicles (EMD-71394) (PDB 9P90) | P. major TRPM8 fully-swapped, closed, unliganded, CaCl₂-free, 4°C adapted in cell vesicles (EMD-71352) (PDB: 9P7S) | P. major TRPM8 undetermined example class 1, unliganded, CaCl₂-free, 4°C adapted in cell vesicles (EMD-74126) | P. major TRPM8 undetermined example class 2, unliganded, CaCl₂-free, 4°C adapted in cell vesicles (EMD-74127) | P. major TRPM8 fully-swapped, closed, unliganded, 5 mM CaCl₂, 4°C adapted in cell vesicles (EMD-74036) (PDB: 9ZCN) | P. major TRPM8 semi-swapped, closed, unliganded, 5 mM CaCl₂, 4°C adapted in cell vesicles (EMD-74037) (PDB: 9ZCO) | P. major TRPM8 fully-swapped, closed, menthol, CaCl₂-free in cell vesicles (EMD-74038) (PDB: 9ZCP) | P. major TRPM8 semi-swapped, closed, menthol, CaCl₂-free in cell vesicles (EMD-74039) (PDB: 9ZCQ) | P. major TRPM8 semi-swapped, open, menthol, CaCl₂-free in cell vesicles (EMD-74123) (PDB: 9ZEZ) |
|---|---|---|---|---|---|---|---|---|---|---|
| **Data collection and processing** | | | | | | | | | | |
| Microscope | Titan Krios TFS | Titan Krios TFS | Titan Krios TFS | Titan Krios TFS | Titan Krios TFS | Titan Krios TFS | Titan Krios TFS | Titan Krios TFS | Titan Krios TFS | Titan Krios TFS |
| Camera | Gatan K3 | Gatan K3 | Gatan K3 | Gatan K3 | Gatan K3 | Falcon 4i | Falcon 4i | Gatan K3 | Gatan K3 | Gatan K3 |
| Magnification | 105,000X | 105,000X | 105,000X | 105,000X | 105,000X | 130,000X | 130,000X | 105,000X | 105,000X | 105,000X |
| Voltage (kV) | 300 | 300 | 300 | 300 | 300 | 300 | 300 | 300 | 300 | 300 |
| Total electron dose | 47.7 | 47.7 | 47.7 | 47.7 | 47.7 | 50 | 50 | 47.7 | 47.7 | 47.7 |
| Defocus range (µm) | -1.0 to -2.5 µm | -1.0 to -2.5 µm | -1.0 to -2.5 µm | -1.0 to -2.5 µm | -1.0 to -2.5 µm | -1.0 to -2.5 µm | -1.0 to -2.5 µm | -1.0 to -2.5 µm | -1.0 to -2.5 µm | -1.0 to -2.5 µm |
| Pixel size (Å/pix) | 0.8189 | 0.8189 | 0.8189 | 0.8189 | 0.8189 | 0.94 | 0.94 | 0.8189 | 0.8189 | 0.8189 |
| Symmetry imposed | C4 | C4 | C4 | C4 | C4 | C4 | C4 | C4 | C4 | C4 |
| Micrographs (no.) | 35,382 | 35,382 | 35,382 | 35,382 | 35,382 | 20,661 | 20,661 | 57,257 | 57,257 | 57,257 |
| Initial particle images | 927,284 | 927,284 | 927,284 | 927,284 | 927,284 | 712,863 | 712,863 | 2,898,478 | 2,898,478 | 2,898,478 |
| Final particle images | 129,441 | 29,248 | 134,664 | 66,467 | 104,856 | 39,440 | 61,895 | 53,401 | 133,033 | 541,958 |
| Map resolution (Å) | 3 | 3.2 | 3.3 | 3.84 | 3.39 | 3.49 | 3.45 | 3.49 | 3.51 | 3.36 |
| FSC threshold | 0.143 | 0.143 | 0.143 | 0.143 | 0.143 | 0.143 | 0.143 | 0.143 | 0.143 | 0.143 |
| **Refinement** | | | | | | | | | | |
| Initial model used | N/A | 6O77 | 6O6A | - | - | 6O6A | N/A | 6O6A | N/A | N/A |
| Model resolution (Å) | 3.18 | 3.38 | 3.38 | - | - | 3.7 | 3.7 | 3.6 | 3.6 | 3.6 |
| FSC threshold | 0.5 | 0.5 | 0.5 | - | - | 0.5 | 0.5 | 0.5 | 0.5 | 0.5 |
| Map sharpening B | - | - | - | - | - | -105.5 | -121 | -130.7 | -155.9 | -178.7 |
| Model composition | | | | | | | | | | |
| Non-hydrogen atoms | 61448 | 60868 | 54368 | - | - | 54004 | 61248 | 53988 | 61292 | 57392 |
| Protein residues | 3892 | 3884 | 3312 | - | - | 3328 | 3896 | 3328 | 3896 | 3632 |
| Ligands | 0 | 0 | 0 | - | - | 0 | 0 | 0 | 4 | 4 |
| Lipids | 0 | 0 | 0 | - | - | 0 | 0 | 0 | 0 | 0 |
| Ions | 0 | 0 | 0 | - | - | 0 | 4 | 0 | 0 | 0 |
| B factor (Å²) | | | | | | | | | | |
| Protein | 142.55 | 140.89 | 36.98 | - | - | 100.03 | 52.4 | 101.09 | 119 | 87.72 |
| Ligands | - | - | - | - | - | - | 14.16 | - | 8.14 | 30.43 |
| R.m.s. deviations | | | | | | | | | | |
| Bond lengths (Å) | 0.003 | 0.002 | 0.003 | - | - | 0.004 | 0.004 | 0.004 | 0.004 | 0.004 |
| Bond angles (°) | 0.703 | 0.531 | 0.528 | - | - | 0.895 | 0.928 | 0.902 | 0.913 | 0.939 |
| Validation | | | | | | | | | | |
| MolProbity Score | 1.12 | 1.45 | 1.38 | - | - | 1.47 | 1.63 | 1.57 | 1.58 | 1.52 |
| Clash Score | 3.26 | 3.95 | 4.79 | - | - | 4.98 | 4.96 | 5.78 | 5.16 | 5.43 |
| Poor rotamers (%) | 0 | 0 | 0 | - | - | 0 | 0 | 0.31 | 1.2 | 0.43 |
| Ramachandran plot | | | | | | | | | | |
| Favored (%) | 98.65 | 96.03 | 97.31 | - | - | 96.68 | 94.58 | 96.19 | 96.29 | 96.48 |
| Allowed (%) | 1.35 | 3.97 | 2.69 | - | - | 3.32 | 5.42 | 3.81 | 3.71 | 3.52 |
| Disallowed (%) | - | - | - | - | - | 0 | 0 | 0 | 0 | 0 |

| | P. major TRPM8 fully-swapped, closed, putative desensitized menthol, CaCl₂-free in cell vesicles (EMD-74125) | P. major TRPM8 semi-swapped, high pH, CaCl₂, 4°C in GDN (EMD-71444) (PDB: 9PAR) | pmTRPM8 Chimera Human(V905-L938) semi-swapped, pH 7.4, CaCl₂, 4°C in GDN (EMD-74040) (PDB: 9ZCR) | Human TRPM8 fully-swapped, closed, unliganded, 4°C in cell vesicles (EMD-71391) (PDB: 9P8Y) | Human TRPM8 fully-swapped, closed, desensitized, unliganded, calcium-free 4°C in cell vesicles (EMD-74042) (PDB: 9ZCV) | Human TRPM8 fully-swapped, open, unliganded, calcium-free 4°C in cell vesicles (EMD-74124) (PDB: 9ZF0) | Human TRPM8 semi-swapped, unliganded, calcium-free 4°C in cell vesicles (EMD-74129) | Human TRPM8, fully-swapped, open, menthol, CaCl₂-free, 4°C in cell vesicles (EMD-71454) (PDB: 9PB5) | Human TRPM8 V915Y, fully-swapped, closed, CaCl₂, 4°C in GDN (EMD-74041) (PDB: 9ZCU) | Human TRPM8 V915Y, semi-swapped, CaCl₂, 4°C in GDN (EMD-74128) |
|---|---|---|---|---|---|---|---|---|---|---|
| **Data collection and processing** | | | | | | | | | | |
| Microscope | Titan Krios TFS | Titan Krios TFS | Titan Krios TFS | Titan Krios TFS | Titan Krios TFS | Titan Krios TFS | Titan Krios TFS | Titan Krios TFS | Titan Krios TFS | Titan Krios TFS |
| Camera | Gatan K3 | Gatan K3 | Gatan K3 | Gatan K3 | Gatan K3 | Gatan K3 | Gatan K3 | Falcon 4i | Gatan K3 | Gatan K3 |
| Magnification | 105,000X | 105,000X | 105,000X | 105,000X | 105,000X | 105,000X | 105,000X | 130,000X | 105,000X | 105,000X |
| Voltage (kV) | 300 | 300 | 300 | 300 | 300 | 300 | 300 | 300 | 300 | 300 |
| Total electron dose | 47.7 | 47.7 | 47.7 | 47.7 | 47.7 | 47.7 | 47.7 | 50 | 47.7 | 47.7 |
| Defocus range (µm) | -1.0 to -2.5 µm | -0.5 to -2.0 µm | -0.5 to -2.0 µm | -0.5 to -2.0 µm | -1.0 to -2.5 µm | -1.0 to -2.5 µm | -1.0 to -2.5 µm | -1.0 to -2.5 µm | -0.5 to -2.0 µm | -0.5 to -2.0 µm |
| Pixel size (Å/pix) | 0.8189 | 0.8189 | 0.8189 | 0.8189 | 0.8189 | 0.8189 | 0.8189 | 0.94 | 0.8189 | 0.8189 |
| Symmetry imposed | C4 | C4 | C4 | C4 | C4 | C4 | C4 | C4 | C4 | C4 |
| Micrographs (no.) | 57,257 | 15,931 | 19,158 | 9,197 | 74,792 | 74,792 | 74,792 | 17737 | 23,420 | 23,420 |
| Initial particle images | 2,898,478 | 3,357,206 | 1,720,102 | 1,215,074 | 2,552,130 | 2,552,130 | 2,552,130 | 484131 | 909,913 | 909,913 |
| Final particle images | 47,392 | 289,726 | 73,172 | 182,203 | 157,048 | 438,880 | 70,324 | 101753 | 96,564 | 28,291 |
| Map resolution (Å) | 3.88 | 3 | 3.03 | 2.6 | 3.65 | 3.71 | 4.44 | 3.5 | 2.86 | 2.99 |
| FSC threshold | 0.143 | 0.143 | 0.143 | 0.143 | 0.143 | 0.143 | 0.143 | 0.143 | 0.143 | 0.143 |
| **Refinement** | | | | | | | | | | |
| Initial model used | - | N/A | N/A | 8BDC | N/A | N/A | - | N/A | 8BDC | - |
| Model resolution (Å) | - | 3.36 | 3.3 | 2.76 | 3.7 | 3.9 | - | 3.74 | 3.1 | - |
| FSC threshold | - | 0.5 | 0.5 | 0.5 | 0.5 | 0.5 | - | 0.5 | 0.5 | - |
| Map sharpening B | - | -144.7 | -95.5 | -101.2 | -156.1 | -200 | - | -152.9 | -113.6 | - |
| Model composition | | | | | | | | | | |
| Non-hydrogen atoms | - | 58008 | 57272 | 58212 | 60044 | 57208 | - | 57768 | 58160 | - |
| Protein residues | - | 3632 | 3632 | 3884 | 4008 | 3844 | - | 3832 | 3884 | - |
| Ligands | - | 4 | 0 | 0 | 0 | 0 | - | 0 | 0 | - |
| Lipids | - | 0 | 0 | 0 | 0 | 0 | - | 4 | 0 | - |
| Ions | - | 0 | 0 | 0 | 0 | 0 | - | 0 | 0 | - |
| B factor (Å²) | | | | | | | | | | |
| Protein | - | 37.35 | 61.57 | 13.96 | 110.75 | 16.74 | - | 38.34 | 56.56 | - |
| Ligands | - | 37.49 | - | - | - | - | - | 21.25 | - | - |
| R.m.s. deviations | | | | | | | | | | |
| Bond lengths (Å) | - | 0.005 | 0.004 | 0.003 | 0.005 | 0.004 | - | 0.005 | 0.006 | - |
| Bond angles (°) | - | 1.031 | 0.937 | 0.524 | 1.001 | 0.984 | - | 0.934 | 0.969 | - |
| Validation | | | | | | | | | | |
| MolProbity Score | - | 1.39 | 1.39 | 1.13 | 1.93 | 1.7 | - | 1.65 | 1.33 | - |
| Clash Score | - | 4.93 | 4.7 | 3.4 | 7.92 | 7.66 | - | 6.5 | 3.64 | - |
| Poor rotamers (%) | - | 0 | 0 | 0 | 0.13 | 0.42 | - | 0 | 0 | - |
| Ramachandran plot | | | | | | | | | | |
| Favored (%) | - | 97.32 | 97.2 | 98.85 | 91.72 | 95.98 | - | 95.82 | 96.94 | - |
| Allowed (%) | - | 2.68 | 2.8 | 1.15 | 8.28 | 4.02 | - | 4.18 | 3.06 | - |
| Disallowed (%) | - | 0 | 0 | 0 | 0 | 0 | - | 0 | 0 | - |

# Reporting Summary

Please do not complete any field with "not applicable" or n/a.  Refer to the help text for what text to use if an item is not relevant to your study.
For final submission: please carefully check your responses for accuracy; you will not be able to make changes later.

## Statistics

For all statistical analyses, confirm that the following items are present in the figure legend, table legend, main text, or Methods section.

| n/a | Confirmed | |
|---|---|---|
| ☐ | ☒ | The exact sample size (*n*) for each experimental group/condition, given as a discrete number and unit of measurement |
| ☐ | ☒ | A statement on whether measurements were taken from distinct samples or whether the same sample was measured repeatedly |
| ☐ | ☒ | The statistical test(s) used AND whether they are one- or two-sided<br>*Only common tests should be described solely by name; describe more complex techniques in the Methods section.* |
| ☒ | ☐ | A description of all covariates tested |
| ☒ | ☐ | A description of any assumptions or corrections, such as tests of normality and adjustment for multiple comparisons |
| ☐ | ☒ | A full description of the statistical parameters including central tendency (e.g. means) or other basic estimates (e.g. regression coefficient) AND variation (e.g. standard deviation) or associated estimates of uncertainty (e.g. confidence intervals) |
| ☐ | ☒ | For null hypothesis testing, the test statistic (e.g. *F*, *t*, *r*) with confidence intervals, effect sizes, degrees of freedom and *P* value noted<br>*Give P values as exact values whenever suitable.* |
| ☒ | ☐ | For Bayesian analysis, information on the choice of priors and Markov chain Monte Carlo settings |
| ☒ | ☐ | For hierarchical and complex designs, identification of the appropriate level for tests and full reporting of outcomes |
| ☒ | ☐ | Estimates of effect sizes (e.g. Cohen's *d*, Pearson's *r*), indicating how they were calculated |

*Our web collection on statistics for biologists contains articles on many of the points above.*

## Software and code

Policy information about availability of computer code

| Data collection | (HDX-MS): Deuterated proteins were proteolyzed and separated using an LC valve system from Trajan LEAP HDX Base automation platform coupled with Thermofisher UltiMate-3000 pumps. Peptide mass analysis was done by a Thermofisher Q Exactive Orbitrap mass spectrometer. | (Cryo-EM): Screening was carried out on either a Glacios TEM or Talos Arctica TEM using SerialEM. CryoEM data collection was carried out on a Titan Krios using SerialEM. | (Ratiometric calcium imaging): For ratiometric calcium imaging, an inverted microscope equipped with a Grasshopper 3 (FLIR) and Sutter Lambda LS Illuminator was controlled using MicroManager 2.0. |
|---|---|---|---|
| Data analysis | (HDX-MS): For HDX-MS data analysis, peptides were identified using SearchGUI (v4.0.25) and Byonic (Protein Metrics). Deuteration calculation for each peptide was performed with HDExaminer 3.4 (Trajan) and HX-Express3.<br>Free energy calculation from HDX data was performed with Python 3.12.1. | (Cryo-EM): For cryoEM data analysis and representation, MotionCor2, Relion5.0.0_cu12.2, cryoSPARC4.7, Phenix1.2.1, Coot0.9.6, and ChimeraX1.8 were used. | (Ratiometric calcium imaging): For ratiometric calcium imaging data analysis, FIJI software (NIH, v2.14) was used during analysis and to generate ratiometric or deltaF/F images. Statistical analyses were done using GraphPad Prism (v10.5.0). |

For manuscripts utilizing custom algorithms or software that are central to the research but not yet described in published literature, software must be made available to editors and reviewers. We strongly encourage code deposition in a community repository (e.g. GitHub). See the Nature Portfolio guidelines for submitting code & software for further information.

## Data

Policy information about availability of data

All manuscripts must include a data availability statement. This statement should provide the following information, where applicable:

- Accession codes, unique identifiers, or web links for publicly available datasets
- A description of any restrictions on data availability
- For clinical datasets or third party data, please ensure that the statement adheres to our policy

For cryo-EM data, EM volumes and fitted atomic coordinates shall be accessible through the EMDB and PDB, respectively.
For HDX-MS data, raw mass spectrometry files, peptide identification results, and HDX calculation results for all available peptides have been deposited to the ProteomeXchange Consortium via the PRIDE partner repository with the dataset identifier PXD064468.

# Research involving human participants, their data, or biological material

Policy information about studies with human participants or human data. See also policy information about sex, gender (identity/presentation), and sexual orientation and race, ethnicity and racism.

| | |
|---|---|
| Reporting on sex and gender | N/A |
| Reporting on race, ethnicity, or other socially relevant groupings | N/A |
| Population characteristics | N/A |
| Recruitment | N/A |
| Ethics oversight | N/A |

Note that full information on the approval of the study protocol must also be provided in the manuscript.

# Field-specific reporting

Please select the one below that is the best fit for your research. If you are not sure, read the appropriate sections before making your selection.

☑ Life sciences    ☐ Behavioural & social sciences    ☐ Ecological, evolutionary & environmental sciences

For a reference copy of the document with all sections, see nature.com/documents/nr-reporting-summary-flat.pdf

# Life sciences study design

All studies must disclose on these points even when the disclosure is negative.

| | | | |
|---|---|---|---|
| Sample size | (HDX-MS):<br>Sample sizes:<br>For HDX-MS, all TRPM8 peptides identified were used for analysis. | (Cryo-EM):<br>Sample sizes:<br>Sample sizes were not predetermined for cryo-EM datasets | Ratiometric calcium imaging:<br>Sample sizes:<br>For statistical comparison of responses, we collected data from 40-86 cells. Sample sizes were not predetermined in these experiments. |
| Data exclusions | Data exclusions:<br>When peptides identified by search algorithms show kinetics of deuterium exchange that are inconsistent with at least three overlapping peptides, we defined them as mis-identified peptides and manually excluded them from analysis. | Data exclusions:<br>Images were excluded based on CTF estimations provided from cryoSPARC (<4Å). Particles were discarded based on standard practices in the single-particle cryoEM field, and exclusions were determined based on 2D or 3D classification. Details are provided as Extended Data Figures 1 and 3. | Data exclusions:<br>For analysis of ratiometric calcium imaging data, cells showing substantial movement during imaging that cannot be further analyzed, and/or cells with abnormally high baseline calcium levels/responses (indicative of unhealthy cells) are routinely excluded from analysis. In these datasets, we did not analyze these cells. |
| Replication | Replication:<br>One-two biological replicates were performed. HDX experiments were conducted with multiple labeling times. Peptides with different charge states were analyzed. Detailed replication information is provided in Extended Data Table | Replication:<br>Sample purifications used in EM were done with at least two to three independent instances. | Replication:<br>Temperature stimuli were repeated twice for a given imaging experiment.<br>Randomization:<br>Randomization was not applied during imaging experiments.<br>Blinding:<br>Blinding is not required because the studies are mechanism focused. |
| Randomization | Randomization:<br>HDX reactions were injected in random order. | Randomization:<br>GSFSC calculations were done using random subsets of finalized particle sets in either cryoSPARC or RELION. | |
| Blinding | Blinding:<br>Blinding is not required because HDX-MS studies is mechanism focused. | Blinding:<br>Blinding is not required because the studies are mechanism focused. | |

# Behavioural & social sciences study design

All studies must disclose on these points even when the disclosure is negative.

| | |
|---|---|
| Study description | N/A |
| Research sample | N/A |
| Sampling strategy | N/A |
| Data collection | N/A |
| Timing | N/A |
| Data exclusions | N/A |
| Non-participation | N/A |
| Randomization | N/A |

# Ecological, evolutionary & environmental sciences study design

All studies must disclose on these points even when the disclosure is negative.

| | |
|---|---|
| Study description | N/A |
| Research sample | N/A |
| Sampling strategy | N/A |
| Data collection | N/A |
| Timing and spatial scale | N/A |
| Data exclusions | N/A |
| Reproducibility | N/A |
| Randomization | N/A |
| Blinding | N/A |

Did the study involve field work? ☐ Yes ☒ No

## Field work, collection and transport

| | |
|---|---|
| Field conditions | N/A |
| Location | N/A |
| Access & import/export | N/A |
| Disturbance | N/A |

# Reporting for specific materials, systems and methods

We require information from authors about some types of materials, experimental systems and methods used in many studies. Here, indicate whether each material, system or method listed is relevant to your study. If you are not sure if a list item applies to your research, read the appropriate section before selecting a response.

### Materials & experimental systems

| n/a | Involved in the study |
|---|---|
| ☒ | ☐ Antibodies |
| ☐ | ☒ Eukaryotic cell lines |
| ☒ | ☐ Palaeontology and archaeology |
| ☒ | ☐ Animals and other organisms |
| ☒ | ☐ Clinical data |
| ☒ | ☐ Dual use research of concern |
| ☒ | ☐ Plants |

### Methods

| n/a | Involved in the study |
|---|---|
| ☒ | ☐ ChIP-seq |
| ☒ | ☐ Flow cytometry |
| ☒ | ☐ MRI-based neuroimaging |

## Antibodies

| | |
|---|---|
| Antibodies used | N/A |
| Validation | N/A |

# Eukaryotic cell lines

Policy information about cell lines and Sex and Gender in Research

| | |
|---|---|
| Cell line source(s) | HEK293T cell line was purchased from ATCC (CRL-3216) |
| Authentication | No authentication was performed on the purchased cell line. |
| Mycoplasma contamination | No mycoplasma contamination test was conduction on the purchased cell line. |
| Commonly misidentified lines (See ICLAC register) | No commonly misidentified cell lines were utilized in this study. |

# Palaeontology and Archaeology

| | |
|---|---|
| Specimen provenance | N/A |
| Specimen deposition | N/A |
| Dating methods | N/A |

☐ Tick this box to confirm that the raw and calibrated dates are available in the paper or in Supplementary Information.

| | |
|---|---|
| Ethics oversight | N/A |

Note that full information on the approval of the study protocol must also be provided in the manuscript.

# Animals and other research organisms

Policy information about studies involving animals; ARRIVE guidelines recommended for reporting animal research, and Sex and Gender in Research

| | |
|---|---|
| Laboratory animals | N/A |
| Wild animals | N/A |
| Reporting on sex | N/A |
| Field-collected samples | N/A |
| Ethics oversight | N/A |

Note that full information on the approval of the study protocol must also be provided in the manuscript.

# Clinical data

Policy information about clinical studies
All manuscripts should comply with the ICMJE guidelines for publication of clinical research and a completed CONSORT checklist must be included with all submissions.

| | |
|---|---|
| Clinical trial registration | N/A |
| Study protocol | N/A |
| Data collection | N/A |
| Outcomes | N/A |

# Dual use research of concern

Policy information about dual use research of concern

## Hazards

Could the accidental, deliberate or reckless misuse of agents or technologies generated in the work, or the application of information presented in the manuscript, pose a threat to:

| No | Yes | |
|---|---|---|
| ☒ | ☐ | Public health |
| ☒ | ☐ | National security |
| ☒ | ☐ | Crops and/or livestock |
| ☒ | ☐ | Ecosystems |
| ☒ | ☐ | Any other significant area |

### Experiments of concern

Does the work involve any of these experiments of concern:

| No | Yes | |
|---|---|---|
| ☒ | ☐ | Demonstrate how to render a vaccine ineffective |
| ☒ | ☐ | Confer resistance to therapeutically useful antibiotics or antiviral agents |
| ☒ | ☐ | Enhance the virulence of a pathogen or render a nonpathogen virulent |
| ☒ | ☐ | Increase transmissibility of a pathogen |
| ☒ | ☐ | Alter the host range of a pathogen |
| ☒ | ☐ | Enable evasion of diagnostic/detection modalities |
| ☒ | ☐ | Enable the weaponization of a biological agent or toxin |
| ☒ | ☐ | Any other potentially harmful combination of experiments and agents |

# Plants

| | |
|---|---|
| Seed stocks | N/A |
| Novel plant genotypes | N/A |
| Authentication | N/A |

# ChIP-seq

### Data deposition

☐ Confirm that both raw and final processed data have been deposited in a public database such as GEO.

☐ Confirm that you have deposited or provided access to graph files (e.g. BED files) for the called peaks.

| Data access links
May remain private before publication. | N/A |
|---|---|
| Files in database submission | N/A |
| Genome browser session
(e.g. UCSC) | N/A |

### Methodology

| | |
|---|---|
| Replicates | N/A |
| Sequencing depth | N/A |
| Antibodies | N/A |
| Peak calling parameters | N/A |
| Data quality | N/A |

| Software | N/A |
|---|---|

# Flow Cytometry

## Plots

Confirm that:

☐ The axis labels state the marker and fluorochrome used (e.g. CD4-FITC).

☐ The axis scales are clearly visible. Include numbers along axes only for bottom left plot of group (a 'group' is an analysis of identical markers).

☐ All plots are contour plots with outliers or pseudocolor plots.

☐ A numerical value for number of cells or percentage (with statistics) is provided.

## Methodology

| Sample preparation | N/A |
|---|---|
| Instrument | N/A |
| Software | N/A |
| Cell population abundance | N/A |
| Gating strategy | N/A |

☐ Tick this box to confirm that a figure exemplifying the gating strategy is provided in the Supplementary Information.

# Magnetic resonance imaging

## Experimental design

| Design type | N/A |
|---|---|
| Design specifications | N/A |
| Behavioral performance measures | N/A |

| Imaging type(s) | N/A |
|---|---|
| Field strength | N/A |
| Sequence & imaging parameters | N/A |
| Area of acquisition | N/A |

Diffusion MRI    ☐ Used    ☐ Not used

## Preprocessing

| Preprocessing software | N/A |
|---|---|
| Normalization | N/A |
| Normalization template | N/A |
| Noise and artifact removal | N/A |
| Volume censoring | N/A |

## Statistical modeling & inference

| Model type and settings | N/A |
|---|---|
| Effect(s) tested | N/A |

Specify type of analysis: ☐ Whole brain ☐ ROI-based ☐ Both

Statistic type for inference

N/A

(See Eklund et al. 2016)

Correction

N/A

## Models & analysis

| n/a | Involved in the study |
|-----|----------------------|
| ☒ | Functional and/or effective connectivity |
| ☒ | Graph analysis |
| ☒ | Multivariate modeling or predictive analysis |

Functional and/or effective connectivity

N/A

Graph analysis

N/A

Multivariate modeling and predictive analysis

N/A

