## [Peer Review File · Nature]

Structural energetics of cold sensitivity

Corresponding Author: Professor David Julius

Version 0:

Reviewer comments:

Referee #1

(Remarks to the Author)

This is an outstanding, clearly argued study of a conducting conformation of the avian and human TRPM8 'cold' temperature-gated channels. This conformation appears as a result of a new 'semi-swap' form in which the S6 helix is significantly rearranged, also altering the TRP helix, and the outer pore region. The authors are able to visualize this conformation by preparing samples with MacKinnon's recent vesicle-purification method, which avoids detergent solubilization, and they employ extensive experiments and thermodynamic analysis of hydrogen-deuterium exchange to convincingly buttress their conclusions. My comments are relatively minor, and recommend publication with minor modifications.

1. Fig. 1 shows the paper's first view of novel semi-swap conformation of avian M8 involving S6 and the outer pore region from a previously known full-swap "desensitized" conformation. The text describes the main features of this conformation clearly and succinctly. Points:

a. Fig. 1 legend, or main text, should indicate conditions experienced by the vesicles before spreading on grid. Cold? Apo or menthol? The methods state that vesicles are kept on ice before grid preparation, so these are presumed to be a cold-adapted apo condition. The reader should be reminded of this key condition in the main body of the paper.

b. Fig. 1g: It's formally correct, but it is a bit confusing to claim that the conformational change into the semi-swap form widens the selectivity-region. The figure actually shows that it is a widening of the "upper constriction" around a.a. 905 from a quite wide-open hole to an even wider one, while leaving the "lower" constriction (around 965-969) unchanged, and the channel presumably still nonconducting. I would change the final phrase to "...Y905 at the selectivity filter, thereby further widening it".

2. p. 6: Some readers may be unfamiliar with H-D exchange and so will be confused by your use of simply "peptides" at the top of the page. It might prevent confusion to reword as "...the majority of the peptides derived from proteolysis following H-D exchange..." Also, a reference is needed after "3 kcal/mol" in the first paragraph.

3. Fig. 2b legend: Define the "nne state".

4. p. 24: Readers may be confused by the use of FFC-8, a detergent, in the preparation. You may want to reassure them that the 0.5 mM concentration used here is well below the ~3 mM CMC.

5. Did the authors attempt to find out if the TRPM8 apo structure in vesicles pre-adapted to a warm temperature (34 °C, say) before fast freezing, would fail to show the semi-swap form, or a far lower prevalence of that form? Although not necessary to perform anew, if true, it would be an interesting detail to add if the authors had already done this.

6. It might be interesting for general readers to speculate/discuss why avians, which have higher metabolic rates and body temperatures of 38–41 °C, might have different M8 cold sensitivity and why this might be adaptive.

Minor: 'remain elusive' in the second paragraph is an overused phrase.

Referee #2

(Remarks to the Author)

How temperature is sensed on a molecular level is one of the most exciting but also one of the most enigmatic questions in modern biophysics. While the role of transient receptor potential (TRP) ion channels in hot and cold sensing has been well established, the underlying molecular details of how changes in temperature result in ion channel pore opening remain unclear. Over the years, different proposals highlighting the potential role of changes in lipid/membrane interactions, partial unfolding of channel regions or dedicated thermosensing elements have been brought forward. However, to date no unifying, or alas any satisfactory explanation for how any of the many "ThermoTRPs" works is available. Even more confounding (or awe-inspiring, depending on one's perspective) is the observation that in many cases, small molecules evoke the same physiological responses as a temperature stimulus, e.g. capsaicin from chili peppers tasting "hot" or menthol resulting in a "cold" sensation. Although the penultimate outcome of a thermal and a chemical stimulus must be the same, i.e. a change in the opening probability of the responsive ion channel, it is not clear whether the underlying molecular pathways are identical or merely partially overlapping. Furthermore, it is well known that ThermoTRP channels from different animals (or their splice variants) can have different threshold temperatures.

In the current manuscript, Choi and Lin et al. set out to answer the enigma of cold sensing, using the cold sensor TRP melastatin 8 (TRPM8) and its activator menthol from mammalian and avian species. To this end, they combine cryo-electron microscopy in native membranes with hydrogen deuterium exchange mass spectrometry to carry out structural and thermodynamic analyses of temperature-driven changes in TRPM8 channels.

Overall, the paper covers a highly important and very exciting topic. The methodological combination is well suited to investigate the question of interest, but the descriptions may be difficult to follow for non-expert readers in places. While the authors present very interesting new structures and ideas, I remain somewhat unconvinced that the mystery of cold sensing is really solved. My major concern in this regard is that obtaining a "thermally activated" state required substantial "experimental contortions". Strictly speaking, the authors assume a converging mechanism for menthol/cold activation in a native membrane environment, but do not show it. Rather, they require use of an avian channel at high pH in detergent to obtain the open semi-swapped state that they assume presents the structural basis of cold sensing. It thus remains unclear whether they really identified a generalizable mechanism of cold sensing for TRPM8 channels.

In addition, the HDX experiments that are the basis for the thermodynamic model of cold sensing were carried out with detergent-solubilized protein. This seems at odds with the strong statements by the authors that the native environment is crucial to understand thermosensing.

Main points:

In the introduction, I am missing a few explanations about what is known about cold sensing of avian TRPM8, and what are the threshold temperatures compared to the mammalian channel. Is cold/menthol sensitivity widely present across bird TRPM8s? Are the residues for menthol binding conserved? Was there any particular reason for choosing *P. major* TRPM8 other than high homogeneity observed in previous studies? And most importantly: What has been proposed so far as potential molecular mechanisms for temperature sensing? It would be good to have an overview of currently debated ideas.

The authors rightfully state that one possible concern about determining the structural basis of thermosensing could be the loss of relevant states by extracting protein from the membrane or generally studying it under non-native conditions. However, since they use a heterologous expression system, i.e. a human cell line to express an avian channel, they may also miss crucial binding partners or lipids. Just to clarify, I am not arguing against the use of heterologous expression systems, but the use of the word "native" under these particular circumstances should probably be used very carefully and differences in the lipid composition of avian and mammalian cell membranes should be described up-front as far as they are known. At the very least, it should be clearly stated in the main text ('Cells expressing the *Parus major* channel...' (p. 4, 2nd sentence)), that the expression system is a HEK293-based system.

Why do the authors presume it has been so difficult to reconstitute TRP(M8) channels into lipid-based systems, compared to other membrane protein families? Across all available structures and TRP channel subfamilies (to the best of my knowledge), it seems that the transmembrane regions are not significantly more dynamic than those of other channels/membrane proteins, including transporters which can undergo major conformational changes, but generally there are not many studies available for TRP channels in liposomes? However, it also seems from the authors own observations (Fig. S1), that the TRPM8 transmembrane domain is much less well-defined than the cytosolic region. Is this difficulty with reconstitution therefore generally more pronounced for thermoTRPs than for other TRP channel family members or a specialty of TRPM8? If this is a cold-sensor specificity, how do TRPA1 structures fare in comparison, which can act as a cold or heat sensor depending on the host organism?

Fig. 1e,f: Experimental densities for side chains should be shown along structural cartoons; how reliable is the sidechain conformation?

Fig. 2 and p. 5, quantification of fully- vs semi-swapped configuration: Can you actually state the respective % of the populations and the details of the incubation (i.e. temperature, conc. of menthol, length of incubation) in the text or the figure legend? Are shorter incubation times or those with sub-stoichiometric menthol concurrent with less populated semi-swapped states? Do you see full occupancy for the menthol density? And in addition, does the significant presence of at least one

additional state in addition to semi-/fully-swapped mean that assumption of a two-state model is too simple? This is also important for the interpretation of the HDX data; how do you interpret HDX if >60% of particles in cryo-EM are neither classified as fully-swapped or semi-swapped? (And what exactly is the definition of “undetermined” here: resolution too low? differences between subunits?)

Pages 5-6: A brief section in the main text should be added to describe how exactly the H/D-exchange rates were converted into free energy and enthalpy differences. Specifically, the authors assume “minimal heat capacity changes” (p. 8). I suggest to clarify, if they minimize heat capacity changes in their fit or if they set heat capacity changes to zero, possibly by including the equations by which they convert HDX rates into free energy and enthalpy changes in the methods section. A table with an overview about the detected peptides and their exchange rates under different conditions should be added to the Supplement.

Page 5: While the authors first observe the novel semi-swapped configuration in cell membranes, they perform the HDX-MS experiments with detergent-solubilized channels. Could the authors clarify whether HDX-MS experiments were not possible or not successful in the cell membrane context?

Furthermore, the number of replicates for some of the HDX data (n=1) is concerning. I am aware that it can be tremendously difficult to properly replicate HDX-MS data, but seeing that there are enormously important conclusions drawn from these data sets, this concern should be addressed.

Another major question concerns the kinetics: the time scale of HDX for interconversion between swapped and semi-swapped is in the range of (min – hr), compared to a (presumably sub-ms) time scale required for cold sensing; how does that fit together? If exposure to cold would trigger a structural interconversion from fully-swapped to semi-swapped, would this be too slow as a sensing mechanism?

What are the pathways the authors propose for the transition between fully-swapped and semi-swapped configurations?? MD simulations may be quite useful here if they can cover the necessary time scales. At the very least the authors should make an effort to describe the potential structural conversions required globally to interconvert swapped and semi-swapped states.

Do TRPM8 channels desensitize with cold/menthol? Do the authors assume that the desensitized and the closed states are structurally identical? In the Extended Data Table, there is a mention of a desensitized state; maybe the authors can explain this in more detail.

The thermodynamic interpretation of HDX requires the EX2 exchange regime. Are these conditions uniformly met? The uptake kinetics should be shown as a supplementary figure across all measured time points and replicates, at least for a more widely distributed selection of peptides across the channel structure, and also including for peptides which show no changes between states as well as examples of isotopic distribution to differentiate between EX1 and EX2. Furthermore, a table with the detected peptides should be provided. Please also map HDX onto a structural model of TRPM8 for a global overview of the effects of menthol and temperature (and comparison of human vs avian channel). This would be a useful supplementary figure.

Fig. 3: Using HDX-MS, the authors show that menthol binding stabilizes the TRP helix. It seems (Ext. Data Table 2) that the data shown in Fig. 3d stem from two biological replicates, so there should be error bars available? What about other channel regions (see Fig. 3c, around residue 875) that seem to differ quite substantially between human and avian TRPM8? The energy landscape in Fig. 6 has both fully-swapped and semi-swapped closed states at similar energy levels for both human and avian TRPM8, but that does not seem to match the observations by HDX??

Page 8 bottom, pH sensitivity: The authors propose a pH sensitivity pocket created by Y905, R875, E896, D907, D908. What is the proposed molecular mechanism of pH sensitivity? The pKa values of these amino acids are either well above or below pH 7 (12/>10 for Arg/Tyr, <5 for E/D). Typically, buried salt bridges result in pKa's even further shifted from physiological pH and to allow protonation of E and D, the pKa values of these acidic residues must be shifted significantly towards neutrality? Is there any evidence for this?

Page 9, “Our calcium imaging experiments indicate that breaking the species specific Y905 interface potentiates cold-evoked responses of the avian channel”: What exactly do the authors mean by “breaking”? What constitutes the break, i.e. the mutation, Ca binding?

Page 12 and Fig. 6: The “cold semi-swap model” the authors present seems to be independent of the cytoplasmic regions? In other TRP channels, e.g. TRPV3, it has been proposed that the cytosolic regions are quite relevant for thermosensing due to conformational changes. Is there any indication here that the MHRs are also important?

Furthermore, the “cold-activated” structure determined here (avian species, high pH, 4 °C) is structurally different from that of mouse TRPM8 with a cooling compound, which the authors state themselves (p. 9). So naively I have to ask: which of these constitutes a true “cold-activated” state? Is there more than one cold activated state? Or do chemically and thermally activated states differ? Is the observed state in the present manuscript then potentially a pH-mediated activated open state?

Furthermore, the authors should at least discuss the available data for TRPV6 and TRPV3 by the Sobolevsky and Scheuring labs that show that the transmembrane interfaces of TRP channels are relatively pliable, which may be exactly

what is also observed here for TRPM8. In the case of TRPV6 for instance, introduction of a single point mutant can lead to a structural rearrangement from a swapped to a non-swapped state (PMID: 28878326, PMID: 27296226). In TRPV3, it seems that entire subunits can be relatively easily exchanged (PMID: 37648856).

Across all figures, pointing out the PDB codes for the structures shown would really help with readability. Throughout the text, it would really help to always clearly state which TRPM8 (which species) you talk about at any given point.

Minor points:

Introduction: "In the case of TRPM8, structural heterogeneity may be exacerbated by the channel's cold sensitivity, resulting in a range of conformational substates that are too transient or numerous to register as major structural subclasses during cryo-EM analysis (Extended Data Fig. 1).": It would be helpful to have a table/figure comparing all available TRPM8 structures and the respective resolutions under the different experimental conditions, since it is also possible that structural heterogeneity stems from the method of preparation.

Page 4 (Description of the swapped architecture): A brief outline of the general architecture of TRP channel transmembrane domains for the uninitiated reader (here or in the introduction) would be very helpful. For instance, it presumably remains completely unclear to the non-ion channel reader that transmembrane helix S6 constitutes a pore forming helix at this point in the manuscript.

Page 4: It may be a bit confusing to begin with "From these data, we obtained high resolution (3 – 3.5Å) structures, including the previously described closed and desensitized states 18,20,27 (Extended Data Fig. 3), as well as a novel substate whose most notable feature is a distinct domain swap." when later it is explained that the "fully swapped state" is the canonical architecture?

Fig. S1a: what are the 3 non-TRPM8 references used?

Figs S3/S4: It would be really helpful to add the PDB/EMDB codes to these figure to more easily compare to the Ext. Data Table 1. Of note, there is no mention of the "semi-swapped" state in hsTRPM8 in the table? There is also no mention of the V915Y mutant in the Table so it seems there are only WT structures listed? Open/closed state is not indicated for all channels. Finally (this is nitpicky, but may help to avoid confusion), extended table 1 states that structures are "in membranes", whereas throughout the rest of the figures "in vesicles" is used.

A supplementary figure that has all structures, their organism, open/closed state, ligand, temperature state (as well as pH), and PDB code side by side would go a long way to improve the readability of this manuscript.

Fig. 2b: Spelling mistake 'nne state' should read 'one state'

Fig. 2c: The data seem to have different y-scales for the different time points; why not use a uniform scaling?

Ext. Data Table 2 (HDX): This is very hard to read. Instead of writing "same as XXX", why not just put the actual conditions (one can still color code or spell out that samples were treated similarly)?

Page 5: The second section of the main text is called "Agonists modulate dynamic equilibrium between states". The authors only show modulation for one agonist (menthol), which does not justify to speak of "agonists" in plural in a strict sense.

Page 5 bottom: When the effects of menthol on the population ratio are described and Fig. 2a is introduced it should be clearly stated that this is in reference to the avian channel.

Page 5: Reference 31 refers to a seminal NMR paper on hydrogen exchange, but its not a suitable reference for HDX-MS. In addition to this reference to HDX, the authors should include a suitable reference to an HDX-MS paper that introduces the mapping to specific residues with the MS approach.

Fig. 2: The use of the term "population" seems not to be consistent in the presentation of the HDX-MS data. In the main text, the populations are identified with configurations (semi-swapped vs. fully-swapped), while in Fig. 2d, "population" refers to the weights in the bimodal distribution/ the fraction of the two distributions (as correctly named in e again). For a very slow exchange, it would appear to be possible to assign the population of peptides to a certain dynamic state, but in the "slow" exchange scenario (2b right side) this seems to be misleading to me.

Page 6, top, "Such features of bimodality in deuterated mass spectra suggest the coexistence of two conformational populations, with their energetics differing by at least 3 kcal/mol.": Should read "..., with their free energies of unfolding (ΔG) differing by at least ...".

Page 7, bottom and Fig. 3, "We resolved structures of human and avian TRPM8 channels in vesicles in the presence of menthol.": Do these structures correspond to the swapped, semi-swapped, or both conformations?

Page 7, "However, our structural analyses in vesicles, as well as HDX-MS measurements, show that the human channel exhibits much greater energetic heterogeneity than previously appreciated.": What exactly do you mean by "energetic heterogeneity"?

Fig. 3: The authors present convincing evidence for ligand density in avian and human channels, into which they dock menthol. However, the density itself does not seem to support a clear positioning of menthol in the binding pocket. The authors should explicitly state the model character in the legend to Fig. 3.

Page 5 subtitle and title Fig. 5: What protein is used: avian high pH or avian Y mutant at high pH? This should be clearly stated in subtitle and figure title.

Fig. 4c: Please present a phylogenetically deeper alignment for avian/mammalian TrpM8 (e.g. in supplement in extension to Fig. 4c). Are there bird TRPM8 with the human valine residue or vice versa? How broadly can the term avian/mammalian TrpM8 be used based on this amino acid difference?

Fig. 6a shows a fully-swapped open conformation for the mammalian channel and a semi-swapped open conformation, but it seems that only the fully-swapped open conformation was observed/resolved by EM (p. 10, bottom). Is there structural evidence for the semi-swapped open conformation? What would be the function of two different open conformations? Functionally redundant? Semi-swapped not functionally important in mammalian systems?

Referee #3

(Remarks to the Author)

I co-reviewed this manuscript with one of the reviewers who provided the listed reports.

Version 1:

Reviewer comments:

Referee #1

(Remarks to the Author)

The authors have addressed all my suggestions for changes.

Referee #2

(Remarks to the Author)

We would like to thank the authors for their patience and the extensive replies to our comments. This is very much appreciated. Readability has been much improved in the revised manuscript. The author's point that the presented cold-activated TRPM8 structures presented were obtained with the least possible "experimental contortions" is well taken. We agree that especially the use of truly native constructs without mutations is very important and necessary to elucidate the mechanism of cold sensing. Their point about the importance of the structure determination in native membranes and the presumption that the semi-swapped state has presumably been missed in detergent-preparations of TRPM8 due to discarded particles seems very intuitive. Finally, we do wish to clarify that our previous comment about the HDX measurements in detergent were not meant as a major criticism, as we are well aware of the technical challenges of HDX on membrane proteins and thus appreciate the major effort of the authors in the current study. Accordingly, the increase in replicate number for these samples in the revised manuscript should also be positively mentioned.

Overall, the paper is a very interesting, technically challenging study and introduces important methods that have been applied to other membrane proteins (cryoEM in cell-derived vesicles and HDX) to the TRP channel field. Without a doubt, this new combination of methods sheds light onto new and unexpected aspects of the structural biology of TRPM8 such as the new semi-swapped configuration, the modulation of dynamics by menthol, the structural impact of menthol binding, the differential thermodynamics in human and avian orthologues and the molecular origin thereof. On the other hand, the conclusions (dynamic equilibria, structural changes in response to a stimulus) result in an expected outcome, i.e. opening of the channel upon cold exposure, but that does not yet explain what the "structural basis of cold sensitivity" is or where/how it is encoded in the protein.

In summary, also with the revised manuscript, it is still difficult to gauge what the authors think the structural basis of cold sensitivity actually is, even 'just' for TRPM8. If you had to say it in 1-2 sentences, what is it? A clear, concise statement would go a long way.

Please see below for more specific replies and comments to the revised manuscript and rebuttal letter (authors' statements in quotation marks):

From new introduction "Indeed, of the numerous TRPM8 conformations in apo or ligand-bound states reported thus far 22,24-28, none convincingly represent a purely cold-evoked open state or provide structural insights into thermal gating mechanisms, which may reflect the fact that they have all been determined with detergent-solubilized material."

"We now focus on introducing limitations of current thermodynamic and structural understanding of temperature sensing in heat or cold activated channels, with better articulation of the rationale of combining cryo-EM in membrane vesicles with HDX-MS to address mechanism of temperature sensing from both structural and thermodynamic perspectives."

Thank you very much for including this in the revised manuscript. We cannot and do not want to judge on the quality or conclusions of various pre-prints, and of course, it remains up the authors (and editors) whether to take preprints into account

at all, but since the authors themselves brought up the paper by Lee and colleagues, we would like to at least alert you to the fact that previously the Nimigean lab also proposed a cold sensing mechanism through salt bridges and lipids (<https://www.biorxiv.org/content/10.1101/2025.06.03.657524v1>).

Unfortunately we cannot properly judge whether the newly added human TRPM8 cold-activated structure in the current manuscript now convincingly presents an open state as this model was not included in the uploaded files by the authors. This might be the result of a misunderstanding: as of now, we do not have access to the structures that were uploaded to the PDB already and which remain on hold.

“We revised the text to make it explicitly clear that, in our model, the transition between fully- and semi-swapped configuration is not a required gating step, but rather an equilibrium between the two configurations in the absence of physiologic stimuli.”

And later: “First, conversion between fully- and semi-swapped configurations does not constitute the gating step; rather, in the absence of stimuli, these configurations are in equilibrium, and chemical or thermal stimuli bias the equilibrium towards the semi-swapped configuration, which is especially pronounced for the avian channel. Second, both fully- and semi-swapped configurations can exist in closed or open states. While our data suggest that the fully swapped channel has a higher barrier to making the gating transition, the transition between fully and semi-swapped configurations is not the gating step per se. Thus, the HDX experiments are not measuring gating, but rather the kinetic barrier between fully- and semi-swapped configurations.”

This is a really important clarification, but it means that the paper does not per se describe a new phenomenon. Proteins are in conformational equilibrium between states and ligands (or physicochemical stimuli) can shift these equilibria.

And since the semi-swapped configuration is not part of the cold gating mechanism as clarified by the authors in the new version, the abstract is somewhat misleading:

“Here, we close this gap by combining cryo-EM with hydrogen-deuterium exchange mass spectrometry (HDX-MS) to elucidate a mechanism for cold-evoked activation of TRPM8. First, we visualize TRPM8 channels in cellular membranes, where bona fide menthol- and cold-evoked open states are captured. We identify a novel ‘semiswapped’ architecture in which interdigitation of channel subunits is substantially rearranged following repositioning of the S6 transmembrane helix and elements of the pore region.”

In a manuscript entitled “The structural basis of cold sensitivity”, it is not too far fetched to draw the conclusion that this newly identified structure represents the novel, relevant cold sensing state.

That said – if it is not cold activation/gating, what is the role of this newly identified structure?

General remarks:

Provision of HDX data in SI: It is in line with the common practice in the field to provide a table with the information of at least the most important peptides (not all of them), their sequences, exchange rate, coverage etc. Seeing the importance of this data for the current manuscript and interpretations therein this is not unreasonable request.

Apologies for not having made this clearer in the last round, however, the title of the paper is incredibly broad, but conclusions in the paper can really only be drawn for TRPM8, possibly even only for TRPM8 from a few species. While it seems possible/likely that the physical basis of cold activation involves heat capacity changes also in other channels, and that the structural transition from pi- to alpha-helical is preserved, this remains to be shown.

Transition between swapped and semi-swapped states: the new movie (Sup 1) is appreciated and will certainly help readers to underscore that the transition might be feasible without disassembly of the tetramer, however this would be even clearer if not shown as a space filling model.

Minor points:

Figure 4d: In the subfigure, data for dH, TdS and dG are presented for three temperature intervals, but the temperature used to calculate TdS is not mentioned, please indicate it e.g. in the figure legend.

Page 5: The reference to Extended Data Figure 3 seems misplaced, as this figure focuses on the pKa calculations of PmTRPM8 in the two configurations.

The numbering of residues in model 8 (structure files) differs from other models by 4 residues, maybe because it's the outer pore chimera? This could be clarified.

There is no description in the material and methods how the conformational morph between swapped and semi-swapped states was carried out.

Referee #3

(Remarks to the Author)

I co-reviewed this manuscript with one of the reviewers who provided the listed reports.

Referee #4

(Remarks to the Author)

This study by Choi et al. aims to understand the molecular basis of thermosensitivity in TRPM8, the ion channel responsible for cold perception and menthol sensitivity in mammals. The authors address a long-standing challenge in the field - elucidating the structural mechanism and thermodynamic changes that underlie temperature-dependent gating, through a combination of cryo-EM and hydrogen-deuterium exchange mass spectrometry (HDX-MS).

A key strength of the work lies in the study of TRPM8 channels within native cellular membranes, which allows the authors to capture authentic cold- and menthol-evoked open states. The identification of a substate with a "semi-swapped" subunit architecture represents an important structural insight, suggesting that intersubunit rearrangements and movements in the S6 helix and pore helices are central to activation. Complementary HDX-MS data, albeit not performed in native cellular membranes but detergent, highlight specific regions, particularly the pore and TRP helices, that undergo the most pronounced temperature-evoked changes. The comparative analysis between human TRPM8 and a menthol-sensitive but cold-insensitive avian orthologue serves to support the proposed gating mechanism and reinforces the physiological relevance of the structural observations. The authors also propose a free energy landscape model which strengthens understanding of the temperature-dependent gating in TRP channels more broadly.

With an eye on my primary area of expertise, the HDX-MS experiments themselves appear to have been well conducted, albeit with one important problem. Also, the analysis is mostly sound but I also think there are some issues wrt. the interpretation of bimodal mass envelopes that needs to be addressed.

Overall, I find the paper interesting as it offers a compelling structural and thermodynamic explanation for TRPM8 cold activation, setting a new benchmark for mechanistic studies of thermosensitive ion channels. Its integrative approach of both Cryo-EM and HDX-MS provides a useful method blueprint for dissecting other temperature-sensitive membrane proteins.

Comments:

1. The authors should follow the HDX-MS community guidelines (Masson et al. 2019) and specifically include HDX Data tables so that the data is more accessible – and deposit the data in a database (e.g. PRIDE). Also, the authors need to clearly define their threshold for significant changes in HDX between states – based on a proper assessment of error. I can advise that the authors adopt a hybrid testing approach (Volcano plot) – see Hageman and Weis. Otherwise follow the guidelines in Masson et al. 2019.

2. Assessment of error in the HDX-MS experiments: The HDX-MS data shown appears to be the average of triplicate measurements conducted on two HDX-MS platforms using two different MS instrument in two labs with slightly different LC times. These are then per definition not replicate measurements – and should not be averaged nor used to assess error. This has to be addressed.

I understand there may be practical reasons for doing the experiments this way – and the fact that the authors state that two datasets show high agreement add confidence to the overall findings. But this is nonetheless not a suitable way of assessing error in the data presented. I can suggest some ways to mitigate this. The authors could record two more replicate data sets at UCSF and show results (and error) from the three replicate measurements recorded at UCSF. Or alternatively, and less labor intensive, show data for only one of the replicates in the manuscript proper (perhaps the third replicate at UCSF) and then in new independent experiments assess the error of this experiment (be performing replicate measurements of some time-points using the same HDX-MS setup at UCSF). The authors could use these to estimate error and calculate a global confidence threshold for significant changes etc. The authors are referred to the HDX-MS community guidelines (Masson et al. 2019).

3. About the bimodal mass envelopes in the HDX-MS data: The authors state that "the pore and TRP helices each showed clear bimodal mass envelopes that increased in mass over labeling time, suggesting the coexistence of two conformational populations..." and "suggesting that these two populations undergo interconversion on the HDX-MS time scale (i.e., minutes to hours)": this section and other sections discussing EX2 vs. EX1 kinetics needs to be revised somewhat as I do not agree with the interpretation in SI Fig. 3. To my eye, the data shown in SI Fig.3b is EX1 or mixed EX1 kinetics and not EX2. As the authors correctly point out, the intensity of the two mass envelopes change during the HDX timecourse, and this is a hallmark of EX1 kinetics, see for instance https://pubs.acs.org/doi/pdf/10.1016/j.jasms.2006.05.014?ref=article_openPDF I think the two examples of raw data shown in SI Fig. 3 appear to be EX1 or mixed EX1 kinetics - not EX2 kinetics. The authors write that interconversion of the conformational states occur on the min-hrs timescale of the HDX experiment – I agree that this is very likely true – and that would certainly be expected for most protein transitions between two distinct states. But that is not why EX1 kinetics would be observed. EX1 is observed if two conformational states with distinct HDX behavior exist in solution that interconvert with a very slow rate relative to the chemical exchange rate (i.e. $k_{cl} \ll k_{ch}$). Thus, it specifically denotes that k_{cl} is "unusually" slow. The authors should try to extrapolate k_{op} (where possible, for instance using HX-Express software) and see if k_{op} is correlated between different segments where EX1/mixed EX1 is observed. This could be of functional significance – see for instance Merkle et al. Science Advances 2018. If the authors wanted to further investigate the slow time conversion between two conformational states representing fully – and semi-swapped states then this could be done using a pulse-labeling HDX experiment.

Importantly, I think the interpretation of two conformational states representing fully – and semi-swapped states can still be reconciled with the observation of EX1 kinetics. However, I am aware that some of the assumptions made to quantify thermodynamic parameters from HDX-MS are then no longer valid.

4. The authors should clearly substantiate the need to perform the elaborate quantification of thermodynamic parameters from the HDX data – which makes several assumptions that one could question or argue. What extra value is precisely

achieved? Fig. 6 does not needs this I believe. I think the paper would benefit from a more rigorous discussion of the assumptions made in this analysis and what precise new insights are obtained. It is a little unclear to me at least.

Version 2:

Reviewer comments:

Referee #2

(Remarks to the Author)

The authors have answered all our questions.

Referee #3

(Remarks to the Author)

I co-reviewed this manuscript with one of the reviewers who provided the listed reports.

Referee #4

(Remarks to the Author)

The authors have made a rigorous attempt at addressing my main concerns.

Concerning comment 1: Thank you for the reply and improving the statistical analysis.

Concerning comment 2: I appreciate the time taken to conduct a limited additional experiment to provide a more accurate estimate of error in the HDX data, even if this is not performed for all the states. However, I am still sceptical that the authors should average data across the two platforms as it gives a wrong impression of their error, especially considering that they can observe differences in LC retention times - and thus presumably back-exchange values etc.

However, based on the good repeatability for the three experiments conducted at USCF (replicates 2-4 which are true technical replicates) I am now more confident that they would not reach significantly different "biological" conclusions if similar "true" technical replicate measurements were performed for all states.

Concerning comment 3: The authors do misunderstand my comments somewhat but provide sufficient additional explanations in my view. They are further commended for conducting the additional experiments as a function of pH. This adds more confidence to the interpretation of their HDX data.

We thank the reviewers for their time and effort in providing valuable comments and suggestions. We begin with a brief summary of major changes to the manuscript, followed by a point-by-point rebuttal.

- We made major revisions to Introduction. We now focus on introducing limitations of current thermodynamic and structural understanding of temperature sensing in heat or cold activated channels, with better articulation of the rationale of combining cryo-EM in membrane vesicles with HDX-MS to address mechanism of temperature sensing from both structural and thermodynamic perspectives. This revision also addresses the comment of Referee 2/3 that “*most importantly, what has been proposed so far as potential molecular mechanisms for temperature sensing?*”
- We revised the text to make it explicitly clear that, in our model, the transition between fully- and semi-swapped configuration is not a required gating step, but rather an equilibrium between the two configurations in the absence of physiologic stimuli.
- We include new data demonstrating that humanized avian TRPM8 (containing the pore loop of human TRPM8) is cold sensitive at physiologic pH, upholding our conclusion that high pH is modulating avian-specific interactions within the pore loop that normally trap this region to hinder cold activation.
- We have also added new vesicle structures of avian TRPM8 acquired in the presence of calcium. These results show that closed states with or without bound calcium are almost identical, which has led us to re-assess whether the calcium-bound closed structure is, indeed, desensitized. Based on these and other considerations, we revised the discussion regarding our proposed model of cold-evoked channel opening, most notably to clarify that the transition between fully- and semi-swapped states is not a gating step *per se*.
- In response to requests from referees 2/3, we now provide a summary of all structures included in this study, with comparisons to previously published structures and conditions of sample preparation (included as Supplementary Information).

We noticed that in a recent manuscript released in *bioRxiv* by Seok-Young Lee’s lab (<https://doi.org/10.1101/2025.09.09.675254>), the authors commented on our studies and raised several points in their discussion section disputing our findings. We do not cite this study in our manuscript, which would compel us to openly respond to their arguments by pointing out the flaws and mistakes in their study - which we would prefer to avoid given that their work is not yet peer reviewed and structures are currently unavailable. Nonetheless, we present our response below so that the referees of our manuscript are aware of our thoughts:

- Regarding the transition from the fully- to semi-swapped configuration: as we now clarify in this revision, the transition between fully- and semi-swapped configurations is not a required gating step. Without stimuli (menthol or cold) both fully- and semi-swapped configuration are in closed gate conformations. Applying stimuli does shift the equilibrium between the fully- and semi-swapped configurations, as articulated in our hypothetical energy landscape model (Fig. 6). This reflects our data indicating that the semi-swapped configuration has a lower energetic barrier towards the open state.

- About the pore lining residues: Lee's group introduced a cysteine mutation to confirm pore lining residues, which is a classic physiologic approach. However, in their study the Phe979Cys mutant channel is not functional, diminishing their argument that our gating model is incorrect. Moreover, there are other reasons why Lee's group did not capture our open structure, as discussed below.
- About the transition time between open to desensitized channel: we address a similar question below in response to comments from Referees 2/3.
- Other major issues of the Lee study that may lead to incorrect conclusions:
 - (1) As stated in their preprint, all of their structures lack a well resolved pore helix region, making it impossible for them to have identified channels in the semi-swapped configuration. In contrast, our vesicle preps provide the first well resolved transmembrane domain structures for TRPM8, making this identification possible.
 - (2) As pointed out from previous studies, PIP2 plays a critical role in TRPM8 gating. In all of the structures reported in Lee's manuscript, soluble diC8-PIP2 was used to mimic PIP2, as is commonly done. However, diC8-PIP2 has shorter hydrophobic tails and in our open structures of both human and avian channels, we show that the long tails of PIP2 insert into a hydrophobic cleft to stabilize the channel in the open state. Without these long tails, diC8-PIP2 cannot play the same functional role as PIP2, and the end structure is unlikely the same, which we have similarly demonstrated in a previous study regarding PIP2 modulation of TRPV1 (PMC11402599).
 - (3) To capture structures of the 'cold activated' channel, Lee's group used three agonistic cofactors to potentiate the channel (menthol, AITC, and diC8-PIP2) plus a mutation. Given the polymodal nature of TRPM8 (as well as other thermosensing TRP channels) this approach makes it impossible to delineate the mechanism of cold activation alone.

Since the referees may have seen this preprint, we feel that it is appropriate for us to preemptively discuss these matters here, but we would prefer not to address them directly in our manuscript and instead leave the debate until after their study is peer reviewed.

Furthermore, for our part, we are confident about our structural analysis. As such, we are in the process of depositing our entire raw cryo-EM dataset into the public database (EMPIAR) so that any reader can independently validate our structural findings by re-processing our cryo-EM data.

Finally, additions or changes to the revised manuscript are highlighted in **blue**.

Referee #1 (Remarks to the Author):

This is an outstanding, clearly argued study of a conducting conformation of the avian and human

TRPM8 'cold' temperature-gated channels. This conformation appears as a result of a new 'semi-swap' form in which the S6 helix is significantly rearranged, also altering the TRP helix, and the outer pore region. The authors are able to visualize this conformation by preparing samples with MacKinnon's recent vesicle-purification method, which avoids detergent solubilization, and they employ extensive experiments and thermodynamic analysis of hydrogen-deuterium exchange to convincingly buttress their conclusions. My comments are relatively minor, and recommend publication with minor modifications.

We thank Referee 1 for his/her encouragement and enthusiastic assessment of this study. Responses to specific comments are provided below.

1. Fig. 1 shows the paper's first view of novel semi-swap conformation of avian M8 involving S6 and the outer pore region from a previously known full-swap "desensitized" conformation. The text describes the main features of this conformation clearly and succinctly. Points:

a. Fig. 1 legend, or main text, should indicate conditions experienced by the vesicles before spreading on grid. Cold? Apo or menthol? The methods state that vesicles are kept on ice before grid preparation, so these are presumed to be a cold-adapted apo condition. The reader should be reminded of this key condition in the main body of the paper.

We appreciate this point and have now included this information in the figure legend. These details are also provided in the Methods section and Supplementary Information.

b. Fig. 1g: It's formally correct, but it is a bit confusing to claim that the conformational change into the semi-swap form widens the selectivity-region. The figure actually shows that it is a widening of the "upper constriction" around a.a. 905 from a quite wide-open hole to an even wider one, while leaving the "lower" constriction (around 965-969) unchanged, and the channel presumably still nonconducting. I would change the final phrase to "...Y905 at the selectivity filter, thereby further widening it".

We thank the Referee for bringing this potentially confusing statement to our attention. We have changed the text as per his/her suggestion to clarify this point.

2. p. 6: Some readers may be unfamiliar with H-D exchange and so will be confused by your use of simply "peptides" at the top of the page. It might prevent confusion to reword as "...the majority of the peptides derived from proteolysis following H-D exchange..." Also, a reference is needed after "3 kcal/mol" in the first paragraph.

We have added text to the Methods section to describe in detail the energetic calculations from HDX data, as requested by the Referee.

3. Fig. 2b legend: Define the "nne state".

We apologize for the typo; this should be “one state” as noted in panel **b** of the figure. We have also slightly revised the wording for clarity.

4. p. 24: Readers may be confused by the use of FFC-8, a detergent, in the preparation. You may want to reassure them that the 0.5 mM concentration used here is well below the ~3 mM CMC.

We thank the Referee for noting this. We have now added the CMC information to the relevant sentence in the Methods section.

5. Did the authors attempt to find out if the TRPM8 apo structure in vesicles pre-adapted to a warm temperature (34 °C, say) before fast freezing, would fail to show the semi-swap form, or a far lower prevalence of that form? Although not necessary to perform anew, if true, it would be an interesting detail to add if the authors had already done this.

This is an interesting suggestion. We have performed a similar experiment in which human TRPM8 in vesicles was warmed to 37°C for 5 ~ 15 minutes, then loaded onto grids in a vitrobot held at 4°C prior to plunge freezing. Indeed, with this new analysis we observe a clearer cold-evoked open state for human TRPM8. We see both fully- and semi-swapped configurations, but the open state is clearly modeled only in the former, presumably reflecting the energetic and structural heterogeneity of mammalian orthologues in the semi-swapped configuration, as noted in our model (Figure 6). We now included these data in Supplementary Information Table 1 and Extended Data Figure 3.

6. It might be interesting for general readers to speculate/discuss why avians, which have higher metabolic rates and body temperatures of 38–41 °C, might have different M8 cold sensitivity and why this might be adaptive.

This is an interesting neuroethological and evolutionary question. Birds are more cold-tolerant than mammals, but as homeotherms they must still defend their core body temperature. Thus, we would posit that birds ‘use’ TRPM8 less for nociception and more for thermoregulation / thermo-adaptation. This may require more subtle and temporally slower input from this channel. In any case, we are reluctant to include this speculation here as these physiologic differences may be more complex and involve attributes beyond the channel itself. Furthermore, Referees 2/3 has asked whether the diminished cold sensitivity of recombinant avian TRPM8 is physiologically relevant, and so we suspect that they would object to any such speculation. Nonetheless, we have added a brief paragraph in the Discussion section that proposes how our structures can account for species-specific cold sensitivity.

Minor: ‘remain elusive’ in the second paragraph is an overused phrase.

Per the suggestion of Referee 2/3, we have reworked the introduction to provide a better lead-in to this study, which no longer uses this phrase.

Referee #2 (Remarks to the Author):

How temperature is sensed on a molecular level is one of the most exciting but also one of the most enigmatic questions in modern biophysics. While the role of transient receptor potential (TRP) ion channels in hot and cold sensing has been well established, the underlying molecular details of how changes in temperature result in ion channel pore opening remain unclear. Over the years, different proposals highlighting the potential role of changes in lipid/membrane interactions, partial unfolding of channel regions or dedicated thermosensing elements have been brought forward. However, to date no unifying, or alas any satisfactory explanation for how any of the many “ThermoTRPs” works is available. Even more confounding (or awe-inspiring, depending on one’s perspective) is the observation that in many cases, small molecules evoke the same physiological responses as a temperature stimulus, e.g. capsaicin from chili peppers tasting “hot” or menthol resulting in a “cold” sensation. Although the penultimate outcome of a thermal and a chemical stimulus must be the same, i.e. a change in the opening probability of the responsive ion channel, it is not clear whether the underlying molecular pathways are identical or merely partially overlapping. Furthermore, it is well known that ThermoTRP channels from different animals (or their splice variants) can have different threshold temperatures.

In the current manuscript, Choi and Lin et al. set out to answer the enigma of cold sensing, using the cold sensor TRP melastatin 8 (TRPM8) and its activator menthol from mammalian and avian species. To this end, they combine cryo-electron microscopy in native membranes with hydrogen deuterium exchange mass spectrometry to carry out structural and thermodynamic analyses of temperature-driven changes in TRPM8 channels.

Overall, the paper covers a highly important and very exciting topic. The methodological combination is well suited to investigate the question of interest, but the descriptions may be difficult to follow for non-expert readers in places. While the authors present very interesting new structures and ideas, I remain somewhat unconvinced that the mystery of cold sensing is really solved. My major concern in this regard is that obtaining a “thermally activated” state required substantial “experimental contortions”. Strictly speaking, the authors assume a converging mechanism for menthol/cold activation in a native membrane environment, but do not show it. Rather, they require use of an avian channel at high pH in detergent to obtain the open semi-swapped state that they assume presents the structural basis of cold sensing. It thus remains unclear whether they really identified a generalizable mechanism of cold sensing for TRPM8 channels.

In addition, the HDX experiments that are the basis for the thermodynamic model of cold sensing were carried out with detergent-solubilized protein. This seems at odds with the strong statements by the authors that the native environment is crucial to understand thermosensing.

We thank Referees 2/3 for their positive comments as well as general concerns about clarity and the extent to which ‘the mystery of cold sensing is really solved.’ Before addressing their specific criticisms, we offer the following responses to these more general concerns.

First, we appreciate the referees' suggestion that we improve clarity for non-expert readers. We have re-worked the introduction (also please see below) to better convey the motivation and logic for our approach. We've also modified the main text to better describe and interpret the results from HDX-MS experiments, which is perhaps most challenging for non-specialists.

Second, we would argue that we have obtained and analyzed a "thermally activated" state that requires among the least "experimental contortions" reported to-date. Thus, in our cold-activated cryo-EM structures or HDX analyses of avian and mammalian TRPM8, we have not introduced mutations or facilitated cold-evoked gating by including agonists or other pharmacological modifiers. The exception is in using high pH to render the bird channel cold sensitive (which mostly tests predictions gleaned from our apo semi-swapped structure); we then examine the 'humanized' avian channel to confirm the mechanism whereby high pH enhances cold sensitivity of the wild-type bird channel, which is otherwise cold insensitive. Importantly, our analysis of the cold-activated human channel does not involve mutagenesis or pharmacological manipulation of any kind and is consistent with structures and mechanisms gleaned from the avian orthologue. Indeed, our cold-activated human TRPM8 structures are now substantially improved with additional data, further elucidating a cold-evoked open state in the absence of any other stimuli or manipulations.

Third, the referees noted that the HDX-MS experiments are carried out with detergent-solubilized protein, which seems at odds with our emphasis on the importance of a more native membrane environment in understanding mechanisms of thermosensing. We agree that our introduction, as originally written, inadvertently created a false dichotomy – mostly reflecting the way in which the project developed. Consequently, we have re-written the intro to better convey the power and logic of our two-pronged approach. At the same time, it is important to note that cell-derived vesicles have allowed us to visualize the novel semi-swapped conformation not previously observed with detergent-solubilized protein, making this an important technical advance for the structural aspect of the study. This does not mean that the semi-swapped conformation does not exist in detergent (indeed, our HDX measurements indicate that fully- and semi-swapped channels are present and interconvert), but that it is not readily captured by cryo-EM outside of a membrane environment. Furthermore, in all published TRPM8 structures determined from detergent, a large portion of particles were discarded from the final structure, and almost all of these structures have a poorly resolved pore region, which is likely reflective of a mixture of fully- and semi-swapped configurations and conformational intermediates, especially in mammalian orthologues in which the outer pore region is more dynamic compared to avian TRPM8.

Main points:

In the introduction, I am missing a few explanations about what is known about cold sensing of avian TRPM8, and what are the threshold temperatures compared to the mammalian channel. Is cold/menthol sensitivity widely present across bird TRPM8s? Are the residues for menthol binding conserved? Was there any particular reason for choosing P. major TRPM8 other than high homogeneity observed in previous studies? And most importantly: What has been

proposed so far as potential molecular mechanisms for temperature sensing? It would be good to have an overview of currently debated ideas.

We agree that our introductory section did not provide sufficient background pertaining to potential molecular mechanisms for temperature sensing, and we thank the referees for bringing this to our attention. In addition to including more information along these lines, we have refocused the introduction to more clearly enunciate our motivation for combining cryo-EM in membrane vesicles with HDX-MS to close existing gaps in delineating temperature sensing mechanisms from both structural and thermodynamic perspectives. Given the limited space for an introductory section, we now focus on the most important general points raised by referees 2/3. Some of the specific questions noted above - such as whether cold/menthol sensitivity is widely present across bird TRPM8s (bird channels examined are menthol sensitive but have diminished cold sensitivity) or if residues for menthol binding are conserved (they are) – are addressed in detail in cited literature.

The authors rightfully state that one possible concern about determining the structural basis of thermosensing could be the loss of relevant states by extracting protein from the membrane or generally studying it under non-native conditions. However, since they use a heterologous expression system, i.e. a human cell line to express an avian channel, they may also miss crucial binding partners or lipids. Just to clarify, I am not arguing against the use of heterologous expression systems, but the use of the word “native” under these particular circumstances should probably be used very carefully and differences in the lipid composition of avian and mammalian cell membranes should be described up-front as far as they are known. At the very least, it should be clearly stated in the main text (‘Cells expressing the *Parus major* channel...’ (p. 4, 2nd sentence)), that the expression system is a HEK293-based system.

We thank the referees for noting this issue. We do not know whether or how the cellular environment of bird sensory neurons differs from that of HEK293 cells and thus we really can’t say much about this. But perhaps the most relevant point is that we are comparing avian to mammalian TRPM8 in the same cell type, so whether the difference reflects true differential ‘native’ properties is perhaps not critical to this study as we are simply exploiting species-specific functional differences in an identical cellular environment to probe gating mechanisms. Nonetheless, we appreciate the referees’ cautionary note on the use of the term ‘native’, which we have eliminated in favor of ‘cell membrane environment.’

Why do the authors presume it has been so difficult to reconstitute TRP(M8) channels into lipid-based systems, compared to other membrane protein families? Across all available structures and TRP channel subfamilies (to the best of my knowledge), it seems that the transmembrane regions are not significantly more dynamic than those of other channels/membrane proteins, including transporters which can undergo major conformational changes, but generally there are not many studies available for TRP channels in liposomes? However, it also seems from the authors own observations (Fig. S1), that the TRPM8 transmembrane domain is much less well-defined than the cytosolic region. Is this difficulty with reconstitution therefore generally more pronounced for thermoTRPs than for other TRP channel family members or a specialty of

TRPM8? If this is a cold-sensor specificity, how do TRPA1 structures fare in comparison, which can act as a cold or heat sensor depending on the host organism?

With regard to this reconstitution issue, our strategy was not guided by a presumption, but rather by our extensive experience in being able to reconstitute some TRP channels (TRPV1, TRPV5, and TRPM4), but not others (TRPA1 and TRPM8) into lipid environments. Whether a detergent-extracted membrane protein can be reconstituted back into a lipid bilayer depends, in large part, on the type of detergent that can be used for extraction, rather than the architecture of the transmembrane domain. Thus, the short answer to the referees' question is that the ability to reconstitute thermosensitive TRP channels (e.g., TRPV1 versus TRPM8) into a lipid bilayer is channel specific and depends on detergent compatibility.

Fig. 1e,f: Experimental densities for side chains should be shown along structural cartoons; how reliable is the sidechain conformation?

We appreciate this suggestion and have now added panels showing relevant densities in the Supplementary Information section.

Fig. 2 and p. 5, quantification of fully- vs semi-swapped configuration: Can you actually state the respective % of the populations and the details of the incubation (i.e. temperature, conc. of menthol, length of incubation) in the text or the figure legend?

We now include this information in respective figure legends.

Are shorter incubation times or those with sub-stoichiometric menthol concurrent with less populated semi-swapped states? Do you see full occupancy for the menthol density?

In these experiments, we have used a saturating dose of menthol (1 mM) that is well above the EC_{50} . Indeed, all substates that we observe in the presence of menthol show pronounced menthol density, attesting to the fact that we are operating under saturating conditions. While computationally identifying particles with sub-stoichiometric binding would be quite difficult given the small size of this ligand, under these conditions we expect that the percentage of unoccupied particles is rather small.

And in addition, does the significant presence of at least one additional state in addition to semi-/fully-swapped mean that assumption of a two-state model is too simple? This is also important for the interpretation of the HDX data; how do you interpret HDX if >60% of particles in cryo-EM are neither classified as fully-swapped or semi-swapped? (And what exactly is the definition of "undetermined" here: resolution too low? differences between subunits?)

With regard to our proposed two-state model, our HDX analysis clearly supports the existence of more than one stable, low energy state because the profile cannot be fit with one binomial function associated with deuterium uptake. The data can be readily fit by two binomial functions, which we interpret as representing two stable, low energy states corresponding to fully- and semi-swapped configurations. However, the HDX spectra can also be fit to accommodate additional states representing meta-stable intermediates, which likely correspond to the class of 'undetermined' particles in which the resolution of key regions that distinguish fully- from semi-swapped

configurations are insufficient to unambiguously categorize these particles. We now specify this criterion in the legend to Figure 2.

Pages 5-6: A brief section in the main text should be added to describe how exactly the H/D-exchange rates were converted into free energy and enthalpy differences. Specifically, the authors assume “minimal heat capacity changes” (p. 8). I suggest to clarify, if they minimize heat capacity changes in their fit or if they set heat capacity changes to zero, possibly by including the equations by which they convert HDX rates into free energy and enthalpy changes in the methods section. A table with an overview about the detected peptides and their exchange rates under different conditions should be added to the Supplement.

We thank the referees for noting this issue. We completely revised this section of the main text to better articulate our quantification strategy. We have also added information to the Methods section to clarify how we convert HDX rate to free energy and enthalpy change, and to specify that ΔC_p is assumed to be zero only for each temperature segment when using van't Hoff analysis. Information about detected peptides and their exchange rates under different conditions would be way too large to include in a table, but all of this information has been deposited as raw and processed data in the ProteomeXchange consortium, as noted in the data availability statement.

Page 5: While the authors first observe the novel semi-swapped configuration in cell membranes, they perform the HDX-MS experiments with detergent-solubilized channels. Could the authors clarify whether HDX-MS experiments were not possible or not successful in the cell membrane context?

In general, performing HDX-MS of membrane proteins in reconstituted lipid environments is exceedingly challenging and we have not attempted this since TRPM8 cannot be reconstituted into such preparations. Moreover, performing HDX-MS with polytopic membrane proteins in cell-derived vesicles would be even more challenging given the low protein:lipid ratio. Indeed, such an experiment has not, to our knowledge, been previously reported.

Furthermore, the number of replicates for some of the HDX data ($n=1$) is concerning. I am aware that it can be tremendously difficult to properly replicate HDX-MS data, but seeing that there are enormously important conclusions drawn from these data sets, this concern should be addressed.

We appreciate this suggestion and have carried out two additional biological replicates of all reported conditions ($n = 3$). HDX data presented in Fig. 3 and 4 now represent the average of these data sets. Replicates and statistics are also updated in Extended Data Table 2.

Another major question concerns the kinetics: the time scale of HDX for interconversion between swapped and semi-swapped is in the range of (min – hr), compared to a (presumably sub-ms) time scale required for cold sensing; how does that fit together? If exposure to cold

would trigger a structural interconversion from fully-swapped to semi-swapped, would this be too slow as a sensing mechanism?

The referees raise a question that has compelled us to further clarify a couple of important points in the revised manuscript. First, conversion between fully- and semi-swapped configurations does not constitute the gating step; rather, in the absence of stimuli, these configurations are in equilibrium, and chemical or thermal stimuli bias the equilibrium towards the semi-swapped configuration, which is especially pronounced for the avian channel. Second, both fully- and semi-swapped configurations can exist in closed or open states. While our data suggest that the fully-swapped channel has a higher barrier to making the gating transition, the transition between fully- and semi-swapped configurations is not the gating step *per se*. Thus, the HDX experiments are not measuring gating, but rather the kinetic barrier between fully- and semi-swapped configurations.

What are the pathways the authors propose for the transition between fully-swapped and semi-swapped configurations?? MD simulations may be quite useful here if they can cover the necessary time scales. At the very least the authors should make an effort to describe the potential structural conversions required globally to interconvert swapped and semi-swapped states.

We appreciate this suggestion, but we believe that MD would be unfeasible as the time scale is too long. In any case, the topological comparison between fully- and semi-swapped states shows that the interconversion does not require tetramer disassembly or global unfolding, which we state in the manuscript. While the exact path of conversion remains unclear from structural analysis, we now include a supplementary movie morphing between fully- and semi-swapped PDB models to demonstrate that this conversion can occur without tetramer disassembly.

Do TRPM8 channels desensitize with cold/menthol? Do the authors assume that the desensitized and the closed states are structurally identical? In the Extended Data Table, there is a mention of a desensitized state; maybe the authors can explain this in more detail.

Yes, TRPM8 desensitizes after activation by cold or menthol in a calcium-dependent manner (PMID31488702). As noted above, we have added new vesicle-derived structures of avian TRPM8 in the presence of calcium and human TRPM8 in the absence of calcium. These results show that closed states with or without bound calcium are almost identical, which has led us to re-assess whether the previously reported calcium-bound closed structure is, indeed, desensitized. We now address this issue in the revised text.

The thermodynamic interpretation of HDX requires the EX2 exchange regime. Are these conditions uniformly met?

All peptides identified for TRPM8 exhibited exchange via EX2 kinetics, supported by the continuous shifts in the single isotopic envelopes toward higher m/z with increasing exchange time. We observed bimodal isotopic envelopes for peptides covering the pore and TRP helices, as shown in Fig2. These peptides, however, did not show signatures of EX1 kinetics resulted from

the rate of reforming the hydrogen bond (k_{close}) being much slower than k_{chem} . In EX1 kinetics, the amplitude of the lighter envelope decreases over exchanging time with a commensurate increase in the heavier amplitude, without a shift in m/z. In contrast, bimodal peptides in our study showed both envelopes shifting to higher m/z over exchange time, with one exchanging more slowly than the other. These spectra reflect the presence of two slowly interconverting, structurally distinct populations, each exchanging via EX2 kinetics.

The uptake kinetics should be shown as a supplementary figure across all measured time points and replicates, at least for a more widely distributed selection of peptides across the channel structure, and also including for peptides which show no changes between states as well as examples of isotopic distribution to differentiate between EX1 and EX2. Furthermore, a table with the detected peptides should be provided.

We appreciate the suggestion to provide all peptides and uptake kinetics. However, providing all the uptake kinetics for over 2000 peptides under all conditions and replicates will result in way too large of a supplementary figure. All of this information (including identified peptides, uptake kinetics, and isotopic distributions) has been deposited as raw and processed data in the ProteomeXchange consortium as noted in the data availability statement. We have also added a figure in Supplemental Information to provide examples of isotopic distributions supporting EX2 kinetics for both peptides exhibiting unimodality and bimodality.

Please also map HDX onto a structural model of TRPM8 for a global overview of the effects of menthol and temperature (and comparison of human vs avian channel). This would be a useful supplementary figure.

We appreciate this helpful suggestion and now include this information in Extended Data Figures 6 and 8.

Fig. 3: Using HDX-MS, the authors show that menthol binding stabilizes the TRP helix. It seems (Ext. Data Table 2) that the data shown in Fig. 3d stem from two biological replicates, so there should be error bars available?

Thank you for noting this omission. Error bars representing the standard deviation of triplicates are now included.

What about other channel regions (see Fig. 3c, around residue 875) that seem to differ quite substantially between human and avian TRPM8?

The apparent positive $\Delta\Delta G$ in human TRPM8 around residue 875 was a calculation error in our initial submission, in which the apo data were missing %D information for one time point, resulting in an artificially larger difference in %D when compared to the %D of menthol-bound state. We acknowledge this error and appreciate the Referee for pointing this out. We have checked all related energetic calculations to account for missing data points. Fig. 3c has now been updated.

The energy landscape in Fig. 6 has both fully-swapped and semi-swapped closed states at similar energy levels for both human and avian TRPM8, but that does not seem to match the observations by HDX??

Thank you for pointing this out. We agree and have revised the figure accordingly.

Page 8 bottom, pH sensitivity: The authors propose a pH sensitivity pocket created by Y905, R875, E896, D907, D908. What is the proposed molecular mechanism of pH sensitivity? The pKa values of these amino acids are either well above or below pH 7 (>10 for Arg/Tyr, <5 for E/D). Typically, buried salt bridges result in pKa's even further shifted from physiological pH and to allow protonation of E and D, the pKa values of these acidic residues must be shifted significantly towards neutrality? Is there any evidence for this?

Calculated pKa values for relevant residues are, indeed, substantially different from standard values for individual amino acids in solution, reflecting their local environment. We now illustrate this in Extended Data Fig. X, which shows that pKa's of residues within this region shift from their standard values towards neutrality in the semi-swapped state (PDB: 9P91). This is especially evident for D907 (calculated pKa of 8.16 versus ~ 3.8 for this amino acid in solution), and to a lesser extent for D908 (5.43). Moreover, these values differ substantially from those calculated for the fully-swapped (PDB: 9P90) configuration (5.26 and 3.92, respectively). In addition to providing this information in extended data, we also note this in the main text.

Page 9, "Our calcium imaging experiments indicate that breaking the species specific Y905 interface potentiates cold-evoked responses of the avian channel": What exactly do the authors mean by "breaking"? What constitutes the break, i.e. the mutation, Ca binding?

We are referring to destabilizing of this interface either by changing extracellular pH or through mutagenesis. We now use the term 'destabilizing' for clarity.

Page 12 and Fig. 6: The "cold semi-swap model" the authors present seems to be independent of the cytoplasmic regions? In other TRP channels, e.g. TRPV3, it has been proposed that the cytosolic regions are quite relevant for thermosensing due to conformational changes. Is there any indication here that the MHRs are also important?

The nature of TRP channel cytoplasmic domains is exceedingly variable and for many subtypes (including TRPM8) of unknown function. In any case, we have not detected notable structural changes in the MHR domain in response to menthol or cold. We do see cold-induced enthalpic stabilization distributed across the MHR domain for human TRPM8, however the magnitude is much smaller than what we observe for the pore helix.

Furthermore, the "cold-activated" structure determined here (avian species, high pH, 4 °C) is structurally different from that of mouse TRPM8 with a cooling compound, which the authors state themselves (p. 9). So naively I have to ask: which of these constitutes a true "cold-activated" state? Is there more than one cold activated state? Or do chemically and thermally activated states differ? Is the observed state in the present manuscript then potentially a pH-mediated activated open state?

In this study, with newly included data, we actually show that the channel gate is in the same configuration (formed by a phenylalanine cage) in cold- or menthol-activated structures (in avian and human orthologues in fully- or semi-swapped configurations). We believe that the previously described mouse TRPM8 open state is not actually a physiologically relevant cold-evoked open state because it was captured in the presence of various agonists and soluble di-C8-PIP₂ (as discussed above). As for whether our avian cold open state represents a pH-mediated state, we have added new data describing a humanized avian channel captured in cold and pH 7.4 that shows the same open configuration, validating it as a *bona fide* open state. We also now see an open human channel at 4°C without any cooling agents. This was achieved through addition of new data in which vesicles containing human TRPM8 were first warmed to 37°C for 5-15 min, then loaded onto grids in a vitrobot held at 4°C prior to plunge freezing. Indeed, with these new data we observe a clearer cold-evoked open state for human TRPM8. We see both fully- and semi-swapped configurations, but the open state is clearly modeled only in the former, presumably reflecting the energetic and structural heterogeneity of mammalian orthologues in the semi-swapped configuration, as noted in our model (Figure 6). We now included these data in the Supplementary Information Table 1 and Extended Data Figure 3.

Together with the analysis of avian and chimeric channels, these new data provide clear support for the structure of a cold-activated state.

Furthermore, the authors should at least discuss the available data for TRPV6 and TRPV3 by the Sobolevsky and Scheuring labs that show that the transmembrane interfaces of TRP channels are relatively pliable, which may be exactly what is also observed here for TRPM8. In the case of TRPV6 for instance, introduction of a single point mutant can lead to a structural rearrangement from a swapped to a non-swapped state (PMID: 28878326, PMID: 27296226). In TRPV3, it seems that entire subunits can be relatively easily exchanged (PMID: 37648856).

Yes, we thought about noting these studies, but then decided that it might mislead readers into thinking that we are suggesting a mechanism involving subunit disassembly, which is not in principle required to achieve a transition between fully- and semi-swapped states. Furthermore, the TRPV6 example is likely a consequence of biogenesis, rather than interconversion between non- and fully-swapped states. In any case, in response to this comment, we have added a movie that shows a morph between fully- and semi-swapped PDB models to illustrate that disassembly is not required for this transition.

Across all figures, pointing out the PDB codes for the structures shown would really help with readability. Throughout the text, it would really help to always clearly state which TRPM8 (which species) you talk about at any given point.

We appreciate this point and now refer to PDB codes and species wherever relevant and helpful.

Minor points:

Introduction: "In the case of TRPM8, structural heterogeneity may be exacerbated by the

channel's cold sensitivity, resulting in a range of conformational substates that are too transient or numerous to register as major structural subclasses during cryo-EM analysis (Extended Data Fig. 1).": It would be helpful to have a table/figure comparing all available TRPM8 structures and the respective resolutions under the different experimental conditions, since it is also possible that structural heterogeneity stems from the method of preparation.

We now include a table in Supplementary Information summarizing all structures presented in this study.

Page 4 (Description of the swapped architecture): A brief outline of the general architecture of TRP channel transmembrane domains for the uninitiated reader (here or in the introduction) would be very helpful. For instance, it presumably remains completely unclear to the non-ion channel reader that transmembrane helix S6 constitutes a pore forming helix at this point in the manuscript.

Page 4: It may be a bit confusing to begin with "From these data, we obtained high resolution (3 – 3.5Å) structures, including the previously described closed and desensitized states 18,20,27 (Extended Data Fig. 3), as well as a novel substate whose most notable feature is a distinct domain swap." when later it is explained that the "fully swapped state" is the canonical architecture? Fig. S1a: what are the 3 non-TRPM8 references used?

These two points are well taken. We now include a brief description of the overall channel architecture to orient the reader and clarify our use of terms.

The 3 non-TRPM8 references refer to arbitrary noise references that facilitate removal of non-TRPM8 particles, which is a standard procedure. We now refer to these as 'junk classes.'

Figs S3/S4: It would be really helpful to add the PDB/EMDB codes to these figure to more easily compare to the Ext. Data Table 1. Of note, there is no mention of the "semi-swapped" state in hsTRPM8 in the table? There is also no mention of the V915Y mutant in the Table so it seems there are only WT structures listed? Open/closed state is not indicated for all channels. Finally (this is nitpicky, but may help to avoid confusion), extended table 1 states that structures are "in membranes", whereas throughout the rest of the figures "in vesicles" is used.

PDB codes have been added, as requested.

A supplementary figure that has all structures, their organism, open/closed state, ligand, temperature state (as well as pH), and PDB code side by side would go a long way to improve the readability of this manuscript.

This information is now included in Supplementary Information.

Fig. 2b: Spelling mistake 'nne state' should read 'one state'

Thank you – typo corrected.

Fig. 2c: The data seem to have different y-scales for the different time points; why not use a uniform scaling?

We're unsure what this refers to, but perhaps there is a misunderstanding about the annotation as this panel has no labeled y-scale. The 4, 40, and 400 minutes refers to deuterium labeling time, not scale, as noted in the legend.

Ext. Data Table 2 (HDX): This is very hard to read. Instead of writing "same as XXX", why not just put the actual conditions (one can still color code or spell out that samples were treated similarly)?

We have revised the table accordingly.

Page 5: The second section of the main text is called "Agonists modulate dynamic equilibrium between states". The authors only show modulation for one agonist (menthol), which does not justify to speak of "agonists" in plural in a strict sense.

Agreed. We have changed this subtitle to read as, "Menthol modulates dynamic..."

Page 5 bottom: When the effects of menthol on the population ratio are described and Fig. 2a is introduced it should be clearly stated that this is in reference to the avian channel.

We have now stated as such.

Page 5: Reference 31 refers to a seminal NMR paper on hydrogen exchange, but its not a suitable reference for HDX-MS. In addition to this reference to HDX, the authors should include a suitable reference to an HDX-MS paper that introduces the mapping to specific residues with the MS approach.

We have added a suitable review by Hamuro (PMID: 34749499).

Fig. 2: The use of the term "population" seems not to be consistent in the presentation of the HDX-MS data. In the main text, the populations are identified with configurations (semi-swapped vs. fully-swapped), while in Fig. 2d, "population" refers to the weights in the bimodal distribution/ the fraction of the two distributions (as correctly named in e again). For a very slow exchange, it would appear to be possible to assign the population of peptides to a certain dynamic state, but in the "slow" exchange scenario (2b right side) this seems to be misleading to me.

We do use the term 'population' in two distinct ways: one refers to particle distribution, whereas the other refers to peptide mass distribution. In the revised manuscript, we now label Fig. 2d as 'fraction' rather than '% population.'

As described in the main text and Methods section (and in response to referee 1), populations are derived from fitting the HDX-MS data with two binomial functions. While they overlap, they are clearly separable by computational methods.

Page 6, top, “Such features of bimodality in deuterated mass spectra suggest the coexistence of two conformational populations, with their energetics differing by at least 3 kcal/mol.”: Should read “..., with their free energies of unfolding (ΔG) differing by at least ...”.

We have changed the language, as recommended.

Page 7, bottom and Fig. 3, “We resolved structures of human and avian TRPM8 channels in vesicles in the presence of menthol.”: Do these structures correspond to the swapped, semi-swapped, or both conformations?

In the presence of menthol, we obtained high resolution structures in semi- and fully-swapped configurations for avian TRPM8. For the human channel, the fully-swapped configuration was of highest resolution. This information is now included in Supplementary Information, as referenced in the text.

Page 7, “However, our structural analyses in vesicles, as well as HDX-MS measurements, show that the human channel exhibits much greater energetic heterogeneity than previously appreciated.”: What exactly do you mean by “energetic heterogeneity”?

What we meant to convey is simply that the human channel shows much greater structural heterogeneity compared to the avian channel, reflecting a greater percentage of unclassified particles and relative difficulty in obtaining high resolution structures for mammalian TRPM8. Moreover, the HDX exchange rate is more temperature dependent for the human orthologue, especially within the pore helix region, all indicative of greater structural and energetic heterogeneity. However, we agree that as written, the term “energetic heterogeneity” is vague and thus we now just used ‘heterogeneity’ in this context.

Fig. 3: The authors present convincing evidence for ligand density in avian and human channels, into which they dock menthol. However, the density itself does not seem to support a clear positioning of menthol in the binding pocket. The authors should explicitly state the model character in the legend to Fig. 3.

We apologize for this poor rendering of the menthol density. We now illustrate this in a better way to show that the density actually fits a menthol molecule quite well.

Page 5 subtitle and title Fig. 5: What protein is used: avian high pH or avian Y mutant at high pH? This should be clearly stated in subtitle and figure title.

As noted in the Fig 5a legend, this structure was determined for pmTRPM8 at 4°C, pH 9. We now further note that this refers to wild-type pmTRPM8.

Fig. 4c: Please present a phylogenetically deeper alignment for avian/mammalian TrpM8 (e.g. in supplement in extension to Fig. 4c). Are there bird TRPM8 with the human valine residue or vice versa? How broadly can the term avian/mammalian TrpM8 be used based on this amino acid difference?

There is truly a dichotomy between birds and mammals that extends to all avian and mammalian species whose sequences we have examined, consistent with the study we reference from Yang et. al. (PMID 32220960). We now include a phylogenetic representation summarizing amino acids in this region for 450 Sauropsida (avian) species and 390 mammalian species (Figure 4e).

Fig. 6a shows a fully-swapped open conformation for the mammalian channel and a semi-swapped open conformation, but it seems that only the fully-swapped open conformation was observed/resolved by EM (p. 10, bottom). Is there structural evidence for the semi-swapped open conformation? What would be the function of two different open conformations? Functionally redundant? Semi-swapped not functionally important in mammalian systems?

We thank the referees for asking these questions, which have prompted us to more clearly address these issues in our Discussion section. We can see a semi-swapped conformation for human TRPM8 by cryo-EM, but the resolution is not high enough to ascertain the gate configuration, presumably reflecting the intrinsic heterogeneity of the mammalian channel. In our revised Discussion section, we postulate that mammalian channels can open from either fully- or semi-swapped states, but that opening from the latter is more likely based on observed π vs α S6 configurations. In contrast, the bird channel can only open in response to cold from the fully-swapped state at physiologic pH, thereby limiting its cold sensitivity. We have revised Figure 6 to reflect these points.

Referee #3 (Remarks to the Author):

I co-reviewed this manuscript with one of the reviewers who provided the listed reports.

Once again, we thank all referees for their time and efforts helping us to improve our manuscript. To make it easier for the Referees, new edits of this round of the revision are in blue texts. In the following, we provide our point-to-point response to comments of all Referees.

Referee #1 (Remarks to the Author):

The authors have addressed all my suggestions for changes.

We thank the referee for his/her efforts in helping us improve our manuscript.

Referee #2 (Remarks to the Author):

We would like to thank the authors for their patience and the extensive replies to our comments. This is very much appreciated. Readability has been much improved in the revised manuscript. The author's point that the presented cold-activated TRPM8 structures presented were obtained with the least possible "experimental contortions" is well taken. We agree that especially the use of truly native constructs without mutations is very important and necessary to elucidate the mechanism of cold sensing. Their point about the importance of the structure determination in native membranes and the presumption that the semi-swapped state has presumably been missed in detergent-preparations of TRPM8 due to discarded particles seems very intuitive. Finally, we do wish to clarify that our previous comment about the HDX measurements in detergent were not meant as a major criticism, as we are well aware of the technical challenges of HDX on membrane proteins and thus appreciate the major effort of the authors in the current study. Accordingly, the increase in replicate number for these samples in the revised manuscript should also be positively mentioned.

We thank the referees for acknowledging our efforts in addressing these important issues.

Overall, the paper is a very interesting, technically challenging study and introduces important methods that have been applied to other membrane proteins (cryoEM in cell-derived vesicles and HDX) to the TRP channel field. Without a doubt, this new combination of methods sheds light onto new and unexpected aspects of the structural biology of TRPM8 such as the new semi-swapped configuration, the modulation of dynamics by menthol, the structural impact of menthol binding, the differential thermodynamics in human and avian orthologues and the molecular origin thereof.

We appreciate the referees' acknowledgement of these important conclusions derived from our experimental approach.

On the other hand, the conclusions (dynamic equilibria, structural changes in response to a stimulus) result in an expected outcome, i.e. opening of the channel upon cold exposure, but that does not yet explain what the “structural basis of cold sensitivity” is or where/how it is encoded in the protein.

In summary, also with the revised manuscript, it is still difficult to gauge what the authors think the structural basis of cold sensitivity actually is, even ‘just’ for TRPM8. If you had to say it in 1-2 sentences, what is it? A clear, concise statement would go a long way.

We appreciate this important suggestion, which has made us drill down and be more specific about our structural and energetic model for cold-evoked channel activation. We now express this clearly through additional wording in the abstract (‘Specifically, cold-evoked stabilization of the outer pore region repositions the pore lining S6 transmembrane helix while enabling binding of a regulatory lipid to stabilize the open channel.’), as well as inclusion of an new panel to Fig. 6 to accompany the energy landscape, thereby providing a structural scheme explaining how we think cold activates TRPM8.

Please see below for more specific replies and comments to the revised manuscript and rebuttal letter (authors’ statements in quotation marks):

From new introduction “Indeed, of the numerous TRPM8 conformations in apo or ligand-bound states reported thus far 22,24-28, none convincingly represent a purely cold-evoked open state or provide structural insights into thermal gating mechanisms, which may reflect the fact that they have all been determined with detergent-solubilized material.”

“We now focus on introducing limitations of current thermodynamic and structural understanding of temperature sensing in heat or cold activated channels, with better articulation of the rationale of combining cryo-EM in membrane vesicles with HDX-MS to address mechanism of temperature sensing from both structural and thermodynamic perspectives.”

Thank you very much for including this in the revised manuscript.

We thank the referees for appreciating the revised introduction.

We cannot and do not want to judge on the quality or conclusions of various pre-prints, and of course, it remains up the authors (and editors) whether to take preprints into account at all, but since the authors themselves brought up the paper by Lee and colleagues, we would like to at least alert you to the fact that previously the Nimigean

lab also proposed a cold sensing mechanism through salt bridges and lipids (<https://www.biorxiv.org/content/10.1101/2025.06.03.657524v1>).

We only mentioned the Lee lab preprint in our rebuttal because it explicitly disagrees with some of our key experimental findings and we wanted to be preemptive in articulating for the referees and editor why their arguments about our findings are incorrect. We did not intend to cite this preprint. Thus, while we appreciate the referees' noting of another relevant preprint, we feel that it is only fair to also not cite the BioRxiv study from the Nimigean lab.

Unfortunately we cannot properly judge whether the newly added human TRPM8 cold-activated structure in the current manuscript now convincingly presents an open state as this model was not included in the uploaded files by the authors. This might be the result of a misunderstanding: as of now, we do not have access to the structures that were uploaded to the PDB already and which remain on hold.

As per the editor's instructions, we uploaded all new structures and maps to the link provided by the editorial office for sharing with the referees (upload was completed by us on November 3). We apologize for any miscommunication.

"We revised the text to make it explicitly clear that, in our model, the transition between fully- and semi-swapped configuration is not a required gating step, but rather an equilibrium between the two configurations in the absence of physiologic stimuli." And later: "First, conversion between fully- and semi-swapped configurations does not constitute the gating step; rather, in the absence of stimuli, these configurations are in equilibrium, and chemical or thermal stimuli bias the equilibrium towards the semi-swapped configuration, which is especially pronounced for the avian channel. Second, both fully- and semiswapped configurations can exist in closed or open states. While our data suggest that the fully swapped channel has a higher barrier to making the gating transition, the transition between fully and semi-swapped configurations is not the gating step per se. Thus, the HDX experiments are not measuring gating, but rather the kinetic barrier between fully- and semi-swapped configurations."

This is a really important clarification, but it means that the paper does not per se describe a new phenomenon. Proteins are in conformational equilibrium between states and ligands (or physicochemical stimuli) can shift these equilibria.

We respectfully disagree with this comment. We do report interesting new phenomena. First, we identify a new and unexpected semi-swapped configuration that exists in equilibrium with the canonical fully-swapped state. Second, existence of these two states explains why the avian channel is only weakly cold sensitive, demonstrating the

physiologic relevance of this phenomenon. Third, we reveal two parallel pathways for cold-evoked activation, with the pathway involving the semi-swapped configuration being the likely more energetically favorable path in mammals. In the revised manuscript, these points are enunciated more clearly.

And since the semi-swapped configuration is not part of the cold gating mechanism as clarified by the authors in the new version, the abstract is somewhat misleading:

“Here, we close this gap by combining cryo-EM with hydrogen-deuterium exchange mass spectrometry (HDX-MS) to elucidate a mechanism for cold-evoked activation of TRPM8. First, we visualize TRPM8 channels in cellular membranes, where bona fide menthol- and cold-evoked open states are captured. We identify a novel ‘semiswapped’ architecture in which interdigitation of channel subunits is substantially rearranged following repositioning of the S6 transmembrane helix and elements of the pore region.”

As noted above, exchange between fully- and semi-swapped configurations does not constitute the final gating step *per se*, but the semi-swapped configuration is an important part of the cold-evoked gating pathway. As such, we do not feel that the abstract is misleading, especially with the addition of the new sentence noted above, and the new schematic included in Fig. 6.

In a manuscript entitled “The structural basis of cold sensitivity”, it is not too far fetched to draw the conclusion that this newly identified structure represents the novel, relevant cold sensing state.

That said – if it is not cold activation/gating, what is the role of this newly identified structure?

As noted above, the novel semi-swapped configuration constitutes part of the cold-evoked gating pathway. Furthermore, our HDX and structural analyses show that menthol favors the semi-swapped state. Thus, we can confidently say that the semi-swapped state is an important segue to channel gating by chemical or thermal stimuli. In any case, we have modified the title to emphasize the combined structural and energetic aspects of the study, which as the other referees agree, is an important advance in the field.

General remarks:

Provision of HDX data in SI: It is in line with the common practice in the field to provide a table with the information of at least the most important peptides (not all of them), their sequences, exchange rate, coverage etc. Seeing the importance of this data for the current manuscript and interpretations therein this is not unreasonable request.

As suggested by Referee 4, we now present data in the format recommended by the HDX-MS community guidelines. We have also included detailed information for the most important peptides, namely those corresponding to the TRP helix and pore helix (in SI).

Apologies for not having made this clearer in the last round, however, the title of the paper is incredibly broad, but conclusions in the paper can really only be drawn for TRPM8, possibly even only for TRPM8 from a few species. While it seems possible/likely that the physical basis of cold activation involves heat capacity changes also in other channels, and that the structural transition from pi- to alpha-helical is preserved, this remains to be shown.

TRPM8 is the major physiologic cold sensor in animals and thus we feel that the revised title is not too far-fetched for a broad readership. Moreover, our main message is that cold stabilizes a dynamic region within TRPM8, thereby driving gating transitions, which is likely a general principle underlying thermosensitivity at the protein level. Certainly, pi- to alpha- transition and gating by phenylalanine is specific for TRPM8, as far as we know by now. In any case, we are happy to defer to the editor in this matter.

Transition between swapped and semi-swapped states: the new movie (Sup 1) is appreciated and will certainly help readers to underscore that the transition might be feasible without disassembly of the tetramer, however this would be even clearer if not shown as a space filling model.

We appreciate this suggestion. As the referee is aware, the movie was generated using UCSF Chimera by interpolating between two well defined end points (fully- and semi-swapped configurations). As such, the interpolated intermediates in between these two end points are not real, but hypothetical. We are concerned that a movie generated from the ribbon diagram, although providing a clearer graphical view, can mislead readers about the details of possible intermediate states.

Minor points:

Figure 4d: In the subfigure, data for dH , TdS and dG are presented for three temperature intervals, but the temperature used to calculate TdS is not mentioned, please indicate it e.g. in the figure legend.

We now add the standard condition under which TdS was calculated, namely 25°C and 1 atm pressure, in the figure legend. This is also clarified in the method section.

Page 5: The reference to Extended Data Figure 3 seems misplaced, as this figure focuses on the pKa calculations of PmTRPM8 in the two configurations.

Thank you bringing this to our attention. We have now corrected this figure citation.

The numbering of residues in model 8 (structure files) differs from other models by 4 residues, maybe because it's the outer pore chimera? This could be clarified.

Thank you for pointing out this error. We have now fixed the numbering in the PDB.

There is no description in the material and methods how the conformational morph between swapped and semi-swapped states was carried out.

We now note in the methods section that the movie was generated with UCSF Chimera.

Referee #3 (Remarks to the Author):

I co-reviewed this manuscript with one of the reviewers who provided the listed reports.

Referee #4:

Remarks to the Author:

This study by Choi et al. aims to understand the molecular basis of thermosensitivity in TRPM8, the ion channel responsible for cold perception and menthol sensitivity in mammals. The authors address a long-standing challenge in the field - elucidating the structural mechanism and thermodynamic changes that underlie temperature-dependent gating, through a combination of cryo-EM and hydrogen-deuterium exchange mass spectrometry (HDX-MS).

A key strength of the work lies in the study of TRPM8 channels within native cellular membranes, which allows the authors to capture authentic cold- and menthol-evoked open states. The identification of a substate with a "semi-swapped" subunit architecture represents an important structural insight, suggesting that intersubunit rearrangements and movements in the S6 helix and pore helices are central to activation. Complementary HDX-MS data, albeit not performed in native cellular membranes but detergent, highlight specific regions, particularly the pore and TRP helices, that undergo the most pronounced temperature-evoked changes. The comparative analysis between human TRPM8 and a menthol-sensitive but cold-insensitive avian orthologue serves to support the proposed gating mechanism and reinforces the physiological relevance of the structural observations. The authors also propose a free energy landscape model which strengthens understanding of the temperature-dependent gating in TRP channels more broadly.

With an eye on my primary area of expertise, the HDX-MS experiments themselves appear to have been well conducted, albeit with one important problem. Also, the analysis is mostly sound but I also think there are some issues wrt. the interpretation of bimodal mass envelopes that needs to be addressed.

Overall, I find the paper interesting as it offers a compelling structural and thermodynamic explanation for TRPM8 cold activation, setting a new benchmark for mechanistic studies of thermosensitive ion channels. Its integrative approach of both Cryo-EM and HDX-MS provides a useful method blueprint for dissecting other temperature-sensitive membrane proteins.

We thank the Referee for his/her enthusiastic comment about our study and appreciate the constructive comments that help to improve our presentation.

Comments:

1. The authors should follow the HDX-MS community guidelines (Masson et al. 2019) and specifically include HDX Data tables so that the data is more accessible – and deposit the data in a database (e.g. PRIDE).

We thank the Referee for ensuring that our HDX data are freely available. Indeed, the HDX Data table was included in our original submission as Extended Data Table 2. Furthermore, all raw mass spectrometry files and HDEaminer projects containing processed HDX data have been deposited to the ProteomeXchange Consortium via the PRIDE partner repository with the dataset identifier PXD064468, as stated in the Data Availability statement in our original submission. This data will be freely available upon publication, but we include reviewer login info below so that the Referee can access the data if s/he wishes:

Username: reviewer_pxd064468@ebi.ac.uk

Password: 4HXQNZf2wWvT

Also, the authors need to clearly define their threshold for significant changes in HDX between states – based on a proper assessment of error. I can advice that the authors adopt a hybrid testing approach (Volcano plot) – see Hageman and Weis. Otherwise follow the guidelines in Masson et al. 2019.

We thank the Referee for this suggestion. We have now performed the hybrid testing approach according to Hageman and Weis. The thresholds for significant changes are now updated in the HDX data table (Extended Data Table 2). The volcano plots are included in Supplementary Information Fig. 8.

2. Assessment of error in the HDX-MS experiments: The HDX-MS data shown appears to be the average of triplicate measurements conducted on two HDX-MS platforms using two different MS instrument in two labs with slightly different LC times. These are then per definition not

replicate measurements – and should not be averaged nor used to assess error. This has to be addressed.

I understand there may be practical reasons for doing the experiments this way – and the fact that the authors state that two datasets show high agreement add confidence to the overall findings. But this is nonetheless not a suitable way of assessing error in the data presented. I can suggest some ways to mitigate this. The authors could record two more replicate data sets at UCSF and show results (and error) from the three replicate measurements recorded at UCSF. Or alternatively, and less labor intensive, show data for only one of the replicates in the manuscript proper (perhaps the third replicate at UCSF) and then in new independent experiments assess the error of this experiment (be performing replicate measurements of some time-points using the same HDX-MS setup at UCSF). The authors could use these to estimate error and calculate a global confidence threshold for significant changes etc. The authors are referred to the HDX-MS community guidelines (Masson et al. 2019).

We appreciate the Referee's concern about our replicates. We understand that the main concern is about systematic errors generated from data collected on two instruments. To estimate this error, we recorded a 4th replicate at UCSF for human TRPM8 at 4°C, making for one replicate from UC Berkeley in this condition and 3 replicates from UCSF. When we compared the mean standard deviation for replicates 1-3 (UCB, UCSF, UCSF) and replicates 2-4 (all UCSF) for all peptides and all time points, we did see a slight increase in repeatability ($\sigma_{\#D}$ changed from 0.14 to 0.12 Da; $\sigma_{\%D}$ changed from 1.79% to 1.59%), but this difference is well within the global confidence threshold for significant changes of 0.36 Da (or 0.34 Da when calculated from replicates 2-4) shown in Extended Data Table 2. We also included a more detailed comparison of repeatability for replicates 1-3 and 2-4 in the Supplementary Information Fig. 8. Because collecting a fourth replicate for all conditions will be very time-consuming, we only did so for one condition to demonstrate that variance between runs using the same instrument (repeatability) is not substantially greater than inter-instrument variance (reproducibility). Because this analysis shows that the datasets collected from UCB and UCSF do not differ statistically, we prefer to include all data from both instruments as this provides stronger evidence for the robustness of our analysis, even though the data variation is slightly larger than if we were to omit data from UCB.

3. About the bimodal mass envelopes in the HDX-MS data: The authors state that "the pore and TRP helices each showed clear bimodal mass envelopes that increased in mass over labeling time, suggesting the coexistence of two conformational populations..." and "suggesting that these two populations undergo interconversion on the HDX-MS time scale (i.e., minutes to hours)": this section and other sections discussing EX2 vs. EX1 kinetics needs to be revised somewhat as I do not agree with the interpretation in SI Fig. 3. To my eye, the data shown in SI Fig.3b is EX1 or mixed EX1 kinetics and not EX2. As the authors correctly point out, the intensity of the two mass envelopes change during the HDX timecourse, and this is a hall-mark of EX1 kinetics, see for

instance https://pubs.acs.org/doi/pdf/10.1016/j.jasms.2006.05.014?ref=article_openPDF

We thank the referee for raising this question about the difference between EX1 and EX2. To definitively identify whether our profiles conform to an EX1 or EX2 mechanism, we performed HDX measurements of human TRPM8 at 4°C at different pD. In the EX2 limit, every unit increase in pD should increase the exchange rate by 10-fold, while in the EX1 limit, the exchange rate should not be dependent on pD. Our analysis shows that the exchange rates of both mass envelopes scale with k_{chem} (Supplementary Information Fig. 3 and 4b), suggesting that the two mass envelopes are both exchanging in the EX2 limit. The pD dependence of exchange rates show a slope of $\log(k)$ vs pD = 0.7-0.8, which is slightly deviated from a perfect EX2 scenario where the slope is 1 ($\Delta\log k = \Delta\text{pD}$). We infer this to be evidence of pH biasing the TRPM8 conformation in addition to affecting k_{chem} , based on our functional analysis of avian TRPM8.

Moreover, in our data, both mass envelopes exhibit an increase in mean m/z over HDX time with an exchange rate between zero and k_{chem} (Supplementary Information Fig. 4a&b) and therefore are inconsistent with the requirement of $k_{\text{close}} \ll k_{\text{chem}}$ in EX1 limit, where mass envelopes should be un-deuterated and deuterated at a level consistent with k_{chem} , respectively. This is because in EX1 every H-bond opening event results in full deuteration of the exchanging unit, while H-bonds that have not undergone opening do not show deuteration, and therefore mass envelopes with partial deuteration are not observed (assuming $t_{\text{HDX}} \gg 1/k_{\text{chem}}$). These features of mass spectra resulted from EX1 is different from what we see, and together with the new pD dependence data noted above, leads us to conclude that our data conform to an EX2 mechanism.

We agree that, in the mass spectra of our original SI Fig3b, the increasing m/z over time may not appear clearly by eye, as the spectra shown are data collected at 22°C with some merging of the two binomial distributions. To more clearly illustrate the exchange pattern, we have edited SI Fig3b (now as Supplementary Information Fig. 5 & 6) to show data at 4°C and the raw fittings from HX-Express3. In these spectra, the separation of two mass envelopes is clearer and thus better demonstrates the exchange behavior of the two mass envelopes in our bimodal peptides and how they are different from EX1.

I think the two examples of raw data shown in SI Fig. 3 appear to be EX1 or mixed EX1 kinetics - not EX2 kinetics.

As stated in our response above, our data are not compatible with EX1 kinetics based on mass envelope centroids shift and significant pD dependence.

Regarding mixed kinetics, we agree that in rare cases, long peptides could contain a portion of residues that exchange in EX2 kinetics and a portion that exchange in EX1 kinetics. This would result in mass spectra similar to what was observed with TRPM8's pore helix and TRP helix (that the two mass envelope centroids shift over time along with an intensity shift). However, our peptides showing bimodality are generally short, with the longest being peptide 901-911 (human TRPM8) that contains 8 exchangeable residues (excluding the N-terminal two residue and proline). Thus, it'll be a rare case that a very small number of residues undergo EX1 exchange under physiologically relevant conditions (pD and temperature). If so, then the difference between the #D of the two mass distributions will be constant over HDX time course (assuming $t_{\text{HDX}} \gg 1/k_{\text{chem}}$) and independent of pD, as this $\Delta\#D$ will represent the number of residues exhibiting EX1

kinetics within the peptide. However, our data show that $\Delta\#D$ of the two mass distributions is not constant over HDX time, and the changes of $\Delta\#D$ in response to pD changes can be mostly attributed to changes in k_{chem} , further demonstrating that the two mass distributions are exchanging via EX2 kinetics independently (Supplementary Information Fig. 4c).

The authors write that interconversion of the conformational states occur on the min-hrs timescale of the HDX experiment – I agree that this is very likely true – and that would certainly be expected for most protein transitions between two distinct states. But that is not why EX1 kinetics would be observed. EX1 is observed if two conformational states with distinct HDX behavior exist in solution that interconvert with a very slow rate relative to the chemical exchange rate (i.e. $k_{\text{cl}} \ll k_{\text{ch}}$). Thus, it specifically denotes that k_{cl} is "unusually" slow.

We are not exactly sure what the Referee is asking in this comment; it seems that s/he is linking the interconversion between fully and semi-swapped states with the EX1. We apologize if our interpretation of this comment is wrong, but the interconversion between the two structural states has nothing to do with EX1. We agree that EX1 is characterized by $k_{\text{close}} \ll k_{\text{chem}}$, but we would like to note that k_{close} represents the closing rate of the H-bonds, which is not related to the interconversion of conformational states we observed structurally.

The authors should try to extrapolate k_{op} (where possible, for instance using HX-Express software) and see if k_{op} is correlated between different segments where EX1/mixed EX1 is observed. This could be of functional significance – see for instance Merkle et al. Science Advances 2018. If the authors wanted to further investigate the slow time conversion between two conformational states representing fully – and semi-swapped states then this could be done using a pulse-labeling HDX experiment.

Importantly, I think the interpretation of two conformational states representing fully – and semi-swapped states can still be reconciled with the observation of EX1 kinetics. However, I am aware that some of the assumptions made to quantify thermodynamic parameters from HDX-MS are then no longer valid.

We appreciate the referee's suggestion to calculate the interconversion rates between two mass distributions. The interconversion timescale is much longer than the timescales for TRPM8 opening (~ milli-second) or desensitization (~ second), and therefore exactly quantifying this interconversion rate to calculate the free energy barrier between the fully- and semi-swapped states will not provide much more biologically relevant conclusions. A pulse-labeling HDX experiment will not be suitable either because the interconversion between fully- and semi-swapped states occurs at equilibrium. Regardless, this calculation will be challenging as the overlap of the two mass distributions makes it difficult to accurately quantify the population change over HDX time. We have included the raw fitting using HX-Express3 in Supplementary Information Fig. 5 & 6.

4. The authors should clearly substantiate the need to perform the elaborate quantification of thermodynamic parameters from the HDX data – which makes several assumptions that one could question or argue. What extra value is precisely achieved? Fig. 6 does not needs this I

believe. I think the paper would benefit from a more rigorous discussion of the assumptions made in this analysis and what precise new insights are obtained. It is a little unclear to me at least.

The elaborate quantification of thermodynamic parameters is motivated by a longstanding theory of thermo-sensation by TRP channels, which suggests that the opening of thermosensitive TRP channels can be solely explained by unusual changes in molar heat capacity (i.e., negative ΔH° for cold-activated TRPs) (PMID: 22109551). A number of experimental studies have also ascribed the robust temperature sensitivity of TRP channels to changes in heat capacity, yet these studies were limited to assessing global energetics. Analysis of ΔH° and heat capacity at localized regions across the protein structure was never achieved. Therefore, we closed this gap by performing extensive thermodynamic analysis with our HDX data. We described this background in the Introduction of the manuscript.

Fig. 6 does not contain detailed information specifically from the thermodynamic analysis, as it is only a schematic describing the relationship between fully- and semi-swapped states, as well as the distinction between avian and human TRPM8.